# The Greuter Herbarium in Palermo: An Inventory of Its Type Specimens Available Online, with Some Thoughts on Type Terminology (Occasional Papers from the Herbarium Greuter, N° 5) [note 1]

**DOI:** 10.3390/plants13081086

**Published:** 2024-04-12

**Authors:** Werner Greuter

**Affiliations:** 1Botanischer Garten, 14195 Berlin, Germany; w.greuter@bgbm.org; 2Herbarium Mediterraneum Panormitanum, 90133 Palermo, Italy

**Keywords:** Greuter Herbarium, Palermo, nomenclatural types, type categories, first-step typification, second-step typification

## Abstract

The separately stored type herbarium of the Herbarium Greuter in Palermo is comprised of 339 sheets, corresponding to 336 specimens pertaining to 328 different names. Material from the Mediterranean area, especially Greece, predominates, followed by that from the Caribbean (Cuba) and Australia. The list includes transcribed label data and links to the digital specimen images and to the protologue texts. A new type of terminology is introduced, with the terms “first-step holotype” and “second-step holotype” designating type categories parallel to the similar terms already in use for lectotypes, and the phrase “detailed here” is used as an equivalent to “designated here” in second-step typification.

## 1. Introduction

In December 2008, the author donated his personal herbarium and botanical library to the Università degli Studi in Palermo to become an autonomous part of the university’s Herbarium Mediterraneum Panormitanum (PAL), kept separately under the official designation PAL-Gr. The herbarium was then estimated to comprise c. 95,000 specimens and included the historical French herbaria of Father Pierre Gave (1846–1916) and Dr. Antoine Bras (1802–1883). Subsequently, by exchange, purchase, and gift, c. 150,000 specimens have been added that are currently owned by the Foundation Herbarium Greuter and are destined to be transferred eventually to and gradually included in the PAL-Gr herbarium.

PAL-Gr hosts specimens from all over the globe, with a neat predominance of material from the Mediterranean countries (Greece in the first place) and including sizeable portions of Australian, South African, and Caribbean material, reflecting the donor’s main areas of interest. Type specimens, in so far as they were identified as such, are kept separately for their better safeguarding. This type herbarium has been digitized, and the high-resolution images are being made freely available and searchable and the high-resolution images are being made freely available and searchable via the University of Palermo Botanical Garden’s Virtual Herbarium (https://herbarium.unipa.it/herbarium_vsimple_en_ma.asp, accessed on 6 March 2024).

## 2. Material and Methods

This paper lists the contents of the PAL-Gr type herbarium. The list provides the following information: (1) the name typified, with reference to its place of valid publication, and, when the protologue is available online, a link (URL) to it; protologues that could not be traced on the internet have been digitized anew and can be similarly accessed through an apposite link; (2) a limited number of synonyms, especially when the currently accepted name differs from the one being typified; (3) the kind of type (such as holotype, neotype, etc.); (4) the PAL-Gr accession number; (5) the country of origin of each specimen, followed, between quotation marks, by the collecting-site data as they appear on the label; (6) the collection date, collector name, and collecting number; and (7) the whereabouts and status of any duplicates of which the location is known.

## 3. Discussion

**Type category terms**. The International Code of Nomenclature, in its current edition (Turland & al. 2018 [1]; hereafter the *Code*), recognizes and defines the following type categories: holotype, syntype, lectotype, paratype, neotype, and epitype, plus the duplicates of each of these types, using the prefix “iso-”. Paratypes having no immediate relevance for nomenclatural purposes are not taken into consideration, nor even mentioned, in the present inventory.

The system of type categories thus defined is, in a general way, consistent and has been working in a satisfactory way for many years and through several successive editions of the *Code*. There is, however, one major illogicality embedded in it. The types of various categories, as defined in Art. 9, all consist of one element: a single specimen (or sometimes illustration). However, as stated in Art. 40.2, a type may also consist of more than one specimen, provided they all belong to the same gathering. That provision applies in the case of names published from 1958 onward, when an indication of the type became a prerequisite for the valid publication of names of new genera and infrageneric taxa. However, even before 1958, cases abound in which reference to multiple specimens of a single gathering is referred to in a protologue, and these specimen runs have then been generally (if inaccurately) accepted as constituting the holotype.

Analogous situations frequently arise with types of other categories, but these are explicitly covered in Art. 9.17. The *Code*, in Art. 9 Ex. 14, in such cases uses the phrase “first-step” lectotype for the multiple-specimen “type” initially designated and permits subsequent selection of a “second-step” type from among the elements of the “first-step” [lecto-]type. This introduces a disparity of treatment between holotypes and other type categories, in so far as with the former, when the original type consists of more than one specimen, all of these are considered to be syntypes (Art. 40 Note 1). As there is no provision parallel to Art. 9.17, restricting subsequent lectotype designation to one of these syntypes, other isosyntypes, even if not seen or cited by the original author, are under Art. 9.12, eligible as types on equal footing with the syntypes proper.

One of the purposes of the following list has been to establish the correct categories of the types concerned. Surprisingly, it turned out that close to one-third of the names concerned had “holotypes” consisting of more than one specimen. As a consequence, I felt it was very desirable to devise a satisfactory way to address this situation in simple and easily understood terms. I, therefore, propose to designate such non-unitary “holotypes” as “first-step holotypes” by analogy to what the *Code* has been using for lectotypes of a similar kind and, by the same analogy, to designate as “second-step holotypes” the lectotypes obtained by selecting one of the first-step holotype elements by analogy to what Art. 9.17 (and not just 9.12) provides. Also, in this kind of situation, I shall use the phrase “specified here” as a direct equivalent of “designated here” to meet the requirements of Art. 7.11. These suggestions are not covered by the provisions of the current *Code*, but they are compatible with them. If found to be useful and generally acceptable, they may hopefully be reflected in the text of some future *Code* edition (the date for submitting proposals for the next edition has unfortunately passed).

## 4. Results: The PAL-Gr Type Inventory

***Acantholimon albanicum*** O. Schwarz and F. Mey. in Haussknechtia 3: 31. 1987. [https://zs.thulb.uni-jena.de/receive/jportal_jparticle_01322116 (accessed on 6 March 2024)].

**Isotype: PAL-Gr 52738**—Albania, “Korca, Mali i Moravës, bei Drenova, ca. 1100–1200 m, Serpentin,” 12. Sep 1961, *F. K. Meyer 6151* (Holotype: JE) (other isotypes: JE [x3]).

***Achillea absinthoides*** Halácsy in, Denkschr. Akad. Wiss. Wien, Math.-Naturwiss. Kl. 61. 243. 1894. [https://www.biodiversitylibrary.org/page/7221937#page/333/mode/1up (accessed on 6 March 2024)].

**Isotype: PAL-Gr 15480**—Greece, “Epirus orientalis. In rupestribus calcareis regionis alpinae inferioris mt. Tsumerka supra pagum Vulgarelion loco dicto Spathes. Alt. 1500 m,” 8 Jul 1893, E. de Halácsy, Iter graecum secundum a. 1893 (holotype [specified here]: WU in Halácsy Greek hb.) (other isotypes: B [×3], JE, K, LD, S).

***Agrostis merxmuelleri*** Greuter and H. Scholz in Mitt. Bot. Staatssamml. München 16, Beih.: 23 1980. [https://www.biodiversitylibrary.org/item/52055#page/617/mode/1up (accessed on 6 March 2024)].

**Holotype: PAL-Gr 24749**—Greece, “Epirus/Macedonia (distr. Konitsa/Kastoria), montes Grammos in vertice orientali, alt. 2350–2442 m. In petrosis graminosis, solo flyscheo,” 14 Aug 1978, *W. Greuter 14360* (Isotypes: ATH, B, C, G, M, UPA).

***Allium akirense*** N. Friesen and Fragman in Phytotaxa 173(2): 143. 2014. [www.academia.edu/7464094/A_new_Allium_from_section_Molium_from_Israel_A_akirense_Amaryllidaceae_ (accessed on 6 March 2024)].

**Isotype: PAL-Gr 123563**—Israel, “Philistean Plain, hill near kibbutz Giv’at-Brenner, Batha and Garigue on calcified sandstone, 31.867130 N, 34.807520 E, Alt. 70 m.”, 2 Apr 2013, O. Fragman-Sapir (holotype: HUJ; other isotypes: HUI, OSBU).

***Allium antonii-bolosii*** (‘*A. Bolosii*’) P. Palau in Anales Inst. Bot. Cavanilles 11: 485. 1953 [? = ***Allium hirtovaginatum*** Kunth]. [25723].

**Isotype: PAL-Gr 25723**—Spain, Balearic Islands, “Cabrera (Baleares), c.d.” [Protologue: Habitat in fis[s]uris angstissimis rupestribus calcareis ad N. vel ad W orientatis. Floret Jul-Sept.], *Palau* (dupl. ex BC; holotype [specified here]: herbarium, Col·legi Oficial de Farmacèutics de les Illes Balears, Palma de Mallorca).

***Allium greuteri*** Brullo and Pavone in Willdenowia 13: 115. 1983 [https://www.jstor.org/stable/3995985?seq=2 (accessed on 6 March 2024)].

**Holotype** [specified here]**: PAL-Gr 35235**—Libya, “Wadi el-Bab,” 16 May 1981, *Brullo and Furnari* (isotypes: B, CAT [2×], FI, G [2×], MPU).

***Allium integerrimum*** Zahar. in Ann. Mus. Goulandris 3: 90. 1977 [19137].

**Holotype: PAL-Gr 19137**—Greece, “Macedonia occ., prov. et distr. Pieria: in colle arci (katro) Platamon, alt. 50–100 m. In rarioribs dumetorum *Quercus cocciferae*. Flores rosei,” 7 Jul 1971, *Greuter 9192* (Isotype: ATH).

***Allium lopadusanum*** Bartolo, Brullo, and Pavone in Willdenowia 16: 89. 1986 [https://www.jstor.org/stable/3996306 (accessed on 6 March 2024)].

**Isotype: PAL-Gr 35757**—Italy, Sicily, “Lampedusa, Vallone Madonna,” 25 Jun 1985, Brullo (holotype: CAT; other isotypes: B, CAT, FI).

***Allium melanogyne*** Greuter in Willdenowia 39: 342. 2010 [https://www.jstor.org/stable/20699183?seq=8 (accessed on 6 March 2024)].

**Holotype: PAL-Gr 45677**—Greece, “Nomos Evros, Eparhia Soufli, 1 km S of Dhadhia, 100 m, 41°07′20″ N, 26°13′ E. Grassland on micaschist hill. Tepals cream with pale green midnerve, anthers light yellow, ovary purplish black,” 13 Jun 1992, *Greuter 23311*, *Zimmer*, *Phitos*, *Kamari*, *Anagnostopoulos, and Athanasiou* (Isotypes: B, SEV).

***Allium optimae*** Greuter in Bocconea 25: 106. 2012 [https://www.herbmedit.org/bocconea/25_005.pdf#page=102 (accessed on 6 March 2024)].

**Holotype: 51792 PAL-Gr**—Greece, “Nom. Lakonia, Ep. Epidhavros Limiras: Bay of Palea Monemvasia, alt. 0–2 m. 36°44′00″ N, 23°02′00″ E. Sandy beach and salt marshes,” 8 Jun 1995, *Iatrou & al.* in OPTIMA Iter Mediterraneum VII, #1792. (Isotype: none).

***Allium phalereum*** Heldr. and Sart. in Atti Congr. Int. Bot. Firenze 1874: 233. 1876 [***Allium staticiforme*** Sm.] [https://books.google.it/books?id=EPEUAAAAYAAJ&pg=PA227&hl=es&source=gbs_toc_r&cad=2#v=onepage&q&f=false (accessed on 6 March 2024)].

**Lectotype (designated here): 66270 p.p. [excl.sterile leafy plant] PAL-Gr**—Greece, “in arenosis maritimis Phaleri, flor. d. 21 Jun 1873, *Heldreich* in Baenitz, Herbarium europaeum.” (Isolectotypes: numerous; paralectotypes: same data but “folia d. 26 Mar 1874,” mixed with the foregoing).

***Allium tardans*** Greuter and Zahar. in Biol. Gallo-Hellen. 6(1): 51. 1975 [17519].

**Holotype: PAL-Gr 17519**—Greece, Crete, “Ep. Temenos: ad cacumen montis Júktas. 700 m. In praeruptis rupestribus calcareisadoccidentem spectantibus parcissime,” 14 Sep 1966, *Greuter 7519* (Isotype: none).

***Alvaradoa amorphoides*** subsp. ***caribaea*** A. Noa in Revista Jard. Bot. Nac. Univ. Habana 40: 90. 2019—[https://revistas.uh.cu/rjbn/article/view/6540/5550 (accessed on 6 March 2024)].

**Isotype: PAL-Gr 63251**—Cuba, “Prov. Villa Clara, Sagua la Grande. Mogotes de Jumagua. Vegetación cársica de mogotes. 50–88 m s.m.,” 10 Nov 2011, *Noa*, *Pérez-Lobregón and Oliver* in UCLV 10024 (holotype: ULV [specimen shown in Revista Jard. Bot. Nac. Univ. Habana 40: 93, in Fig. 5]; other isotypes: B, HAC, HAJB, JE, ULV).

***Alyssum doerfleri*** var. ***parnassicum*** Greuter in Candollea *2*9(1): 140. 1974 [=***Alyssum doerfleri*** Degen] [https://www.e-periodica.ch/digbib/view?pid=can-002%3A1974%3A29%3A%3A4#148can-002%3A1974%3A29%3A%3A4#148 (accessed on 6 March 2024)].

**Holotype: PAL-Gr 15974**—Greece, “Sterea, prov. Viotia, distr. Levadhia: in latere austro-occidentali montis Parnassos supra Arahova, alt. 1400 m. In clivis asperis rupestribus calcareis meridiem spectantibus. In scansilibus sole illustribus,” 15 Jun 1963, *Greuter 5974* (Isotypes: G, MEL, SALA, UPA, Z).

***Androcymbium rechingeri*** Greuter in Candollea 22: 248. 1967 (≡***Colchicum rechingeri*** (Greuter) J. C. Manning and Vinn.) [www.e-periodica.ch/digbib/view?pid=can-002%3A1967%3A22%3A%3A172#265 (accessed on 6 March 2024)].

**Holotype: PAL-Gr 14667**—Greece, Crete, “Ep. Kissamoas: W-Teil der Insel Elafonìsi, Kuppe beim Leuchtfeuer, 10 m.ü.M., Kalk. Auf Flugsand auf felsigem Grund; stark abweichende neue Sippe des Rassenkreises,” 4 Jun 1962, *Greuter 4667* (Isotypes: G, HUJ, LD, W, Z).

***Anthemis pignattiorum*** Guarino, Raimondo and Domina in Pl. Biosystems 147: 821. 2013 [https://www.tandfonline.com/doi/full/10.1080/11263504.2013.829888 (accessed on 6 March 2024)].

**Isotype: PAL-Gr 58585**—Italy, Sicily, “Cavagrande del Cassibile (Siracuse), 36°97′89″ N, 15°07′95″ E, 400 m a.s.l., on vertical calcarenite cliffs,” 2 May 2012, *Guarino* “ *Raimondo* (holotype [to be specified]: PAL; isotypes: B, FI, PAL).

***Anthemis pusilla*** subsp. ***ammanthiformis*** Greuter and Rech. f. in Boissiera 13: 142. 1967 (≡***Anthemis rigida*** subsp. ***ammanthiformis*** (Greuter and Rech. f.) Greuter in Candollea 23: 263. 1968 [https://www.e-periodica.ch/digbib/view?pid=boi-001%3A1967%3A13#146 (accessed on 6 March 2024)].

**Isotype: PAL-Gr 30612**—Greece: “insula Antikythera: Potamos, in parte boreali insulae, in saxosis calc. litoreis,” 8 May 1964, *Rechinger 24367a* (Holotype [specified here]: G; isotype: W?).

***Anthemis tomentella*** Greuter in Candollea 23: 148. 1968 [https://www.e-periodica.ch/digbib/view?pid=can-002%3A1968%3A23%3A%3A4#170 (accessed on 6 March 2024)].

**Holotype PAL-Gr 13642**—Greece, Crete, “Ep. Ierapetra/Sitia: N-Hang des Berges Afèndis Kavoùsi, 1200 m.ü.M., Kalk. Auf Schutt an offenen Stellen in der Phrygana stellenw. sehr hfg.,” 19 Jun 1961, *Greuter 3642* (Isotypes: G, W, Z).

***Aralia duplex*** R. Chaves in Willdenowia 45: 37 2015. [https://doi.org/10.3372/wi.45.45103 (accessed on 6 March 2024)].

**Isotype: PAL-Gr 133293**—Cuba, “Parque Nacional Viñales, Sierra de la Caoba, ladera hacia Cañadores, Pinar del Río. Individuo único sobre carso de mogote. 225 m smn,” 3 Aug 2011, *Chaves* in HAC 43069 (holotype: HAC [specimen with flowers]; other isotypes: B, HAC, HAJB [2×], JE).

***Ardisia manitzii*** Panfet in Willdenowia 33: 173. 2003.—[https://www.bgbm.org/sites/default/files/documents/w33-1Panfet.pdf (accessed on 6 March 2024)].

**Isotype: PAL-Gr 128078**—Cuba, “Isla de la Juventud (Isla de Pinos) (Mun. especial), llanura cársica del sur: camino de Cayo Piedra (Cayo las Piedras) a Punta del Este, monte seco, calizo,” 4 Nov 1981, *Álvarez & al.* In HFC 45586 (holotype: HAJB [specimen #53, shown in the protologue, Figure 1]; other isotypes: B, JE [2×]).

***Arenaria peloponnesiaca*** Rech. f. in Oesterr. Bot. Z. 112: 186. 1965 [https://www.jstor.org/stable/pdf/43339278.pdf?refreqid=excelsior%3Ab21fa319b194f9f77a5264195fc0d751&ab_segments=&origin=&initiator=&acceptTC=1 (accessed on 6 March 2024)].

**Isotype: PAL-Gr 30751**—Greece: “Messenia, Methoni, in ruinis castelli,” 11 May 1964, *Rechinger 24861* (Holotype [specified here]: G; isotype: W?):

***Aristida neglecta*** subsp. ***breviglumis*** Catasús in Greuter and Rankin, Fl. Rep. Cuba, Ser. A, Pl. Vasc. 21A: 81. 2015 [63250].

**Isotype: PAL-Gr 63250**—Cuba, “Prov. Granma, municipio de Buey Arriba, Palmarito, área protegida Refugio de Fauna ‘Bosque de Palmarito’, al SE de Yara,” 15 Dec 2014, *Catasús and Arjona* (Holotype: HAJB #451; other isotypes: B, HAJB ##449–450, JE).

***Aristolochia merxmuelleri*** Greuter and E. Mayer in Bot. Jahrb. Syst. 107: 321. 1985 [35745].

**Isotype: PAL-Gr 35745**—“Yugoslavia,” SW-Serbia (Kosovo): Miruša sub m. Koznik—in declivibus glareosis fruticosis, solo serpentin., cca 500 m s.m.,” 20 Apr 1985, *Mayer 11071* Holotype: LJU; other isotypes: B, M).

***Arrhenatherum elatius*** subsp. ***braun-blanquetii*** P. Monts. and L. Villar in Doc. Phytosoc. 78: 13. 1974 [=***Arrhenatherum elatius*** (L.) P. Beauv.] [7110].

**Isotype: PAL-Gr 7110**—Spain, “ditione oscensis (Huesca), Aragonia, in glareosis cum *Cochlearia aragonensis* ad sepemtrionem Sierra de Guara, 1550 m alt., lapides minutas, calcareas, 1–3 cm diam.,” 4 Jul 1968, *Montserrat 2848/68* (Holotype: ?JACA).

***Asperula crassula*** Greuter and Zaffran in Willdenowia 14: 289. 1985 [https://www.jstor.org/stable/3996241?seq=21 (accessed on 6 March 2024)].

**Holotype: PAL-Gr 48357**—Greece, Crete, “Ep. Sitia N-u. NO-Seite des Kap Màvro nwl. Erimùpolis, 20–160 m.ü.M., Kalk u. pleistozänes Konglomerat. Offene *Lygeum*-Fluren, Felsritzen etc. vebr. U. stellenw. Hfg.; neue Art aus der Verwandtschaft v. *A. turnefortii*, hierher auch “*A. crassifolia* L.” b. Gandog.,” 12 May 1962, *Greuter 4446* (Isotypes: ATH, G, HUJ, LD, UPA, W, Z, hb. Zaffran).

***Asperula heteroclada*** Hausskn. in Mitth. Thüring. Bot. Vereins, ser. 2, 6: 31. 1894 [=***Asperula cynanchica*** L.] [https://www.zobodat.at/pdf/Mitt-thueringischen-Bot-Ver_NF_6_0022-0037.pdf (accessed on 6 March 2024)].

**Isotype: PAL-Gr 66334**—Italien: “supra Bordighera in sax. M. Nero [‘Alles nur überwinterte Formen von neuem blühend! Daher kleinblumiger’, Bornmüller],” 25 Apr 1893, *Haussknecht* (holotype [specified here]: JE #16649).

***Asperula lutea*** subsp. ***griseola*** Greuter in Bocconea 25: 107. 2012. [https://www.herbmedit.org/bocconea/25_005.pdf#page=103 (accessed on 6 March 2024)].

**Holotype: PAL-Gr 53217**—Greece “Nom. Arkadia, Ep. Megalopolis: NW foothills of the Taijetos range, c. 0.5 km W of Neochori, alt. c. 100 m, 37°11′00″ N 22!13′45″ E.” Mixed fir, pone and chestnut wood by a stream, on mixed substrate,” 11 Jun 1995, *Kamari & al.* in OPTIMA Iter Mediterraneum VII, #2217. (isotypes: B, BEO, BRNM, MA, SALA, UPA, W).

***Asperula majorii*** (‘*Majori*’) Barbey in Bull. Herb. Boissier 2: 243. 1894 [=***Asperula tournefortii*** Sieber ex Spreng.] [https://www.biodiversitylibrary.org/item/105264#page/267/mode/1up (accessed on 6 March 2024)].

**Isotype: PAL-Gr 6014**—Greece: “Saria ins.,” 26 May 1886, Major 544 (holotype [specified here]: FI #15355; other isotypes: FI ##15353–15354, 15356–15361 [8×], G, K, M, MPU, NY [2×], RSA, US).

***Astroloma microdontum*** (‘*microdonta*’) F.Muell. ex Benth., Fl. Austral. 4: 155. 1868 (≡***Styphelia microdonta*** (F.Muell. ex Benth.) F.Muell. [https://www.biodiversitylibrary.org/page/26123477#page/171/mode/1up (accessed on 6 March 2024)].

**Isotype: PAL-Gr 39662**—Australia, [Western Australia: “Murchison River”] (see Chapman, Austral. Pl. Name Index: 305. 1991) [*Oldfield* in *Drummond VI*] *121* (holotype [specified here]: K #277584; isotypes: BM, G, K, L, MEL).

***Asystasia noliae*** R. J. A. Puente in Revista Jard. Bot. Nac. Univ. Habana 43: 120. 2022 [https://www.jstor.org/stable/48731988?seq=2 (accessed on 6 March 2024)].

**Isotype: PAL-Gr 133292**—Cuba, “Prov. Villa Clara. Santa Clara. En los exteriores de la sala de oncología del hospital infantil ‘José Luis Miranda’. Vegetación ruderal,” 9 Jan 2022, *Puente and Castañeda* (holotype: HAJB #1256; other isotypes: B, JE, ULV [2×]).

***Aubrieta albanica*** F. K. Mey. and J. E. Mey. in Haussknechtia, Beih., 15: 62. 2011 [52737].

**Isotype: PAL-Gr 52737**—Albania: “Nemërçka, über Permet, südlich Qafa e Dhembelit, ca. 1750 m, Westhang,” 12 Jul 1959, *Meyer 3952* (holotype [specified here]: JE #16678; other isotype: JE #16679).

***Ballota nigra*** subsp. ***anomala*** Greuter in Bocconea 25: 108. 2012. [https://www.herbmedit.org/bocconea/25_005.pdf#page=104 (accessed on 6 March 2024)].

**Holotype: PAL-Gr 53554**—Greece, “Nom. Messinia, Ep. Kalamata: 6–8 km NE of Ano Amfía along the road to Poliana, alt. 600–800 m, 37°08′20″ N, 22°07′30″ E. Dry river bed and rocky limestone slopes with maquis,” 14 Jun 1995, *Kamari & al.* in OPTIMA Iter Mediterraneum VII, #2554 (isotypes: B, BEO, BRNM, MA, SALA, UPA, W).

***Betonica officinalis*** subsp. ***haussknechtii*** Nyman, Consp. Fl. Eur., Suppl. 2, 2: 251. 1890 (≡***Stachys***
***officinalis*** subsp. ***haussknechtii*** (Nyman) Greuter and Burdet in Willdenowia 15: 79. 1985 [https://bibdigital.rjb.csic.es/viewer/9913/?offset=#page=261&viewer=picture&o=bookmark&n=0&q= (accessed on 6 March 2024)].

**Isotype: PAL-Gr 66672**—Greece, “Agrapha (Dolopia veterum): in reg. inf. Pindi circa monasterium Corona, in nem. querc., alt. 3500–3370′, substr. Schist.,” 20–28 Jun 1885, *Haussknecht* (holotype [specified here]: JE #3794; isotypes: JE #3792–3793, 3795–3796, K, BR).

***Beyeriopsis cygnorum*** Müll. Arg.in Linnaea 34(1): 56. 1865 (≡***Beyeria cygnorum*** (Müll.Arg.) Baill. ex Benth. 1873). [https://www.biodiversitylibrary.org/page/119715#page/60/mode/1up (accessed on 6 March 2024)].

**Isotype: PAL-Gr 39670**—Australia, [Western Australia, “Nov. Holland. Swan River (from holotype label), [*Drummond VI*] *85″* (holotype [specified here]: G #434218; other isotypes: K #959438, MEL #114150, 114151, PERTH?).

***Bolanthus creutzburgii*** Greuter in Candollea 20: 210. 1965. [https://www.e-periodica.ch/cntmng?pid=can-002%3A1965%3A20%3A%3A242 (accessed on 6 March 2024)].

**Holotype: PAL-Gr 13733**—Greece, Crete, “Ep. Pirjotissi: NW-Hang des Berges Mavri ob der Quelle karonero, 1800–1900 m.ü.M., Tripolitsa-Kalk. Auf Feischuttflecken am felsigen Abhang, nicht selten,” 30 Jun 1961, *Greuter 3733* (isotypes: G, W, Z).

***Bolanthus intermedius*** Phitos in Bot. Chron. 1: 39. 1981 [=***Bolanthus thymifolius*** (Sm.) Phitos] [29534].

**Isotype: PAL-Gr 29534**—Greece, “Ins. Euboea: in saxosis serpentinicis et magnesiticis litoreis ad pagum Mantudi,” 23 Jun 1978, *Georgiadis 1657* (holotype: UPA); other isotypes: none traced).

***Brachypodium sylvaticum*** subsp. ***creticum*** H. Scholz and Greuter in Willdenowia 15: 30. 1985 [https://www.jstor.org/stable/3996541?seq=8 (accessed on 6 March 2024)].

**Isotype: PAL-Gr 35236**—Greece, Crete: “eparch. Sfakia: in monte Volakias, alt. 1700 m,” 3 Aug 1966, *Zaffran 3223* (Holotype: B).

***Bromus caroli-henrici*** Greuter in Ann. Naturhist. Mus. Wien 75: 83. 1972 (≡***Bromus* *alopecuros*** subsp. ***caroli-henrici*** (Greuter) P. M. Sm.) [https://www.zobodat.at/pdf/ANNA_75_0083-0089.pdf (accessed on 6 March 2024)].

**Holotype: PAL-Gr 15518**—Greece, “Regio aegaea austr., insula Karpathos: in faucibus Kounoupa ad septemtriones oropedii Lastos, alt. 600 m. In scansilibus rupium calcarearum et in herbidis inter saxa,” 23 May 1963, *Greuter 5518* (isotypes: G, HUJ, LD, M, W, Z).

***Bromus transsilvanicus*** var. ***montenegrinus*** Sagorski in Mitth. Thüring. Bot. Vereins, ser. 2, 16: 34. 1901 [https://www.zobodat.at/pdf/Mitt-thueringischen-Bot-Ver_NF_16_0033-0050.pdf (accessed on 6 March 2024)].

**Isotype: PAL-Gr 35236**—Montenegro, “In asperis calcareis ad Njegus,” Jul 1898, *Sagorski* (holotype: JE?).

***Bumelia neglecta*** Bisse and J. E. Gut. in Revista Jard. Bot. Nac. Univ. Habana 6: 21. 1985, homon. illeg. [non *Bumelia neglecta* (Cronquist) Lundell 1975] [=***Sideroxylon cubense*** (Griseb.) T. D. Penn.] [https://www.jstor.org/stable/42596610?seq=3 (accessed on 6 March 2024)].

**Isotype: PAL-Gr 123176**—Cuba, “Prov. Pinar del Río, Loma Peluda de Cayajabos, 200–400 m, IX? 1972, *Bisse* in HFC 23422 (holotype [specified here]: HAJB #273; other isotypes: B, FR, HAJB #274–278 [5×], JE).

***Bupleurum kakiskalae*** Greuter in Bauhinia 3: 250. 1967. [https://botges.ch/bauhinia/Bauhinia_3_0243-0254.pdf (accessed on 6 March 2024)].

**Holotype: PAL-Gr 17714**—Greece, Crete, “Ep. Sfakia: supra hiatum Kakískala prope fontem Linoséli, 1450 m. Ad rupes praeruptas calcáreas sole illustres haud raro, sed nil praeter folia vidi,” 11 Oct 1966, *Greuter 7714* (isotypes: E, G, K, LD, M, W).

***Buxus bissei*** Eg. Köhler in Feddes Repert. 109: 351. 1998.—[https://onlinelibrary.wiley.com/doi/epdf/10.1002/fedr.19981090503 (accessed on 6 March 2024)].

**Isotype: PAL-Gr 121244**—Cuba, “Prov. Holguín, Moa. Alrededores del Campamento Arroyo Limones. Suelo laterita,” 18 Apr 1985, *Álvarez & al.* in HFC 56156 (holotype [specified here]: HAJB #1067; other isotypes: B [2×], HAJB ##1068–1069 [2×], JE [2×]).

***Buxus leivae*** Eg. Köhler in Feddes Repert. 109: 353. 1998.—[https://onlinelibrary.wiley.com/doi/epdf/10.1002/fedr.19981090503 (accessed on 6 March 2024)].

**Isotypes: PAL-Gr 121243 and 126208**—Cuba, “Prov. Holguín, Mun. Moa. Cuchillas de Moa, Alrededores del Aserrío La Melba. Suelo arciloso ácido, charrascales altos con *Bonetia cubensis*, 450–500 m s.m.,” 28 Apr 1980, *Álvarez & al.* in HFC 42165 (holotype [specified here]: HAJB #1065; other isotypes: B [2×], HAJB ##1064, 1066 [2×], JE [2×]).

***Buxus sclerophylla*** Eg. Köhler in Wiss. Z. Friedrich-Schiller Univ. **Jena**, Mat.-Naturwiss. Kl. 31(2): 239. 1982. [123180].

**Isotype: PAL-Gr 123180**—Cuba, “Prov. Guantánamo. Mun. San Antonio del Sur: Abra Mariana, loma al este del barranco,” 9 Feb 1979, *Berazaín & al.* in HFC 39083 (holotype [specified here]: HAJB #239; other isotypes: B, HAC, HAJB ##240–242 [3×], JE ##25874–25875 [2×]).

***Calamintha montenegrina*** Sagorski in Oesterr. Bot. Z. 1903: 20. 1903 [=***Clinopodium alpinum*** subsp. ***majoranifolium*** (Mill.) Govaerts, ***Satureja majoranifolia*** (Mill.) K.Malý]. [https://www.biodiversitylibrary.org/page/28306575#page/21/mode/1up (accessed on 6 March 2024)].

**Isotype: PAL-Gr 66703**—Montenegro, “zwischen Njegus u. Cetinje,” *Sagorski* (holotype [specified here]: JE #5419; isotypes: JE #5418, MEL).

***Calendula suffruticosa*** subsp. ***greuteri*** Ohle in Feddes Repert. 85: 269. 1974 [17133].

**Holotype: PAL-Gr 17133**—Spain, “Andalusien, Prov. Granada: unterste Talenge des Rio Guadalfeo nwl. Motril, 70 m.ü.M. Steile w-exp. Felshänge, Kalk; auf Schutt nicht selten zwischen Gebüsch,” 5 May 1965, *Greuter 7133* (isotypes: HAL, W).

***Campanula aizoides*** Zaffran in Greuter, Florist. Report Cretan Area: 15. 1972 [3427].

**Holotype: PAL-Gr 3427**—Greece, Crete, “Crête à l’E des gorges de Samaria. Calcaire. Altitude 2000 m,” 16 Jul 1965, *Zaffran* (isotype: hb. Zaffran).

***Campanula creutzburgii*** Greuter in Boissiera 13: 133. 1967 (≡***Campanula drabifolia*** subsp. ***creutzburgii*** (Greuter) Fed. 1973) [https://www.e-periodica.ch/digbib/view?pid=boi-001%3A1967%3A13#137 (accessed on 6 March 2024)].

**Second-step holotype (specified here): PAL-Gr 14190**—Greece, Crete, “Ep. Temenos: Insel Dìa, Bucht Panajìa an der Südküste, 1–2 m.ü.M., Kalk. Ritzen und Löcher der Küstenfelsen, hfg. mit *Bellium minutum*,” 20 Apr 1962, *Greuter 4190* (isotypes: G, HUJ, LD, W, Z).

***Campanula flagellaris*** Halácsy in Denkschr. Kaiserl. Akad. Wiss., **Wien**. Math.-Naturwiss. Kl. 61: 246, 1894 [non Kunth 1819] [=***Campanula tymphaea*** Hausskn.] [Gr].

**Isolectotype: PAL-Gr 5487**—Greece, “Epirus boreali-orientalis. In herbidis regionis superioris m. Peristeri. Alt. 1700 m. Solo calcareo,” 14 Jul 1893, Halácsy (lectotype [designated here]: WU in Halácsy Greek hb.).

***Campanula kamariana*** Kyriak., Liveri and Phitos in Fl. Medit. 27: 132. 2017 [https://www.herbmedit.org/flora/FL27_131-136.pdf (accessed on 6 March 2024)].

**Isotype: PAL-Gr 122719**—Greece, “Nomos Lakonias, Ep. Gythiou: 1.5 km N-NW of Areopolis, east entrance of Limeni village, above the road: on limestone rocks. Alt. 20–30 m. Lat. 36°40′ N, Long. 22°22′ E,” 30 Apr 2017, *Kyriakopoulos 1321* (holotype: UPA; other isotype: ACA).

***Campanula kastellorizana*** Carlström in Willdenowia 15: 384. 1986 [https://www.jstor.org/stable/pdf/3996451.pdf?refreqid=fastly-default%3A030e0bf656c3a95746a19448a3c340d1&ab_segments=0%2Fbasic_search_gsv2%2Fcontrol&origin=&initiator=search-results&acceptTC=1 (accessed on 6 March 2024)].

**Second-step holotype (specified here): PAL-Gr 44153; and isotype: PAL-Gr 27034**—Greece, “ex insula Kastellorizo (Megiste). Insula Ro, alt. 2–100 m. Profusius in totâ insulâ,” 12 Apr 1974, *Greuter 11829* (other isotype: ATH).

***Campanula korabensis*** F. K. Mey. in Haussknechtia, Beih., 15: 149. 2011 (≡***Campanula versicolor*** subsp. ***korabensis*** (F. K. Mey.) I. Jankovic and D. Lakusic [52736].

**Isotype: PAL-Gr 52736**—Albania, “Korab, Wiesen oberhalb Radomir, ca. 1400 m, an Felsen,” 5 Aug 1959, *Meyer 4759* (holotype [specified here]: JE #16705; isotype: JE #16706).

***Campanula pinatzii*** Greuter and Phitos in Boissiera 13: 134. 1967 [https://www.e-periodica.ch/digbib/view?pid=boi-001%3A1967%3A13#138 (accessed on 6 March 2024)].

**Second-step holotype (specified here): PAL-Gr 15532**—Greece, “Regio aegaea austr., insula Karpathos: in latere occidentali montis Kali Limni prope fontem Holetra, alt. 800 m. In clivis dumulosis calcareis, solo rupestri, abunde,” 25 May 1963, *Greuter 5532* (isotypes: LJU, M, W, UPA, Z, hb. Grims).

***Capsella abortiva*** Hausskn. in Mitth. Thüring. Bot. Vereins, ser. 2, 3–4: 116. 1893 [https://www.zobodat.at/pdf/Mitt-thueringischen-Bot-Ver_NF_3-4_0096-0116.pdf (accessed on 6 March 2024)].

**Isolectotype: PAL-Gr 66358**—Greece, “(Agrapha (Dolopia veterum): in reg. infer. m. Pindi circa monasterium Koróna, in nemorosis quercinis alt 3500–3700′ ssubstratu schistoso,” 20–28 Jun 1885, *Haussknecht* (holotype [designated here]: JE #5947; other isolectotype: JE #5948).

***Carex idaea*** Greuter, Matthäs and Risse in Willdenowia 15: 25. 1985 [https://www.jstor.org/stable/3996541?searchText=au%3A%22Ursula+Matth%C3%A4s%22&searchUri=%2Faction%2FdoBasicSearch%3Fsi%3D1%26Query%3Dau%3A%2Ursula%2BMatth%25C3%25A4s%2522&ab_segments=0%2Fbasic_phrase_search%2Fcontrol&refreqid=fastly-default%3Ac921948b441ea2b8de41c8677ab29079&seq=3 (accessed on 6 March 2024)].

**Holotype: PAL-Gr 12885**—Greece, Crete, “Ep. Pirjotissi: S-Hang des Berges Màvri ob Kamàres, ca. 1400 m.ü.M., Kalk. Kermeseichenwald, bei einer Quelle,” 3 May 1960, *Greuter 2885* (isotypes:none).

***Castela greuteri*** A. Noa in Revista Jard. Bot. Nac. Univ. Habana 43: 4. 2022.—[https://revistas.uh.cu/rjbn/article/view/64/TREE (accessed on 6 March 2024)].

**Isotype: PAL-Gr 129939**—Cuba, “Prov. Guantánamo, Baracoa, parte alta del Yunque de Baracoa. Bosque siempre verde mesófilo,” 17 Apr 1986, *Arias & al.* in HFC 58862 (holotype: HAJB #1260; other isotypes: B, HAJB #1261–1263 [3×], JE [2×], ULV).

***Castela manitzii*** A. Noa in Revista Jard. Bot. Nac. Univ. Habana 43: 11. 2022.—[https://revistas.uh.cu/rjbn/article/view/64/TREE (accessed on 6 March 2024)].

**Isotype: PAL-Gr 59075**—Cuba, “Las Tunas, Jesús Menéndez, al este de Playa Herradura, complejo de vegetción de costa arenosa, 76°25′44″ W, 16°49′ N, alt. 5 m,” 27 Mar 2013, *S. Fuentes & al. 749* (holotype: ULV (specimen shown in the protologue, Figure 9); other isotypes: B, HAJB #1266–1271 [6×], JE, ULV).

***Centaurea aegusae*** Domina, Greuter, and Raimondo in Israel J. Pl. Sci. 64: 51. 2017 (≡***Centaurea panormitana*** subsp. ***aegusae*** (Domina, Greuter, and Raimondo) Guarino and Pignatti 2019) [https://www.researchgate.net/publication/315694332 (accessed on 6 March 2024)].

**Isotype: PAL-Gr 122018**—Italy, Sicily, “Island of Favignana, Mt. Santa Caterina, Scindo Passo, 37.920730° N, 12.307299° E, 100 m a.s.l., maritime carbonate cliffs,” 25 May 2016, *Domina* (holotype [to be specified]: PAL; isotypes: B, FI, PAL, RO).

***Centaurea chrysocephala*** Phitos and T. Georgiadis in Biol. and Ecol. Médit. 4(1): 4. 1978 [25628].

**Isotype: PAL-Gr 25628**—Greece, “Prov. Trikala: supra oppidum Kalampaka, ad monasterium Varlaam-Meteora; in conglomeratis,” 4 Jul 1974, *Phitos 12563* (holotype: UPA).

***Centaurea cithaeronea*** Phitos and Constantin. in Fl. Medit. 3: 273. 1993 [https://www.herbmedit.org/flora/3-273.pdf (accessed on 6 March 2024)].

**Isotype: PAL-Gr 38807**—Greece, “Nomos “Viotias, Ep. Megaridos/Thivon: Mt. Kitheron, W of an abandoned military camp; open, rocky, grazed ground, alt. 1290 m; limestone, 38°11′ N, 23°15′ E. Flowers yellow,” 12 Jul 1992, *Constantinidis 2690* (holotype: UPA; other isotype: B).

***Centaurea cycladum*** Heldr., Herb. Graec. Norm. 1902: #1652. 1902 (=***Centaurea spinosa*** subsp. ***cycladum*** (Heldr.) Hayek) [66454].

**Isolectotype: PAL-Gr 66454**—Greece, “Flora Aegaea: legimus in Cycladum ins. Mykonos ubi vulgatissima in collibus et montibus graniticis,” 15 Jul 1901, *Heldreich* in Herbarium Graecum Normale #1652 (Lectotype [designated here]: WU #67939; other isolectotypes: GB, JE [2×], LD, M, PI, W).

***Centaurea greuteri*** E. Gamal-Eldin and Wagenitz in Willdenowia 13: 324. 1984 [https://www.jstor.org/stable/3995847 (accessed on 6 March 2024)].

**Holotype: PAL-Gr 19638**—Greece, “Regio Aegaea sept., penins Sithonia: in regione Aliki inter pagos Neos Marmaras et Toroni, alt. 20 m. In fissuris et at pedes rupium marmoreo-schistosorum praeruptarum secus viam,” 18 Jun 1972, *Greuter 10514 and Rechinger 44684* (isotypes: ATH, BSB, ELVE, G, LTR, UPA, hb. Damboldt, hb. Grims).

***Centaurea grisebachii*** subsp. ***occidentalis*** in Strid and Kit Tan, Mount. Fl. Greece 2: 500. 1991 [20855].

**Isotype: PAL-Gr 20855**—Greece, “Macedonia occ., prov. and distr. Grevena: ad occidentem pagi Kranea, in latere boreo-orientali montis Simandro, loco ‘Baltsa’ vocato, alt. 1500 m. In regione sillvae *Fagi*, *Pini leucodermis* et *pallasianae*, solo ophiolithico,” 16 Aug 1974, *Charpin 11126*, *Dittrich*, *Greuter 12295 and van Auw* (holotype: C; other isotypes: G, LD, DUIS, UPA, W, hb. Dittrich).

***Centaurea mantoudii*** T. Georgiadis, Contrib. Phylogén. *Centaurea* L. (sect. *Acrolophus* (Cass.) DC.) en Grèce: 56. 1980 [=***Centaurea laureotica*** Heldr. ex Halácsy 1898] [29715].

**Isotype: PAL-Gr 29715**—Greece, “Ins. Euboea: in saxosis maritimis, propre pagum Mantoudi,” 23 Jun 1978, *Georgiadis 1723* (holotype: UPA; other isotypes: none known).

***Centaurea musakii*** T. Georgiadis in Bot. Not. 132: 311. 1979 [29709].

**Isotype: PAL-Gr 29709**—Greece, “Prov. Trikala: prope pagum Pyli, in rupestribus calc. alt. c. 200 m,” 22 Sep 1977, *Georgiadis 1732* (holotype: UPA; other isotypes: none known).

***Centaurea pangaea*** Greuter and Papan. in Bot. Not. 132: 471. 1979 [https://media.e-taxonomy.eu/flora-greece/protologue/Protologue_Centaurea%20pangaea_Greuter_&_Papanicolaou_1979.pdf (accessed on 6 March 2024)].

**Isotype: PAL-Gr 32045**—Greece, “Macedonia or., prov. Kavala, eparhia Pangeo: in crista orientali montis Pangeo, ad occidentem cacuminis Kara Göl tepe, alt. 1300 m (40°54′30″ N/24°08′ E). In clivis rupestribus marmoreis praeruptis septemtriones spectantibus,” 18 Jul 1978, *Greuter 16055* (holotype: B; other isotypes: AAU, ATH, C, DUIS, G, GOET, M, UPA).

**Centaurea panormitana** Lojac., Fl. Sicul. 2(1): 137. 1903 [xxx].

**isoneotype: PAL-Gr 67524**—Italy, Sicily, “Punta Mastrangelo, Monreale, prov. of Palermo. Coordinates: 38°03′52″ N, 13°14′37″ E, alt. 850 m a.s.l. Habitat: carbonate cliffs facing the sea,” 3 Aug 2015, *Domina* (neotype (Domina & al. in Willdenowia 46: 25. 2016 [https://www.jstor.org/stable/24753265 (accessed on 6 March 2024)]: PAL; other isoneotypes: B, BM, FI, RO).

***Centaurea pawlowskii*** Phitos and Damboldt in Veröff. Geobot. Inst. ETH Stiftung Rübel Zürich 56: 185. 1976 [https://www.e-periodica.ch/digbib/view?pid=gbi-002%3A1976%3A56#189 (accessed on 6 March 2024)].

**Isotype: PAL-Gr 25642**—Greece, “Prov. Ipiros: prope vicum Monodendrion, ad muros monast. Hagia Paraskevi, ca 1100 m,” 13 Jun 1972, *Phitos 11716* (holotype: UPA; other isotypes: none known).

***Centaurea pelia*** var. ***trigona*** T. Georgiadis, Contrib. Phylogén. Centaurea L. (sect. Acrolophus (Cass.) DC.) en Grèce: 59. 1980.

**Isotype: PAL-Gr 29737**—Greece, “Prov. Trikala: inter vicos Metsovo et Kalambaka, prope pagum Trigona,” 4 Jul 1974, *Phitos 12564 and Tsanoudakis* (holotype: UPA; other isotypes: none known).

***Centaurea poculatoris*** Greuter in Bauhinia 3: 252. 1967 [https://botges.ch/bauhinia/Bauhinia_3_0243-0254.pdf (accessed on 6 March 2024)].

**Holotype: PAL-Gr 17689**—Greece, Crete, “Ep. Sfakia: in faucibus infra pagum Asféndos, 280–320 m. In rupium calcarearum praeruptarum fissuris sole illustribus frequens. Flores desunt, sed ex verbis pastoris indigenae certe rosei,” 8 Oct 1966, *Greuter 7689* (isotypes: ATH, B, E, G, GB, K, LD, M, P, W, hb. Phitos, hb. Zaffran).

***Centaurea prespana*** Rech. f. in Ann. Mus. Goulandris 2: 55. 1975 [? = ***Centaurea soskae*** Hayek 1926] [30762].

**Isotype: PAL-Gr 30762**—Greece, “Macedonia occidentalis: distr. Phlorina: ad occidentem pagi Vrondera ad occidentem lacus Prespa ad fines Albaniae. In saxosis calc., 1300–1400 m,” 23 Jun 1972, *Stamatiadou* (holotype: W; other isotype: ATH).

***Centaurea raphanina*** f. ***interdicta*** Greuter in Candollea 31: 229. 1976 [https://www.e-periodica.ch/digbib/view?pid=can-002%3A1976%3A31%3A%3A188#236 (accessed on 6 March 2024)].

**Isotype: PAL-Gr 20089**—Greece, “Regio Aegaea orientalis, insula Psara: in colle ‘Karajianni’ ad sept. sinûs Arhondiki litoris occidentalis, alt. 50–100 m. In dumulosis spinescentibus, solo rupestri eruptivo,” 24 Apr 1973, *Greuter 11064* (holotype: ATH; other isotypes: none).

***Centaurea thasia*** Hayek in Repert. Spec. Nov. Regni Veg. Beih. 30(2): 762. 1931 [20433].

**Neotype** [designated here]**: PAL-Gr 20433**—Greece, “Regio aegaea sept., insula Thasos: in vertice montis Ipsario, alt. 1100 m. In praeruptis rupestribus schistosis septemtriones spectantibus regionis silvaticae *Pini nigrae*,” 6 Jul 1973, *Greuter 11431* (isoneotypes: ATH, C, G, H).

***Centaurium limoniiforme*** Greuter in Boissiera 13: 124. 1967 (≡***Centaurium erythraea*** subsp. ***limoniiforme*** (Greuter) Greuter in Willdenowia 11: 279. 1981) [https://www.e-periodica.ch/digbib/view?pid=boi-001%3A1967%3A13#128 (accessed on 6 March 2024)].

**Holotype: PAL-Gr 16558**—Greece, “Aegäis; nom. Attikí, Ep. Kíthira. Bei der Kirche Odijìtria söl. Livàdi, 170 m.ü.M., sandig verwitterndes Neogen. *Cistus*-phrygana, vereinzelt,” 12 May 1964, *Greuter 6558* (isotypes: none).

***Cerastium epiroticum*** Möschl and Rech.f. in Bol. Soc. Brot., ser. 2, 36: 41. 1962 [30586].

**Isotype: PAL-Gr 30586**—Greece, “Epirus: in saxosis calc. vallis fluvii Thiamis prope pagum Vrusina, 51 km W Ioannina,” 15 May 1961, *Rechinger 23385* (holotype: W #12853; other isotypes: B, E, G, GZU, MA, MO, S, WU).

***Cerastium grandiflorum*** subsp. ***serpentini*** F. K. Mey. in Haussknechtia Beih., 15: 47. 2011 [52735].

**Isotype: PAL-Gr 52735**—Albania, “Kolgecaj (Bajram Curri), an der Straße zwischen Kam und Kolgecaj, ca. 400 m. Serpentin,” 13 May 1960, *Meyer 5609* (holotype [specified here]: JE #16669; other isotype: JE #16670).

***Cerastium scaposum*** subsp. ***peninsularum*** Greuter, N. Böhling and R. Jahn in Willdenowia 32: 49. 2002 [https://www.bgbm.org/sites/default/files/documents/w32-1Greuter%2Bal.pdf (accessed on 6 March 2024)].

**Holotype: PAL-Gr 13273**—Greece, Crete, “Ep. Kidonia: H.-I. Akrotìri, NW-Grat des Hügels Sklòcha, 430 m.ü.M., Kalk. Rasenfg. eine flache, erdgefüllte Gesteinsmulde ausfüllend, streng lokal; Zwergform mit auffällig verkürzten Btn.-Stielen,” 17 Mar 1961, *Greuter 3273* (isotype: Z).

***Chamaecrista lineata*** var. ***imiasensis*** (‘*iniasensis*’) A. Barreto and Yakovlev in Novosti Sist. Vysš. Rast. 28: 93. 1991. [128079].

**Isotype: PAL-Gr 128079**—Cuba, “Prov. Guantánamo. Mun. Imías: Alto de Cotilla (al norte de Cajobabo), pinares y charrascos, suelo ultrabásico y laterita, 400–500 m,” 29 May 1982, *Bässler & al.* in HFC 47172 (holotype [specified here]: HAJB #330); other isotypes: B, HAJB #331–334 [4×], JE).

***Chamaepeuce atropurpurea*** Boiss. [& Heldr. in Boissier, Diagn. Pl. Orient., ser. 2, 3: 47. 1856, pro syn.] ex Pančić, Fl. Princip. Serbiae: 432. 1874 [6130] [=***Ptilostemon strictus*** (Ten.) Greuter].

**Isolectotype: PAL-Gr 6130**—Serbia, “In dumetis Neresina Serb. austr.,” Aug 1872, *Pančić* (lectotype (Greuter in Boissiera 22: 79. 1973): LAU; other isolectotype: BEO?).

***Chamaepeuce leptophylla*** Pau and Font Quer, sched. Impr. Iter Marocc. 1927: #691. 1928 (≡***Ptilostemon leptophyllus*** (Pau and Font Quer) Greuter) [https://www.e-periodica.ch/digbib/view?pid=boi-001%3A1973%3A22#130 (accessed on 6 March 2024)].

**Isolectotype: PAL-Gr 6006**—Morocco, “Hab. In decivibus montis Iguermalez, versus Tizzi Iffri (Atlante rhiphaeo), 1500 m alt., solo schistoso,” 24 Jun 1927, *Font Quer* in It. Marocc. 1927 #691 (lectotype [Greuter in Boissiera 22: 127. 1973]: G; other isolectotypes: BCN, BM, FI, LAU, MA, MPU [2×], RO, S).

***Clematis elisabethae-carolae*** Greuter in Candollea 20: 213. 1965 [https://www.e-periodica.ch/cntmng?pid=can-002%3A1965%3A20%3A%3A242 (accessed on 6 March 2024)].

**Holotype: PAL-Gr 14848**—Greece, Crete, “Ep. Sfakia: Karst-Kraterlandschaft in N-Teil des Talkessels Ammudzarà ndl. ob Anòpolis, 1800 m.ü.M., Tripolitsa-Kalk. In grossen Büschen mehrf. in sonst kahlen senkrechten Karren u. Schründen, intensiv duftend; neue Art,” 2 Jul 1962, *Greuter 4848* (isotypes: ATH, G, HUJ, LD, W, Z).

***Clinopodium rankiniae*** I. E. Méndez in Willdenowia 47: 173. 2017 [https://doi.org/10.3372/wi.47.47209 (accessed on 6 March 2024)].

**Isotype: PAL-Gr 122003**—Cuba, “Provincia Santiago de Cuba. Altiplanicie de Santa María del Loreto. Finca los Monieles, al este de la subestación eléctrica con paneles fotovoltaicos. Vegetación secundaria sobre suelos derivados de areniscas y conglomerados. Al borde de los farallones,” 1 Oct 2016, *Méndez*, *Rifá and Regalado* in HPC 12016 (holotype: HIPC; other isotypes: B, HAC, HAJB, JE, ULV).

***Coccoloba berazainiae*** (‘*berazainae*’) I. Castañeda in Willdenowia 43: 319. 2013 [https://doi.org/10.3372/wi.43.43213 (accessed on 6 March 2024)].

**Isotype: PAL-Gr 63252**—Cuba, “Prov. Holguín. Municipio Moa. “Concesión minera Santa Teresita” al este de Yamanigüey. Pinares y charrascales,” 31 Mar 2011, *Borsch & al. 4624* (holotype: ULV [specimen shown in the protologue, Figure 1]; other isotypes: B, FR, HAJB, JE, ULV).

***Colchicum cretense*** Greuter in Candollea 22: 246. 1967 [www.e-periodica.ch/digbib/view?pid=can-002%3A1967%3A22%3A%3A172#263 (accessed on 6 March 2024)].

**Holotype: PAL-Gr 17738**—Greece, Crete, “Ep. Kidonia: circa refugium Vólika supra Kámbi, 1350 m. In apricis graminosisque, solo saxoso calcareo (ad 2100 m. ascendens),” 15 Oct 1966, *Greuter 7738* (isotypes: ATH, B, E, FI, G, GB, K, LD, LE, M, P, S, UPA, W, hb. Zaffran).

***Commersonia microphylla*** Benth., Fl. Austral. 1: 244. 1863 (≡***Androcalva microphylla*** (Benth.) C. F. Wilkins and Whitlock 2011). [https://www.biodiversitylibrary.org/page/26216983#page/306/mode/1up (accessed on 6 March 2024)].

**Isotype: PAL-Gr 39667**—Australia, [Western Australia, “between Moore and Murchison R^s^”] (from holotype label), [*Drummond VI*] *98* (holotype [specified here]: K #190019; other isotypes: MEL, NSW, PERTH).

***Convolvulus argyrothamnos*** Greuter in Bauhinia 3: 251. 1967 [https://botges.ch/bauhinia/Bauhinia_3_0243-0254.pdf (accessed on 6 March 2024)].

**Holotype: PAL-Gr 17802**—Greece, Crete, “Ep. Ierapetra: in faucibus inter Christòs et Metaxochóri, 450 m. In rupium calcarearum praeruptarum sole illustrium fissuris vix accessibilibus rarissima,” 27 Oct 1966, *Greuter 7802* (isotypes: E, G, LD, W).

***Crepis baldaccii*** subsp. ***carpini*** Greuter in Candollea 30: 328. 1975 [https://www.e-periodica.ch/digbib/view?pid=can-002%3A1975%3A30%3A%3A272#347 (accessed on 6 March 2024)].

**I*s*otype: PAL-Gr 20660**—Greece, “Epirus, prov. Ioannina, distr. Dhodhoni: montes Timfi, in latere boreo-orientali montis Astraka, alt. 1900–2000 m. Ad rupes calcareas praeruptas,” 21 Aug 1974, *Charpin 11336*, *Dittrich*, *Greuter 12508 and von Auw* (holotype: G; other isotypes: LD, UPA, W, hb. Dittrich).

***Crocus boulosii*** Greuter in Candollea 23: 45. 1968 [https://www.e-periodica.ch/digbib/view?pid=can-002%3A1968%3A23#66 (accessed on 6 March 2024)].

**Holotype: PAL-Gr 4055**—Libya, “Marwah, Gebel Akhdar,” 23 Jan 1967, *Boulos 1380* (isotypes: CAI?, K).

***Cynoglossum marifolium*** Roxb., Fl. Ind. 2: 8. 1824 (≡***Bothriospermum marifolium*** (Roxb.) DC. and A. DC. 1846) [https://www.digitale-sammlungen.de/en/view/bsb10303028?page=22,23 (accessed on 6 March 2024)].

**Lectotype** (Greuter in Fl. Medit. 25 [Spec. Iss.]: 161. 2015) [https://www.herbmedit.org/flora/FL25SI_157-166.pdf]): **PAL-Gr 62676**):—original watercolor drawing of *Cynoglossum marifolium* by Roxburgh (isotype: K).

***Daucus conchitae*** Greuter in Willdenowia 8: 574. 1979 [https://www.jstor.org/stable/3996168 (accessed on 6 March 2024)].

**Holotype: PAL-Gr 27081**—Greece, “ex insula Kastellorizo (Megiste) Dodekanesi, in collibus inter planos Barpouti et Parboutti (in parte austro-occidentali insulae), alt. 50–100 m. In regione dumuletorum sempervirentium, solo lapidoso vel rupestri calcareo,” 18 Apr 1974, *Greuter 11964* (isotypes: ATH, C, ERE, G, MA).

***Dianthus fruticosus*** subsp. ***sitiacus*** Runemark in Bot. Not. 133: 488. 1980 [17601].

**Isotype: PAL-Gr 17601**—Greece, Crete, “Ep. Sitia: in insulae Psíra parte austro-occidentali, 150 m. In praeruptis rupestribus calcareis ad septentriones spectantibus frequens,” 27 Sep 1966, *Greuter 7601* (holotype: E; other isotypes: K, LD, M, W).

***Dianthus haematocalyx*** subsp. ***phitosianus*** Constantin. in Phyton (Horn) 39(2): 279. 1999 [https://www.zobodat.at/pdf/PHY_39_2_0277-0291.pdf (accessed on 6 March 2024)].

**Second-step holotype (specified here): PAL-Gr 38797**—Greece, “Nomos Viotias, Eparchia Thivon. C. 6.6 km S of Xironomi village, along the road to Aliki. An ophiolithic area W of the road. Slopes with phrygana and low sparse shrubs, stony places. Alt. c. 320–350 m. Lat. 38°12′ N, Long. 23°02′ E,” 22 May 1998, *Constantinidis 7693* (first-step holotype: UPA [2×]; other isotypes: B, C, UPA).

***Dianthus** juniperinus*** subsp. ***heldreichii*** Greuter in Candollea 20: 187. 1965 [https://www.e-periodica.ch/cntmng?pid=can-002%3A1965%3A20%3A%3A191 (accessed on 6 March 2024)].

**Isotype** (holotype fragment)**: PAL-Gr 1848**—Greece, Crete, “in rupe Asprokremnos dicta Sphakia [in fauce Kalous Lakkous, 2000′],” [15] July 1846, *Heldreich [1640]* (holotype: G-BOIS; other isotypes: G, G-BOIS, G-BU, FI, K [2x], WU [Halácsy, Greek hb.]).

***Dianthus pulviniformis*** Greuter in Candollea 20: 189. 1965 (≡***Dianthus juniperinus*** subsp. ***pulviniformis*** (Greuter) Turland in Bull. Brit. Mus. (Nat. Hist.), Bot. 22: 166. 1992] [https://www.e-periodica.ch/cntmng?pid=can-002%3A1965%3A20%3A%3A193 (accessed on 6 March 2024)].

**Isotype: PAL-Gr 1847**—Greece, Crete: “N. Rethymnis: cliffs N of Kria Vrisi (NW of Melabes),” 1 Jun 1964, *Snogerup*, *Strid and Bothmer 20955* (holotype: LD; other isotypes: none).

***Elodes acifera*** Greuter in Candollea 20: 215. 1965 (≡***Hypericum aciferum*** (Greuter) N. K. B. Robson) [https://www.e-periodica.ch/digbib/view?pid=can-002%3A1965%3A20#219 (accessed on 6 March 2024)].

**Holotype and isotype: PAL-Gr 14669 and 14669a**—Greece, Crete: “Ep. Sfakia: unterer Teil der Schlucht Dòmata wl. Aj. Rùmeli, 20–40 m.ü.M., Plattenkalk. Hfg. In Ritzen buschiger O-exp. Schluchtwände,” 5 Jun 1962, *Greuter 4669* (holotype: LD; other isotypes: G, LD, W, Z).

***Elytraria serpens*** Greuter and R. Rankin in PhytoKeys 177: 2021. [https://phytokeys.pensoft.net/article/64764 (accessed on 6 March 2024)].

**Holotype: PAL-Gr 126731**—Cuba, “Prov. Villa Clara, Munic. Corralillo, entre las Cañas y el arroyo Clarita, alt. 85 m. 22°50′55″ N, 80°28′22″ W. Sabana en suelo mocarrero (con capa superficial de glomérulos ferralíticos),” 4 Mar 2019, *Greuter 29687 & al.* (isotypes: B, HAJB, JE, ULV).

***Erysimum candicum*** subsp. ***carpathum*** Snogerup in Opera Bot. (Lund) 13: 38. 1967 [15516].

**Isotype: PAL-Gr 15516**—Greece: “Regio aegaea austr., insula Karpathos: in faucibus Kounoupa ad septemtriones oropedii Kato Lastos, alt. 600 m. In fissuris rupium calcarearum praeruptarum,” 23 May 1963, *Greuter 5516* (holotype: LD; other isotype: Z).

***Euphorbia orphanidis*** Boiss., Diagn. Pl. Orient., ser. 2, 4: 89. 1859 [https://bibdigital.rjb.csic.es/viewer/10731/?offset=#page=91&viewer=picture&o=bookmark&n=0&q= (accessed on 6 March 2024)].

**Isolectotype: PAL-Gr 2772**—Greece: “In m. Panassi reg. media,” Aug 1855, *Guicciardi* in Heldreich, Fl. Graec. Exs. (lectotype [designated here]: G #754289; other isolectotypes: C, G #147474, K, LD, MPU, WU).

***Euphorbia rechingeri*** Greuter in Candollea 20: 170. 1965 (≡***Euphorbia myrsinites*** subsp. ***rechingeri*** (Greuter) Aldén 1986) [https://www.e-periodica.ch/cntmng?pid=can-002%3A1965%3A20%3A%3A174 (accessed on 6 March 2024)].

**Holotype: PAL-Gr 14849**—Greece, Crete: “Ep. Sfakia: Hügel zwischen den Alpen Màvro Korfàli u. Eklissìdia am N-Fuss des Berges Pàchnes, 1800–1900 m.ü.M., Kalk. Nicht selten, doch nur noch spärl. mit Früchten im Verwitterungsschutt auf schwarzem Fels; abweichend duch punktierte Kapseln, glatte Samen und Blattränder,” 3 Jul 1962, *Greuter 4849* (isotypes: G, HUJ, LD, W, Z).

***Euphorbia sultanhassei*** (‘*sultan-hassei*’) Strid & al. in Willdenowia 19: 63. 1989 [https://www.jstor.org/stable/3996919 (accessed on 6 March 2024)].

**Isotype: PAL-Gr 35829**—Greece, Crete: “Nom. Chanion, Ep. Sfakion: Middle and lower parts of the Imbros ravine, NE of Chora Sfakion, 200–400 m. Forming loose cushions up to 1 m diam. in rock crevices. Lvs turning yellow,” 12 May 1988, *Strid 26697* (holotype [to be specified]: C [2×]; other isotypes: ATH, B, E, G, LD [3×], UPA).

***Exostema lucidum*** Borhidi and M. Fernández Fernández in Acta Bot. Hung. 35: 302. 1989. (≡***Exostema lancifolium*** var. ***lucidum*** (Borhidi and M. Fernández) Borhidi) [http://real-j.mtak.hu/3649/ (accessed on 6 March 2024)].

**Isotype: PAL-Gr 126209**—Cuba, “Prov. Holguín, Mun. Moa: Cuchillas de Moa, barranco del arroyo Jaragua cerca de la Mina Jaragua, charrascales sobre roca ultrabásica, 200 m,” 3 May 1980, *Álvarez de Zayas & al.* in HFC 42659 (holotype [specified here]: HAJB #453; other isotypes: B, HAJB ##454–459 [6×], JE [2×]).

***Festuca armoricana*** Huon ex Kerguélen in Lejeunia, ser. 2, 75: 9. 1975 [9416].

**Isotype: PAL-Gr 9416**—France: “Hillion (France, Côtes-du-Nord), Pointe de Lermaot, sables maritimes fixés et pelouses sur falaises basses, avec *Thymus drucei*, *Dactyis glomerata* et *Foeniculum vulgare*. 2n = 28,” 27 May 1972, *Kerguélen* in Soc. Échange Pl. Vasc. Europe Occid. Bassin Médit. #7211 (holotype [specified here]: P #748568; other isotypes: CHAPA, G, GENT, MA [2×], MICH, MSB, P #748567, SEV).

***Festuca cyllenica*** subsp. ***pindica*** Markgr.-Dann. in Veröff. Geobot. Inst. ETH Stiftung Rübel Zürich 56: 160. 1976 (≡***Festuca pindica*** (Markgr.-Dann.) Markgr.-Dann. 1978) [https://www.e-periodica.ch/digbib/view?pid=gbi-002%3A1976%3A56#164 (accessed on 6 March 2024)].

**Isotype: PAL-Gr 30709**—Greece: “Macedonia: Distr. Grevena: Montes Pindus: In declivibus boreali-orientalibus montis Aphtia, in valle Arkudolaka (Valea Kalda) ditionis pagi Perivoli, substr. serpent., ca. 1700 m,” 30–31 Jul 1956, *Rechinger 18489* (holotype [specified here]: W #197900012707; other isotypes: B, GZU, MA, MO, RSA, S, W #197200001938).

***Festuca grandiaristata*** Markgr.-Dann. in Veröff. Geobot. Inst. Rübel Zürich 56: 101. 1976 [https://www.e-periodica.ch/digbib/view?pid=gbi-002%3A1976%3A56#105 (accessed on 6 March 2024)].

**Isotype: PAL-Gr 30710**—Greece: “Macedonia: Penins. Chalkidike. Montes Cholomonda, in quercetis, ca.500–800 m,” 5 Jun 1955; *Rechinger 17253* (holotype: [specified here]: W #197900012638; other isotype: W #197200001966).

***Festuca huonii*** Auquier in Candollea 28: 16. 1973 [https://www.e-periodica.ch/digbib/view?pid=can-002%3A1973%3A28#21 (accessed on 6 March 2024)].

**Isotype: PAL-Gr 9406**—France: “Kerloc’h, près Crozon (France, Finistère), falaises maritimes au S de la plage, avec *Dactyis glomerata, Armeria maritima, Hieracium pilosella, Scilla verna, Sanguisorba minor, Hypochoeris radicata, Anagallis parviflora, Thymus drucei, Sedum acre, Euphorbia portlandica, Daucus gummifer et Anthyllis vulneraria*, pH = 5.8,” 26 May 1972, *Kerguélen and Authier 790* in Soc. Échange Pl. Vasc. Europe Occid. Bassin Médit. #7221 (holotype: LG; other isotypes: BR, CHAPA, G, K, MA, MSB, P [3×], SEV).

***Festuca ophioliticola*** Kerguélen in Lejeunia, ser. 2, 75: 13. 1975 (≡***Festuca ovina*** subsp. ***ophioliticola*** (Kerguélen) M. Wilk. 1985) [9398].

**Isotype: PAL-Gr 9398**—France: “Peumérit (France, Finistère), entre Peumérit et Lespurit-Quélenn, sur les rochers de serpentines (ancienne carrière) et dans la lande à *Erica cinerea*, *Ulex europaeus*, *Thesium humifusum*, *Sieglingia decumbens*, *Serratula tinctoria*, *Festuca arundinacea*, *…*, pH = 6, 2n = 28,” 24 May 1972, *Kerguélen* in Soc. Échange Pl. Vasc. Europe Occid. Bassin Médit. #7229 (holotype [specified here]: P #1887862; other isotypes: AV, CHAPA, G, GENT, K, L, MA [2×], P #1887293, SEV).

***Festucopsis serpentini*** subsp. ***lurensis*** F. K. Mey. in Haussknechtia Beih. 15: 26. 2011 [52734].

**Isotype: PAL-Gr 52734**—Albania, “Lura, Kunora e Lures, ca. 1900 m,” 2 Aug 1959, *Meyer 4682* (Holotype [specified here]: JE #16665; other isotype: JE #16666).

***Gymnospermium altaicum*** subsp. ***peloponnesiacum*** Phitos in Strid and Tan, Fl. Hellenica 2: 81. 2002 (≡***Gymnospermium peloponnesiacum*** (Phitos) Strid 2009) [42787].

**Isotype: PAL-Gr 42787**—Greece, “Prov. Achaia: mons Panachaikon, in declivibus occidentalibus, 1400–1500 m; in petrosis,” 13 May 1987, *Phitos and Kamari 19922* (holotype: UPA; other isotype: C).

***Hagaea*** (‘*Hagea*’) ***polycarpoides*** Biv.-Bern., Stirp. Rar. Sicilia 2: 9. 1814 (≡***Polycarpon tetraphyllum*** subsp. ***polycarpoides*** (Biv.) Iamonico 2013) [https://www.biodiversitylibrary.org/item/106543#page/37/mode/1up (accessed on 6 March 2024)].

**Isoneotype: PAL-Gr 62429**—Italy, Sicilia: “Monte Gallo, near Palermo, 38°12′57″ N, 18°36′ E, 450 m a.s.l., on rocky limestone habitat,” 9 Jun 2014, *Domina* (neotype [see Iamonico and Domina in Pl. Biosyst. 149: 725. 2015]: PAL; other isoneotypes: FI, G, RO).

***Haloragis foliosa*** Benth, Fl. Austral. 2: 477. 1864 [https://www.biodiversitylibrary.org/page/26123257#page/491/mode/1up (accessed on 6 March 2024)].

**Isotype: PAL-Gr 39673**—Australia, [Western Australia, “between Moore and Murchison R^s^”] (from holotype label), [*Drummond VI*] *82* (holotype [specified here]: K #704050; other isotypes: MEL [2×], PERTH [fragment]).

***Harpalyce greuteri*** P. A. González and R. Rankin González in Willdenowia 51: 210. 2021.—[https://doi.org/10.3372/wi.51.51204 (accessed on 6 March 2024)].

**Isotype: PAL-Gr 128204**—Cuba, “Prov. Holguín, municipio Rafael Freyre, al este de la Bahía de Naranjo, camino al restaurante ‘conuco de Mongo Viña’, matorral xeromorfo espinoso sobre serpentina. 10 m s.m., 21°06′00″ N, 75°52′14.6″ W,” 30 May 2019, *González Gutiérrez 1905-1* (holotype: HAJB #1216; other isotype: B).

***Harpalyce marianensis*** R.Rankin, P.A.González and Greuter in PhytoKeys 225: 83–97. [https://phytokeys.pensoft.net/article/99321/ (accessed on 6 March 2024)].

**Isotype: PAL-Gr 133251**—Cuba, “Prov. Guantánamo, San Antonio. Abra de Mariana, en el barranco 5 km al NO de San Antonio del Sur,” 11 May 1980, *Álvarez & al.* in HFC 43077 (holotype: JE #28983; other isotypes: B, HAJB, JE #28984).

***Hedysarum boveanum*** subsp. ***palentinum*** Valdés in Lagascalia 21: 253. 1999 [https://institucional.us.es/revistas/lagascalia/21.1/floristic.pdf (accessed on 6 March 2024)].

**I*s*otype: PAL-Gr 55254**—Spain, “Palencia: Cevico Navero, 30TVM3602, 850 m, en matorrales sobre Margas y suelos de costra yesífera, en comunidades de Aphyllanthion,” 11 Jun 1982, *Fdez. Díez* (holotype: SEV #93257; other isotypes: BC, G, MAF, SALA).

***Helianthemum fasciculi*** Greuter in Boissiera 13: 55. 1967, nom. illeg. [=***Helianthemum syriacum*** (Jacq.) Dum. Cours.] [https://www.e-periodica.ch/digbib/view?pid=boi-001%3A1967%3A13#59 (accessed on 6 March 2024)].

**Second-step holotype (specified here): PAL-Gr 16559**—Greece, “Aegäis, Nom. Attikì, Ep. Kìthira: bei der Kirche Odijìtria söl. Livàdi, 170 m.ü.M., sandig verwitterndes Neogen. Nicht selten in der *Cistus*phrygana,” 12 May 1964, *Greuter 6559* (isotypes: G, W).

***Hieracium aequimontis*** Gottschl. and Meierott in Ber. Bayer. Bot. Ges. 77: 141. 2007 (≡***Pilosella aequimontis*** (Gottschl. and Meierott) S. Bräut. and Greuter, **comb. nov.**) [https://www.bbgev.de/_files/ugd/f05de5_841140e9617d42f7a93994e46fde2d68.pdf (accessed on 6 March 2024)].

**Isotype: PAL-Gr 45143**—Germany, “Thüringen, Landkreis Hildburghausen, Gleichamberg, Großer Gleichberg, 650 m s.m., ehemaliger Steinbruch, Schutthalden, MTB 5629/21, 50°22′ N, 10°35′ E,” 30 May 1998, *Gottschlich 35725 and Meierott* in *Hieracia* Europ. Sel. #112 (holotype: M; isotypes: G, FI, GOET).

***Hieracium armerioides*** subsp. ***paucitrichum*** Gottschl. in Stapfia 107: 20. 2017 [https://www.zobodat.at/pdf/STAPFIA_0107_0011-0028.pdf (accessed on 6 March 2024)].

**Isotype: PAL-Gr 125290**—Italy, “Friaul-Julisch Venetien, Prov. Udine, Comeglians, M. Crostis, (), 1883 m s.m., Feinschutt (Kalk?), MTB 9443/2, 46°33′21″ N 12°55′32″ E,” 22 Jul 2016, *Gottschlich-65761* in *Hieracia* Europ. Sel. #706 (holotype: FI; isotypes: FR, H, IBF, LI, M, PRA, W, Hb. Brandstätt., Hb. Dunkel, Hb. Gottschl.).

***Hieracium atrocalyx*** Gottschl. in Stapfia 95: 33. 2011 [https://www.zobodat.at/pdf/STAPFIA_0095_0033-0045.pdf (accessed on 6 March 2024)].

**Isotype: PAL-Gr 52625**—Italy, “Friaul-Julisch Venetien, Prov. Pordenone, Erto e Casso, untere Kehren im Aufstieg zum Val Zemola, 1000 m s.m., Kalkschutt und Gebüsche, MTB 9740/1, 46°16′ N, 12°23′ E,” 8 Aug 2009, *Gottschlich-55066* in *Hieracia* Europ. Sel. #362 (holotype: FI; Isotypes: B, GOET, H, IBF, LI, M, PRC, W, hb. Brandstätt., hb. Dunkel, hb. Gottschl.).

***Hieracium barrelieri*** Gottschl., Raimondo, Greuter, and Di Grist. in Phytotaxa 208(1): 71. 2015 [https://www.dropbox.com/preview/reprints/Hieracium%20barrelieri.pt00208p074.pdf (accessed on 6 March 2024)].

**Isotype: PAL-Gr 62680**—Italy, “Campania, Montevergine, Mercogliano (AV). Coordinate: 40°56′34.52″ N, 14°43′35.63″ E. Alt.: 1290 m. Esposiz.: NO. Habitat: carbonate rocks,” 07 Jul 2014, *Di Gristina [101733]* (holotype: PAL!; other isotypes: B!,hb. Gottschl.).

***Hieracium bifidum*** subsp. ***nummulariifolium*** Gottschl. in Stapfia 89: 108. 2009 [https://www.zobodat.at/pdf/STAPFIA_0089_0001-0328.pdf (accessed on 6 March 2024)].

**Isotype: PAL-Gr 45237**—Italy, “Abruzzen, Prov. L’Áquila, Gran Sasso, Calascio: S “Piano Buto,” 1050 m, Kalkfelsböschung, MTB 3648/3, 42°18′ N, 13°40′ E,” 3 Jun 2004, *Gottschlich 48828* in *Hieracia* Europ. Sel. #208 (holotype [to be specified]: FI; Isotypes: B, FI, GOET, H, IBF, LI, M, PR, W, hb. Brandstätt., hb. Dunkel, hb. Gottschl.).

***Hieracium bifidum*** subsp. ***subhastatum*** Gottschl. in Stapfia 89: 107. 2009 [https://www.zobodat.at/pdf/STAPFIA_0089_0001-0328.pdf (accessed on 6 March 2024)].

**Isotype: PAL-Gr 45236**—Italy, “Abruzzen, Prov. Téramo, Gran Sasso, Sent. 2: zwischen Rif. Duca degli Abruzzi und M. Portella, 2300–2320 m, nordexponierte Kalkschutthänge, MTB 3547/3, 42°26′ N, 13°32′ E,” 31 Jul 2001, *Gottschlich 44649 and Dunkel* in *Hieracia* Europ. Sel. #207 (holotype [to be specified]: FI; isotypes: B, FI, GOET, IBF, H, LI, M, PR, W, hb. Brandstätt., hb. Dunkel, hb. Gottschl.).

***Hieracium bohatschianum*** subsp. ***onosmoidiforme*** Gottschl. and Melikoki in Willdenowia 43: 62. 2013 [https://www.jstor.org/stable/42751684?seq=4 (accessed on 6 March 2024)].

**Isotype: PAL-Gr 58829**—Greece, “Prefecture of Chalcidice, C. Chalcidice, 40°30′13,6″ N, 23°33′13,5″ E, 547 m, At the slope of road. Rocky position near at heathland,” 6 Jun 2011, *Melikoki 6.13.3* (holotype: TAUF; other isotypes: ATHU, B, C, FI, LD, UPA, W, hb. Gottschl.).

***Hieracium boreoapenninum*** Gottschl. in Webbia 64: 3. 2009 [48439].

**Isotype: PAL-Gr 48439**—Italy, “Ligurien, Prov. Génova, Montebruno: Passo della Scoglina, 920–930 m s.m., Kammweg. MTB 1521/1, 44°28′ N, 9°14′ E, *Gottschlich 52362* in *Hieracia* Europ. Sel. #275 (holotype: FI; other isotypes: B, GOET, IBF, H, LI, M, PR, W, hb. Brandstätt., hb. Dunkel, hb. Gottschl.).

***Hieracium bupleuroides*** subsp. ***tririvicola*** Gottschl. in Stapfia 105: 85. 2016 [https://www.zobodat.at/pdf/STAPFIA_105_0064-0091.pdf (accessed on 6 March 2024)].

**Isotype: PAL-Gr 121155**—Italy, “Toskana, Prov. Lucca, Alpi Apuane, Arni, aufgelassener Marmor-Steinbruch bei “Tre Fiumi,” 761 m, Marmorfeinschutt mit Weidenbüschen, MTB 1927/2, 44°03′18″ N, 10°16′21″ E,” 1 Jul 2015, *Gottschlich 64325* in *Hieracia* Europ. Sel. #611 (holotype: FI; other isotypes: B, FR, H, IBF, LI, M, PRA, W, hb. Brandstätt., hb. Dunkel, hb. Gottschl.).

***Hieracium caesiolympicum*** Gottschl. and Dunkel in Stapfia 111: 9. 2019 [https://www.zobodat.at/pdf/STAPFIA_0111_0005-0032.pdf (accessed on 6 March 2024)].

**Isotype: PAL-Gr 129006**—Greece, “Ostmakedonien, Nom. Xanthi, Lydodromio → Ag. Georgios, 550 m s.m., Gebüsch, 41°14′22″ N, 24°45′51″ E,” 24 Jun 2019, *Gottschlich 72651 and Dunkel* in *Hieracia* Europ. Sel. #801 (holotype: B; other isotypes: FI, FR, H, IBF, LI, M, MSTR, UPA, W, Hb. Brandstätt., Hb. Dunkel, Hb. Gottschl.).

***Hieracium calodon*** subsp. ***rhenovulcanicum*** Gottschl. and Heinrichs in Decheniana 154: 8. 2001 (≡***Pilosella rhenovulcanica*** (Gottschl. and Heinrichs) Bomble 2020) [45056].

**Isotype: PAL-Gr 45056**—Germany, “Rheinland-Pfalz, Landkreis Mayen-Koblenz, Plaidt, Autobahn-Abfahrt Plaidt, 146 m s.m., Straßenrand, MTB 5610/12, 50°23′ N, 7°24′ E,” 25 May 1995, *Gottschlich 27383*, *Heinrichs and Raabe* in *Hieracia* Europ. Sel. #25 (holotype: B #100024153; other isotypes: BONN, GOET, HBG, M).

***Hieracium cavallense*** Gottschl. in Stapfia 89: 93. 2009 [https://www.zobodat.at/pdf/STAPFIA_0089_0001-0328.pdf (accessed on 6 March 2024)].

**Isotype:PAL-Gr 48433**—Italy, “Abruzzen, Prov. Chieti, Majella: Sent. 1: Blockhaus → Monte Cavallo → Bivacco Fusco, 2045–2100 m s.m., *Pinus mugo*-Gebüsche. MTB 3850/4, 42°07′ N, 14°06′ E” 31 Jul 2000, *Gottschlich-41661 and Dunkel* in *Hieracia* Europ. Sel. #269 (holotype [to be specified]: FI; other isotypes: B, FI, GOET, IBF, H, LI, M, PR, W, hb. Brandstätt., hb. Dunkel, hb. Gottschl.).

***Hieracium cepitinum*** Gottschl. in Stapfia 89: 63. 2009 [≡***Pilosella cepitina*** (Gottschl.) Gottschl. 2010; =***Pilosella arnoserioides*** (Nägeli and Peter) Soják in Preslia 43: 186. 1971] [https://www.zobodat.at/pdf/STAPFIA_0089_0001-0328.pdf (accessed on 6 March 2024)].

**Isotype: PAL-Gr 45190**—Italy, “Abruzzen, Prov. Téramo: Gran Sasso, Pietracamela, Straße Richtung Prati di Tivo, Pietracamela: E C.^le^ Cepito, 1220 m s.m., Kalkmagerrasen, MTB 3447/3, 42°31′ N, 13°33′ E,” 6 Jun 2001, *Gottschlich 44066 and Dunkel* in *Hieracia* Europ. Sel. #159 (holotype [to be specified]: FI; other isotypes: B, FI, GOET, H, IBF, LI, M, PR, W, hb.Brandstätt., hb.Dunkel, hb.Gottschl.).

***Hieracium cerinthoides*** L., Sp. Pl.: 803. 1753 [https://www.biodiversitylibrary.org/item/84236#page/249/mode/1up (accessed on 6 March 2024)].

**Isoepitype: PAL-Gr 66254**—Spain, “Lleida: Tredòs (Vall d’Aran), cap al Lac de Baish de Bacivèr, 31TCH3329, datum ED50, 1920 m, roques calcàries,” 24 Jul 2012, *Mateo*, *Rosselló*, *Sàez and del Egido* (epitype [Ferrer-Gallego & al. in Willdenowia 45: 388. 2015: https://www.jstor.org/stable/24753246?seq=3]: VAL #210645; other isoepitypes: B, LEB, VAL #227665 [interpreting the lectotype: Linn. No. 954.28 (LINN) [http://linnean-online.org/8846]).

***Hieracium chloropsis*** subsp. ***apuanorum*** Gottschl. in Stapfia 105: 85. 2016 [https://www.zobodat.at/pdf/STAPFIA_105_0064-0091.pdf (accessed on 6 March 2024)].

**Isotype: PAL-Gr 121152**—Italy, “Toskana, Prov. Lucca, Alpi Apuane, Umgebung Rif. Val Serenaia, 1040–1060 m s.m., Schuttflächen, Abbruchkanten, Bachbett, MTB 1827/3, 44°08′09″ N, 10°11′49″ E,” 17 Jun 2014, *Gottschlich 62663* in *Hieracia* Europ. Sel. #608 (holotype: FI; other isotypes: B, FR, H, IBF, LI, M, PRA, TO, W, hb. Brandstätt., hb. Dunkel, hb. Gottschl.).

***Hieracium chlorifolium*** subsp. ***rendinaricum*** Gottschl. in Stapfia 89: 87. 2009 [https://www.zobodat.at/pdf/STAPFIA_0089_0001-0328.pdf (accessed on 6 March 2024)].

**Isotype: PAL-Gr 45215**—Italy, “Prov. L’Áquila, Val Roveto, Rendinara: NNE-exponiertes Kar des M. Pratillo, 1900–1970 m s.m., Kalkschutt, Felsrasen. MTB 4146/4, 41°48′ N 13°29′ E,” 30 Jul 2004, *Gottschlich 48977 and Dunkel* in *Hieracia* Europ. Sel. #186 (holotype [to be specified]: FI; isotypes: (B, FI, GOET, IBF, H, LI, M, PR, W, hb. Brandstätter, hb. Dunkel, hb. Gottschl.).

***Hieracium contii*** Gottschl. in Stapfia 89: 120. 2009 [https://www.zobodat.at/pdf/STAPFIA_0089_0001-0328.pdf (accessed on 6 March 2024)].

**Isotype: PAL-Gr 45241**—Italy, “Prov. L’Áquila, Gran Sasso,: Barisciano: N-Hang des unbenannten Berges “1516 m” zwischen M. della Selva und M. Capanello, 1350 m s.m., lichter Kiefernforst, MTB 3647/2, 42°21 N, 13°35′ E,” 9 Jun 2003, *Gottschlich 47509*, *Dunkel*, *Conti*, *Tinti* and *Londrillo*” in Hieracia Europ. Sel. #212 (holotype [to be specified]: FI; isotypes: B, FI, GOET, IBF, H, LI, M, PR, W, hb. Brandstätter, hb. Dunkel, hb. Gottschl.).

***Hieracium dasycraspedum*** Buttler in Strid, Mount. Fl. Greece 2: 630. 1991 [20638].

**Holotype: PAL-Gr 20638**—Greece, “Epirus, prov. Ioannina, distr.Dhodhoni: montes Timfi, in latere boreali montis Astraka supra refugium, alt. 2200 m. In scansilibus graminosis rupium calcarearum,” 21 Aug 1974, *Charpin 11358*, *Dittrich*, *Greuter 12530* and *von Auw* (isotypes: G, LD, M, UPA, hb. Dittrich).

***Hieracium dentatum*** subsp. ***montiscatriae*** Gottschl. in Webbia 66: 216. 2011 [52608].

**Isotype: PAL-Gr 52608**—Italy, “Marche, Prov. Pésaro-Urbino, M. Cátria, NW-Hang, S Rif. della Vernosa, 1500–1600 m s.m., offene Kalkfelsrasen, MTB 2542/1, 43°27′56″ N, 42°14′ E,” 1 Aug 2000, *Gottschlich 55854* and *Dunkel* in *Hieracia* Europ. Sel. #355 (holotype: FI; other isotypes: B, GOET, IBF, H, LI, M, PR, W, hb. Brandstätt., hb. Dunkel, hb. Gottschl.).

***Hieracium dentatum*** subsp. ***setuligerum*** Gottschl. in Webbia 66: 218. 2011 [52606].

**Isotype: PAL-Gr 52606**—Italy, “Marche, Prov. Macerata, Monti Sibillini, Ussita, Frontignano, S-Hang M. Bicco oberhalb Sent. 158, 1820–1880 m s.m., Kalkschutt. MTB 3045/3, 42°54′52″ N, 13°11′07,”E” 12 Jul 2009, *Gottschlich 54828* in *Hieracia* Europ. Sel. #353 Holotyppe: FI; other isotypes: B, GOET, IBF, H, LI, M, PR, W, hb. Brandstätt., hb. Dunkel, hb. Gottschl.).

***Hieracium diaphanoides*** subsp. ***chaucorum*** Gottschl. and Schabelr. in Kochia 11: 37. 2018 [https://ojs.ub.uni-frankfurt.de/kochia/index.php/kochia/article/view/50/221 (accessed on 6 March 2024)].

**Isotype: PAL-Gr 52606**—Germany, “Niedersachsen, Landkreis Friesland, Urwaldstraße westlich Bockhorn, 5 m s.m., Straßenböschung, MTB 2614/11 (53°23′47′′ N, 8°0′15′′ E), 10 Jun 2015, *Schabelreiter 1444* in *Hieracia* Europ. Sel. #702 (holotype: B; other Isotypes: FI, FR, H, IBF, LI, M, PRA, W -1444, hb. Dunkel, hb. Gottschl.-64487, hb. Schabelreit.).

***Hieracium diaphanoides*** subsp. ***dimonense*** Gottschl. in Stapfia 107: 24. 2017 [https://www.zobodat.at/pdf/STAPFIA_0107_0011-0028.pdf (accessed on 6 March 2024)].

**Isotype: PAL-Gr 125292**—Italy, “Friaul-Julisch Venetien, Prov. Udine, Paularo, Ligusullo → Rif. Valdajer, 1179 m s.m., oberhalb Straßenmauer unter Fichtenwald, MTB 9444/3, 46°32′34″ N, 13°04′50″ E,” 26 Jul 2013, *Gottschlich 60728* in *Hieracia* Europ. Sel. #708 (holotype: FI; other isotypes: B, FR, H, IBF, LI, M, PRA, W, hb. Brandstätt., hb. Dunkel, hb. Gottschl.).

***Hieracium diaphanoides*** subsp. ***forojuliense*** Gottschl. in Stapfia 107: 20. 2017 [https://www.zobodat.at/pdf/STAPFIA_0107_0011-0028.pdf (accessed on 6 March 2024)].

**Isotype: PAL-Gr 125291**—Italy, “Friaul-Julisch Venetien, Prov. Udine, Cividale, Riekatal: E Clodig, 319 m s.m., Straßenböschung. MTB 9847/2, 46°9′25″ N, 13°36′4″ E,” 30 May 2012, Gottschlich 58379 in *Hieracia* Europ. Sel. #707 (holotype: FI; other isotypes: B, FR, H, IBF, LI, M, PRA, W, hb. Brandstätt., hb. Dunkel, hb. Gottschl.).

***Hieracium diaphanoides*** subsp. ***forstense*** Gottschl. in Mitt. Pollichia 98: 76. 2017 [file:///C:/Users/wg/Dropbox/PC/Downloads/Mitt-Pollichia_98_0075-0078(4).pdf.

**Isotype: PAL-Gr 123321**—Germany, “Rheinland-Pfalz, Landkreis Bad Dürkheim/Weinstraße, Forst: “Grüne Bank” westlich oberhalb Forst, 390 m s.m., Eichen-Kiefern-Wald mit Kastanien. MTB 6514/42, 49°25′33″ N, 8°9′44″ E,” 28 May 2015 *Gottschlich 63987* in *Hieracia* Europ. Sel. #651 (holotype: B; other isotypes: FI, FR, H, IBF, LI, M, POLL, PRA, W, hb. Brandstätt., hb. Dunkel, hb. Gottschl.).

***Hieracium** diaphanoides*** subsp. ***volakasense*** Gottschl. and Dunkel in Stapfia 111: 18. 2019 [file:///C:/Users/wg/Dropbox/PC/Downloads/STAPFIA_0111_0005-0032.pdf.

**Isotype: PAL-Gr 129010**—Greece, “Ostmazedonien, Nom. Drama, M Falakro: Straße zum Skicenter Falakro, 1050 m s. m., Kalkfels unter Buchenwald. 41°17′46″ N, 24°0′11″ E” 21 Jun 2019, *Gottschlich 72493* and *Dunkel* in *Hieracia* Europ. Sel. #805 (holotype: B; isotypes: FI, FR, H, IBF, LI, M, MSTR, UPA, W, hb. Brandstätt., hb. Dunkel, hb. Gottschl.).

***Hieracium dunkelii*** Gottschl. in Biol. Beitr. 33: 585. 2001 [https://www.zobodat.at/pdf/LBB_0033_1_0583-0594.pdf (accessed on 6 March 2024)].

**Isotype: PAL-Gr 45180**—Italy, “Südtirol, Prov. Bozen, Schnalstal: Straße entlang des Vernagt-Stausees, 200 m E Finailbach, 1750 m s.m., felsiger Steilhang mit Lärchen. MTB: 9230/4, 46°44′ N, 10°49′ E,” 22 Jul 2000, *Gottschlich 41544* in *Hieracia* Europ. Sel. #149 (holotype: LI, other isotypes: FI, G, IBF, W, hb. Gottschl.).

***Hieracium duronense*** Gottschl. in Linzer Biol. Beitr. 38: 1049. 2006 [https://www.zobodat.at/pdf/LBB_0038_2_1045-1059.pdf (accessed on 6 March 2024)].

**Isotype: PAL-Gr 45124**—Italy, “Südtirol, Prov. Trento: Campitello di Fassa, Valle di Duron, E Rif. Micheluzzi, 1840–1880 m s.m., Schneise im Bergfichten-Wald auf Basalt. MTB 9536/1, 46°29′ N, 11°42′ E,” 2 Jul 2005, *Gottschlich 44748*, *Dunkel*, *Prosser*, *Pujatti,* and *Zidorn* in *Hieracia* Europ. Sel. #93 (holotype: FI; other isotypes: ROV, hb. Gottschl.).

***Hieracium balearicum*** Arv.-Touv., Hier. Gall. Hisp. Cat.: 157. 1913) [8536] (≡***Hieracium elisaeanum*** subsp. ***balearicum*** (Arv.-Touv.) Greuter 2007).

**Isotype: PAL-Gr 8536**—Spain, Balearic Islands: “Balearium insula Majore in rupestribus m^is^ Puig mayor de Torella fal. calcar., 1000–1300 m s.m.,” 8 Jun 1885, *Porta* & *Rigo* in Iter Hisp. ‚608 (holotype [specified here]: GRM; other isotypes: many).

***Hieracium erucopsis*** Gottschl. in Stapfia 105: 69. 2016 [https://www.zobodat.at/pdf/STAPFIA_105_0064-0091.pdf (accessed on 6 March 2024)].

**Isotype: PAL-Gr 121156**—Italy, “Toskana, Prov. Lucca, Alpi Apuane, Val Serenaia, Umgebung Rifugio Val Serenaia, 1040–1060 m s,m., Schuttflächen, Abbruchkanten, Bachbett. MTB 1827/3, 44°8′9″ N, 10°11′49″ E” 17 Jun 2014, *Gottschlich 62662* in *Hieracia* Europ. Sel. #612 (holotype: FI; other isotypes: B, FR, H, IBF, LI, M, PRA, W, hb. Brandstätt., hb. Dunkel, hb. Gottschl.).

***Hieracium exilicaule*** Gottschl. in Stapfia 89: 119. 2009 [https://www.zobodat.at/pdf/STAPFIA_0089_0001-0328.pdf (accessed on 6 March 2024)].

**Isotype: PAL-Gr 45264**—Italy, “Abruzzen, Prov. L’Áquila, Gran Sasso, Calascio: S “Piano Buto,” 1050 m s.m., Kalkfelsböschung. MTB 3648/3, 42°18′ N, 13°40′ E” 3 Jun 2004, *Gottschlich 48826* in *Hieracia* Europ. Sel. #235 (holotype [to be specified]: FI; other isotypes: B, FI, GOET, IBF, H, LI, M, PR, W, hb. Brandstätt., hb. Dunkel, hb. Gottschl.).

***Hieracium faucis-jovis*** (‘*faucisjovis*’) Gottschl. in Webbia 64: 182. 2009 [48447].

**Isotype: PAL-Gr 48447**—Italy, “Toskana, Lucca, Bagni di Lucca: S unterhalb Foce a Giovo, 1650 m s.m., Felsheiden, Wiesen, Sandstein. MTB 1829/4, 44°7′ N, 10°36′ E,” 1 Jul 2005, *Gottschlich 50227* in *Hieracia* Europ. Sel. #283 (holotype: FI; other isotypes: B, GOET, IBF, H, LI, M, PRA, W, hb. Brandstätt., hb. Dunkel, hb. Gottschl.).

***Hieracium froelichianum*** subsp. ***cavicola*** Gottschl. in Stapfia 95: 36. 2011 [https://www.zobodat.at/pdf/STAPFIA_0095_0033-0045.pdf (accessed on 6 March 2024)].

**Isotype: PAL-Gr 52611**—Italy, “Piemont, Prov. Torino, Valle di Susa, zwischen Salbertrand und Pramand, S Galleria dei Sarazeni, 2170 m s.m., feuchte Balmen. MTB 0609/2, 45°4′33″ N, 6°49′33″ E,” 10 Aug 2010, *Gottschlich 56082* in *Hieracia* Europ. Sel. #358 (holotype: FI; other isotypes: B, GOET, H, IBF, LI, M, PRC, W, hb. Brandstätt., hb. Dunkel, hb. Gottschl.).

***Hieracium galeroides*** Gottschl. in Stapfia 89: 116. 2009 [https://www.zobodat.at/pdf/STAPFIA_0089_0001-0328.pdf (accessed on 6 March 2024)].

**Isotype: PAL-Gr 45246**—Italy, “Abruzzen, Prov. L’Áquila, Roccaraso: an der Straße zum Wintersportgebiet “Aremogna,” oberhalb der großen Hotels, Kalkschutthang, 1530 m s.m., MTB 4150/1, 41°50′ N, 14°3′ E,” 1 Jun 2001, *Gottschlich 43900* in *Hieracia* Europ. Sel. #217 (holotype: FI [to be specified]; other isotypes: B, FI, GOET, IBF, H, LI, M, PR, W, hb. Brandstätt., hb. Dunkel, hb. Gottschl.).

***Hieracium glaucinum*** subsp. ***pseudobasalticum*** Gottschl. in Stapfia 89: 98. 2009 [https://www.zobodat.at/pdf/STAPFIA_0089_0001-0328.pdf (accessed on 6 March 2024)].

**Isotype: PAL-Gr 45230**—Italy, “Abruzzen, Prov. Téramo, Montagna dei Fiori: zwischen San Giacomo und Ripe: SW Abzweigung Richtung Cerqueto, 890 m s.m., Kiefernforst. MTB 3247/2, 42°46′ N, 13°37′ E,” 30 May 2004, *Gottschlich 48696* in *Hieracia* Europ. Sel. #201 (holotype: FI [to be specified]; other isotypes: B, FI, GOET, IBF, H, LI, M, PR, W, hb. Brandstätt., hb. Dunkel, hb. Gottschl.).

***Hieracium glaucinum*** subsp.***valpergae*** Gottschl. in Rivista Piemont. Storia Nat. 36: 52. 2015 [https://www.storianaturale.org/anp/PDF%20ANP/36_2015_Gottschlich_Hieracium%20glaucinum%20valpergae%20nuovo%20endemita%20puntiforme%20del%20Sacro%20Monte%20di%20Belmonte_51-56.pdf (accessed on 6 March 2024)].

**Isotype: PAL-Gr 121148**—Italy, “Piemont, Prov. Torino, Canavese, Valperga: Auffahrt Sacro Monte di Belmonte 626 m s.m., Granitfels. MTB 0611/2, 45°21′57″ N, 7°37′24″ E,” 25 May 2013, *Gottschlich 59937* in *Hieracia* Europ. Sel. #604 (holotype: FI; other isotypes: B, FR, H, IBF, LI, M, PR, TO, W, hb. Brandstätt., hb. Dunkel, hb. Gottschl.).

***Hieracium glaucum*** subsp. ***serenaiae*** Gottschl. in Stapfia 105: 82. 2016 [https://www.zobodat.at/pdf/STAPFIA_105_0064-0091.pdf (accessed on 6 March 2024)].

**Isotype: PAL-Gr 121154**—Italy, “Toskana, Prov. Lucca, Alpi Apuane, Val Serenaia, sent. 179 unterhalb “Cresta Garnerone,” 1475 m s.m., Marmorfelsrasen. MTB 1827/3, 44°7′30″ N, 10°11′33″ E,” 11 Aug 2014, Gottschlich 63270, Soldano and Soldano in *Hieracia* Europ. Sel. #610 (holotype: FI; other isotypes: B, FR, H, IBF, LI, M, PRA, W, hb. Brandstätt., hb. Dunkel, hb. Gottschl.).

***Hieracium grandipetalum*** Sennen in sched. Pl. Espagne #4768. 1923; et in Bol. Soc. Ibér. Ci. Nat. 28: 119. 1929 [48589 and https://bibdigital.rjb.csic.es/viewer/10467/?offset=#page=120&viewer=picture&o=bookmark&n=0&q= (accessed on 6 March 2024)].

**Isotype: PAL-Gr 48589**—France [Pyrénées-orientales], “Cerdagne: Gorges de Llo [Sègre], éboulis herbeux, 1520 m. 30 Jul 1923, *Sennen* in Pl. Espagne #4768. Holotype [specified here]: BC; other isotypes: BM, MPU, PH [2×], RNG).

***Hieracium grossicephalum*** Gottschl., Brandst. and Dunkel in Linzer Biol. Beitr. 38: 1052. 2006 [https://www.zobodat.at/pdf/LBB_0038_2_1045-1059.pdf (accessed on 6 March 2024)].

**Isotype: PAL-Gr 45125**—Austria, “Kärnten, Ankogelgruppe: Kölnbreinspeicher, Nordufer des Stausees, Wanderweg 02, zwischen Abzweigung ins Kleinelendtal und Osnabrückerhütte, ca. 500 m SSW P. 1916 (sphalm. “2916“), 1910–1920 m s.m., Grünerlengebüsch mit *Calluna* und Hochstaudenfragmenten, Silikat. MTB 8945/2, 47° 4′ N, 13°18′ E,” 13 Aug 2005, *Dunkel 13747* in *Hieracia* Europ. Sel. #94 (holotype: W, other isotypes: KL, hb. Dunkel, hb. Gottschl.).

***Hieracium grovesianum*** subsp. ***luteobarbatum*** Gottschl. in Stapfia 89: 155. 2009 [https://www.zobodat.at/pdf/STAPFIA_0089_0001-0328.pdf (accessed on 6 March 2024)].

**Isotype: PAL-Gr 45272**—Italy, “Abruzzen, Prov. Téramo, Gran Sasso: Vado di Sole → Castelli: ESE Mte. Vito, 1350 m s.m., Buchenwaldrand. MTB 3548/4, 42°24′ N, 13°47′ E,” 30 Jul 2002, *Gottschlich 46235* in *Hieracia* Europ. Sel. #243 (holotype [to be specified]: FI; other isotypes: B, FI, GOET, IBF, H, LI, M, PR, W, hb. Brandstätt., hb. Dunkel, hb. Gottschl.).

***Hieracium grovesianum*** subsp. ***nigrotectorium*** Gottschl. in Stapfia 89: 154. 2009 [https://www.zobodat.at/pdf/STAPFIA_0089_0001-0328.pdf (accessed on 6 March 2024)].

**Isotype: PAL-Gr 45269**—Italy, “Abruzzen, Prov. L’Áquila, Carsóli: oberhalb Tufo, 900 m s.m., Buchenwald mit *Castanea sativa*. MTB 3844/4, 42°9′ N, 13°6′ E,” 29 May 2004, *Gottschlich 48660* in *Hieracia* Europ. Sel. #240 (holotype [to be specified]: FI; other isotypes: B, FI, GOET, IBF, H, LI, M, PR, W, hb. Brandstätt., hb. Dunkel, hb. Gottschl.).

***Hieracium hypochoeroides*** subsp. ***paikoanum*** Gottschl. and Dunkel in Stapfia 111: 22. 2019 [https://www.zobodat.at/pdf/STAPFIA_0111_0005-0032.pdf (accessed on 6 March 2024)].

**Isotype: PAL-Gr 129011**—Greece, “Nom. Kilkis, M. Paiko, Straße S Livadia, beim Aussichtspunkt, 1110 m s.m., felsige Straßenböschung, Kalk. 40°59′22″ N, 22°20′48″ E,” 19 Jun 2019, *Gottschlich 72391 and Dunkel* in *Hieracia* Europ. Sel. #806 (holotype: B; other isotypes: FI, FR, H, IBF, LI, M, MSTR, UPA, W, hb. Brandstätt., hb. Dunkel, hb. Gottschl.).

***Hieracium juengeri*** Gottschl. in Stapfia 105: 65. 2016 [https://www.zobodat.at/pdf/STAPFIA_105_0064-0091.pdf (accessed on 6 March 2024)].

**Isotype: PAL-Gr 121154**—Italy, “Toskana, Prov. Massa-Carrara, Alpi Apuane, sent. 172: Foce di Pianza → M. Sagro, 1300–1500 m s.m., Marmorfelsspalten und Felsrasen. MTB 1826/4, 44°6′ N, 10°9′ E,” 2 Jul 2005, *Gottschlich 50251* and *Marchetti* in *Hieracia* Europ. Sel. #605 (holotype: FI; other isotypes: B, FR, H, IBF, LI, M, PRA, W, hb. Brandstätt., hb. Dunkel, hb. Gottschl.).

***Hieracium kofelicum*** Gottschl. in Linzer Biol. Beitr. 32: 368. 2000 [https://www.zobodat.at/pdf/LBB_0032_1_0363-0398.pdf (accessed on 6 March 2024)].

**Isotype: PAL-Gr 45068**—Italy, “Südtirol, Prov. Bozen: Vinschgau, Straße Kastellbell Richtung St. Martim im Kofel, Kehren unterhalb Koben, 1500 m s.m., lichter Kiefernwald, Orthogneis/Glimmerschiefer. MTB: 9331/3, 46°38′ N, 10°51′ E,” 19 Jun 1999, *Gottschlich 38147 and Dunkel* in *Hieracia* Europ. Sel. #37 (holotype: LI; other isotypes: IBF, hb. Dunkel, hb. Gottschl., hb. Naturkundemus. Bozen).

***Hieracium laevigatum*** subsp. ***trichocalyx*** Gottschl. in Rivista Piemont. Storia Nat. 37: 20. 2016 [https://www.storianaturale.org/anp/pdf2/ANP_37_2016/13_ANP_37_Gottschlich.pdf (accessed on 6 March 2024)].

**Isotype: PAL-Gr 121144**—Italy, “Piemont, Prov. Biella, Rosazza, T. Pragnetta:. Parkplatz/Einstieg sent. E 30, Parkplatzrand, Alluvionen, 949 m s.m. MTB 0313/2, 45°40′26″ N, 7°58′22″ E,” 6 Aug 2014, *Gottschlich-63201* in *Hieracia* Europ. Sel. #601 (holotype: FI: other isotypes: B, FR, H, IBF, LI, M, PRA, W, hb. Brandstätt., hb. Dunkel, hb. Gottschl.).

***Hieracium lanudae*** Gottschl. in Webbia 64: 179. 2009 [48446].

**Isotype: PAL-Gr 48446**—Italy, “Emilia-Romagna, Prov. Reggio, Passo del Cerreto, M. La Nuda: Bergstation Sessellift bis Ende Schlepplift, 1500–1700 m ss.m., Silikat-Felsschutt. MTB 1727/1, 44°16′ N, 10°14′ E,” 3 Jul 2005. *Gottschlich 50262* in *Hieracia* Europ. Sel. #282 (holotype: FI; other isotypes: B, GOET, IBF, H, LI, M, PR, W, hb. Brandstätt., hb. Dunkel, hb. Gottschl.).

***Hieracium lawsonii*** subsp. ***elvae*** in Stapfia 101: 28. 2014 [https://www.zobodat.at/pdf/STAPFIA_0101_0027-0037.pdf (accessed on 6 March 2024)].

**Isotype: PAL-Gr 63660**—Italy, “Piemont, Prov. Cuneo, Valle Máira, Vallone di Elva, 1213–1250 m s.m., Kalkfels an der Straße. MTB 1408/4, 44°31′3″ N, 7°5′24″ E,” 8 Jul 2013, Gottschlich-60566 and Dunkel in *Hieracia* Europ. Sel. #551 (holotype: FI; other isotypes: B, FR, H, IBF, LI, M, PR, W, hb. Brandstätt., hb. Dunkel, hb. Gottschl.).

***Hieracium marchesonii*** Gottschl. in Webbia 66: 204. 2011 [52605].

**Isotype: PAL-Gr 52605**—Italy, “Marche, Prov. Macerata, Monti Sibillini, Ussita, Frontignano, S-Hsng M. Bicco oberh. Sent. 158, 1820–1880 m s.m., Kalkschutt. MTB 3045/3, 42°54′52″ N, 13°11′7″ E,” 12 Jul 2009, *Gottschlich 54829* in *Hieracia* Europ. Sel. #352 (holotype: FI; other isotypes: B, GOET, IBF, H, LI, M, PR, W, hb. Brandstätt., hb. Dunkel, hb. Gottschl.).

***Hieracium medense*** Gottschl. and Dunkel in Stapfia 107: 12. 2017 [https://www.zobodat.at/pdf/STAPFIA_0107_0011-0028.pdf (accessed on 6 March 2024)].

**Isotype: PAL-Gr 125293**—Italy, “Friaul-Julisch Venetien, Prov. Udine, Ravascletto, Sträßchen zum Pizzo di Mede, 1302 m s.m., *Calamagrostis*-Bestand am Fichtenwaldrand. MTB 9443/4, 46°31′58″ N, 12°56′7″ E,” 7 Aug 2011, *Gottschlich-57551* in *Hieracia* Europ. Sel. #709 (holotype: FI; other isotypes: B, FR, H, IBF, LI, M, PRA, W, hb. Brandstätt., hb. Dunkel, hb. Gottschl.).

***Hieracium montis-florum*** subsp. ***soldanoi*** Gottschl. in Stapfia 105: 79. 2016 [https://www.zobodat.at/pdf/STAPFIA_105_0064-0091.pdf (accessed on 6 March 2024)].

**Isotype: PAL-Gr 121163**—Italy, “Toskana, Prov. Massa-Carrara, Alpi Apuane, “Dogna della Techhia” NE Castelpoggio, 900 m s.m., Gebüsche auf Kalkfeinschutt. MTB 1826/4, 44°7′ N, 10°5′ E,” 26 May 2004, *Gottschlich-48586* in *Hieracia* Europ. Sel. #619 (holotype: FI; Isotypi: Hier. Eur. Sel. Nr. 619 (B, FR, H, IBF, LI, M, PAL, PRA, W, hb. Brandstätt., hb. Dunkel, hb. Gottschl.).

***Hieracium murorum*** subsp. ***macilentigenum*** Gottschl. in Linzer Biol. Beitr. 38: 1056. 2006 [https://www.zobodat.at/pdf/LBB_0038_2_1045-1059.pdf (accessed on 6 March 2024)].

**Isotype: PAL-Gr 45112**—Austria, “Niederösterreich, Östliche Niederösterreichische Voralpen, Mitterbach, Gemeindealpe, Umgebung Gipfelkreuz, 1600–1626 m s.m., Weiden, Latschengebüsche. MTB 8157/3, 47°48′ N, 15°14′ E,” 17 Jul 2005, *Gottschlich 50396* in *Hieracia* Europ. Sel. #81 (holotype: W #2007–20446, other isotypes: hb. Gottschlich, etc.).

***Hieracium murorum*** subsp. ***nivimarginatum*** Gottschl. in Webbia 66: 224. 2011 [54848].

**Isotype: PAL-Gr 54848**—Italy, “Marche, Prov. Macerata, M. Sibillini. Ussita: Frontignano, Abstieg neben der Seilbahn, 1400–1700 m, Kalkschutt u. Kalkrasen mit Bucheninseln. MTB 3045/3, 42°54′59″ N, 13°10′45″ E,” 12 Jul 2009, *Gottschlich 54833* (holotype: FI; other isotypes: B, M, hb. Dunkel, hb. Gottschl.).

***Hieracium necopinum*** Buttler in Strid, Mount. Fl. Greece 2: 639. 1991 [28275].

**Holotype: PAL-Gr 28275**—Greece, “Epirus (UTM: DK-4; Distr. No. 514), montes Timfi: in ascensu a lacuna media (infra refugium E.O.S. sita) ad lacunam Dhrakolimni, praesertim in latere boreali monis Ploskos, alt. 1750–2000 m. 23 Jul 1977, *Greuter 15015 & al.* p.p. (two plants) (isotypes: none).

***Hieracium neogelmii*** Gottschl. in Linzer Biol. Beitr. 32: 365. 2000 1974 (≡*Pilosella neogelmii* (Gottschl.) Gottschl. 2010) [https://www.zobodat.at/pdf/LBB_0032_1_0363-0398.pdf (accessed on 6 March 2024)].

**Isotype: PAL-Gr 45038**—Italy, “Südtirol, Prov. Trento, Brentonico: M. Baldo, M. Altissimo, “Pra delle Varsive” N Rif. Graziani, 1650- 1800 m s.m., alpine Matten. MTB: 0131/3, 49°48′ N, 10°53′ E,” 15 Jun 1984, *Gottschlich 28238*, *Festi*, *Prosser* & *Pujatti* in *Hieracia* Europ. Sel. #7 (holotype: LI; other isotypes: ROV, hb. Gottschl.).

***Hieracium neoplatyphyllum*** Gottschl. in Ber. Bayer. Bot. Ges. 77: 138. 2007 [https://www.zobodat.at/pdf/Berichte-Bayerischen-Bot-Ges-Erforschung-Flora_77_0135-0140.pdf (accessed on 6 March 2024)].

**Isotype: PAL-Gr 52610**—Italy, “Südtirol, Prov. Bozen, Vinschgau, Schnalstal: Waalweg 3 zwischen Schloß Juval und Altratteis, 800–900 m s.m., Laubmischwald. MTB 9331/2, 46°39′ N, 10°57′ E,” 5 Sep 1999, *Gottschlich-39050* in *Hieracia* Europ. Sel. #112 (holotype: FI, other isotypes: B, GOET, H, IBF, LI, M, W, hb. Brandstätt., hb. Dunkel, hb. Gottschl.).

***Hieracium niveobarbatum*** Gottschl. in Linzer Biol. Beitr. 33: 583. 2001 [https://www.zobodat.at/pdf/LBB_0033_1_0583-0594.pdf (accessed on 6 March 2024)].

**I*s*otype: PAL-Gr 45081**—Italy, “Friaul-Julisch Venetien, Prov. Udine, Forni Avoltri: zwischen Frasseneto und Sigilleto, 1115 m s.m., Schieferböschung. MTB: 9442/2, 46°34′ N, 12°48′ E,” 10 Aug 2000, *Gottschlich 41738* in *Hieracia* Europ. Sel. #50 (holotype: LI; other isotypes: FI, MFU, W, hb. Gottschl.).

***Hieracium nubitangens*** Gottschl. in Stapfia 89: 172. 2009 [https://www.zobodat.at/pdf/STAPFIA_0089_0001-0328.pdf (accessed on 6 March 2024)].

**I*s*otype: PAL-Gr 48435**—Italy, “Abruzzen, Prov. Téramo, Gran Sasso: Kammweg Vado di Corno → M. Brancastello, N-Seite, 1950–2000 m s.m., Kalkfelsrasen, Kalkfelstreppen. MTB 3547/2, 42°27′ N, 13°36′ E,” 30 Jul 2005, *Gottschlich 50521* in *Hieracia* Europ. Sel. #271 (holotype [to be specified]: FI; other isotypes: B, FI, GOET, IBF, H, LI, M, PR, W, hb. Brandstätter, hb. Dunkel, hb. Gottschl.).

***Hieracium orodoxum*** Gottschl. in Stapfia 89: 68. 2009 [https://www.zobodat.at/pdf/STAPFIA_0089_0001-0328.pdf (accessed on 6 March 2024)].

**I*s*otype: PAL-Gr 45200**—Italy, “Abruzzen, Prov. L’Áquila, Campo di Giove, M. Porrara: N-Ende der “Paradina,” Kammjoch, 1750 m s.m., zerklüfteter Kalkfels. MTB 4050/2, 41°59′ N, 14°5′ E,” 29 Jul 2004, *Gottschlich 48951* in *Hieracia* Europ. Sel. #169 (holotype: FI [to be specified]; other isotypes: B, FI, GOET, IBF, H, LI, M, PR, W, hb. Brandstätter, hb. Dunkel, hb. Gottschl.).

***Hieracium orodoxum*** subsp. ***pseudonaegelianum*** Gottschl. in Stapfia 105: 79. 2016 [https://www.zobodat.at/pdf/STAPFIA_105_0064-0091.pdf (accessed on 6 March 2024)].

**I*s*otype: PAL-Gr 121153**—Italy, “Toskana, Prov. Lucca, Alpi Apuane, Val Serenaia, sent. 179, unterhalb “Cresta Garnerone,” 1476 m s.m., Marmorfelsspalten. MTB 1827/3, 44°7′41″ N, 10°11′21″ E,” 11 Aug 2014, *Gottschlich 63271*, *Soldano* and *Soldano* in *Hieracia* Europ. Sel. #609 (holotype: FI; other isotypes: B, FR, H, IBF, LI, M, PRA, W, hb. Brandstätter, hb. Dunkel, hb. Gottschl.).

***Hieracium orsierae*** Gottschl. in Stapfia 95: 34. 2011 [https://www.zobodat.at/pdf/STAPFIA_0095_0033-0045.pdf (accessed on 6 March 2024)].

**I*s*otype: PAL-Gr 52613**—Italy, “Piemont, Prov. Torino, Valle di Susa, Straße zwischen Susa und Colle di Finestre bei “Piano del Tiraculo,” 2165 m s.m., Zwergstrauchheiden. MTB 0908/1, 45°4′21″ N, 7°3′13″ E,” 8 Aug 2010, *Gottschlich 55947* in *Hieracia* Europ. Sel. #360 (holotype: FI; other isotypes: B, GOET, H, IBF, LI, M, PRC, W, hb. Brandstätt., hb. Dunkel, hb. Gottschl.).

***Hieracium permaculatum*** subsp. ***cacuminicola*** Gottschl. in Webbia 66: 226. 2011 [52607].

**I*s*otype: PAL-Gr 52607**—Italy, “marche. Prov. Pésaro-Urbino, Nordseite M. Nerone, 1470 m s.m., Kalkfelsstufen, Kalkschutt. MTB 2441/1, 43°33′30″ N, 12°31′3″ E,” 10 Jul 2010, *Gottschlich 55485* in *Hieracia* Europ. Sel. #354 (holotype: FI; other isotypes: B, GOET, IBF, H, LI, M, PESA, PR, W, hb. Brandstätt., hb. Dunkel, hb. Gottschl.).

***Hieracium pietrae*** Zahn in Notizbl. Bot. Gart. Berlin-Dahlem 9: 422. 1925 [https://www.jstor.org/stable/3994407?origin=crossref (accessed on 6 March 2024)].

**Isoneotype: PAL-Gr 45239**—Italy, “Abruzzo, Prov. Áquila, Gran Sasso M. Corvo: Trockental “Solagne” SE Stazzo di Solagne, 1700–1900 m s.m., lichter Buchenwald, Kalkschutt. MTB 3546/2, 42°28′ N, 13°27′ E,” 27 Jul 2004, Gottschlich 48919 in *Hieracia* Europ. Sel. #210 (neotype [Gottschlich in Stapfia 89: 160. 2009: [https://www.zobodat.at/pdf/STAPFIA_0089_0001-0328.pdf; to be specified]): FI; other isoneotypes: B, FI, GOET, IBF, H, LI, M, PR, W, hb. Brandstätt., hb. Dunkel, hb. Gottschl.).

***Hieracium pintodasilvae*** de Retz in Agron. Lusit. 35: 307. 1974 (≡*Pilosella pintodasilvae* (de Retz) Mateo, Cat. Fl. Prov. Teruel: 143. 1990) [9226].

**Isotype: PAL-Gr 9226**—Portugal, “Bragança (Lusitania, Trás-os-Montes e Alto Douro), pr. Carragosa, 850 m s.m., in pratis,” 12 Jul 1971, *Pinto da Silva*, *Teles* and *Martins 8849* in Soc. Échange Pl. Vasc. Europe Occid. Bassin Médit. #7027 (holotype: LISE #75233; other isotypes: many).

***Hieracium pseudogrovesianum*** Gottschl. in Stapfia 89: 156. 2009 [https://www.zobodat.at/pdf/STAPFIA_0089_0001-0328.pdf (accessed on 6 March 2024)].

**Isotype: PAL-Gr 45275**—Italy, “Abruzzen, Prov. L’Áquila, Monte Simbruini: Capistrello → Valico Serra S. Antonio, 1150 m s.m., Straßenböschung. MTB 4046/1, 41°57′ N, 13°21′ E,” 5 Jun 2006, *Gottschlich 51143* in *Hieracia* Europ. Sel. #246 (holotype [to be specified]: FI; other isotypes: B, FI, G, GOET, IBF, H, LI, M, PR, W, hb. Brandstätt., hb. Dunkel, hb. Gottschl.).

***Hieracium pseudogrovesianum*** subsp. ***opertum*** Gottschl. in Stapfia 89: 158. 2009 [https://www.zobodat.at/pdf/STAPFIA_0089_0001-0328.pdf (accessed on 6 March 2024)].

**Isotype: PAL-Gr 45276**—Italy, “Abruzzen, Prov. L’Áquila, Gran Sasso, Campo Imperatore: Trockental W Canali di Marcegli, 1600 m s.m., Kalkfels, Felsrasen, Schutt. MTB 3648/1, 42°23′ N, 13°42′ E,” 25 Jul 2004, *Gottschlich 48863* in *Hieracia* Europ. Sel. #247 (holotype [to be specified]: FI; other isotypes: B, FI, GOET, IBF, H, LI, M, PR, W, hb. Brandstätt., hb. Dunkel, hb. Gottschl.).

***Hieracium pseudolachenalii*** Gottschl. and Dunkel in Stapfia 109: 4. 2018 [https://www.zobodat.at/pdf/STAPFIA_0109_0003-0024.pdf (accessed on 6 March 2024)].

**Isotype: PAL-Gr 129007**—Greece, “Thessalien, Nom. Magnísia, Volos, SW Chanía, 750 m NW Agriolefkes, Mt. Pilio Ski Resort, 1220 m s.m., Straßenböschung unter Buchenwald. 39°23′28″ N, 23°4′35″ E,” 5 Jul 2013, *Dunkel 30124* in *Hieracia* Europ. Sel. #802 (holotype: B; other isotypes: FI, FR, H, IBF, LI, M, UPA, W, hb. Brandstätt., hb. Dunkel, hb. Gottschl.).

***Hieracium racemosum*** subsp. ***pulmonariifolium*** Gottschl. in Stapfia 89: 148. 2009 [https://www.zobodat.at/pdf/STAPFIA_0089_0001-0328.pdf (accessed on 6 March 2024)].

**Isotype: PAL-Gr 45261**—Italy, “Abruzzen, Prov. Téramo, Monti della Laga: Ceppo, 1340 m s.m., Buchen-Bergahorn-Wald. MTB 3346/2, 42°40′ N, 13°27′ E,” 26 Jul 2002, *Gottschlich 46119* in *Hieracia* Europ. Sel. #232 (holotype: FI; other isotypes: B, FI, GOET, IBF, H, LI, M, PR, W, hb. Brandstätt., hb. Dunkel, hb. Gottschl.).

***Hieracium ragognae*** Gottschl. in Stapfia 107: 12. 2017 [https://www.zobodat.at/pdf/STAPFIA_0107_0011-0028.pdf (accessed on 6 March 2024)].

**Isotype: PAL-Gr 125287**—Italy, “Friaul-Julisch Venetien, Prov. Udine, Ragogna, Monte di Ragogna: “tornante panoramico,” 358 m s.m., schattige Straßenböschung mit *Salix*-Gebüsch. MTB 9843/2, 46°11′13″ N, 12°57′58″ E,” 27 Jul 2012, *Gottschlich 59008* in *Hieracia* Europ. Sel. #704 (holotype: FI; other isotypes: B, FR, H, IBF, LI, M, PRA, W, hb. Brandstätt., hb. Dunkel, hb. Gottschl.).

***Hieracium retyezatense*** subsp. ***macilentoides*** Gottschl. and Dunkel in Stapfia 111: 25. 2019 [https://www.zobodat.at/pdf/STAPFIA_0111_0005-0032.pdf (accessed on 6 March 2024)].

**Isotype: PAL-Gr 129013**—Greece, “Ostmakedonien, Nom. Xanthi, Lykodromio, Ag. Georgios → M. Erymanthos, 1350 m s. m. 41°18′12″ N, 24°43′42″ E,” 24 Jun 2019, *Gottschlich 72671* and *Dunkel*, in *Hieracia* Europ. Sel. #808 (holotype: B; other isotypes: FI, FR, H, IBF, LI, M, MSTR, UPA, W, hb. Brandstätt., hb. Dunkel, hb. Gottschl.).

***Hieracium rottii*** Gottschl. in Linzer Biol. Beitr. 39: 727. 2007 [https://www.zobodat.at/pdf/LBB_0039_2_0727-0730.pdf (accessed on 6 March 2024)].

**Isotype: PAL-Gr 45161**—Italy, “Piemont, Prov. Vercelli, Varallo, Gambararo, Nordportal des südlichen Eisenbahntunnels, 440 m s.m, Mauerfugen. MTB 0115/4, 45°48′ N, 8°16 E,” 30 May 2007, *Gottschlich 40509*, *Soldano* and *Rotti* in *Hieracia* Europ. Sel. #130 (holotype: FI, other isotypes: B, GOET, H, IBF, LI, M, W, hb. Brandstätt., hb. Dunkel, hb. Gottschl.).

***Hieracium schmidtii*** subsp. ***cochleariforme*** Gottschl. in Webbia 66: 228. 2011 [52604].

**Isotype: PAL-Gr 52604**—Italy, “Umbrien, Prov. Perúgia, Monti Sibillini, Forca Campanina, Straße Richtung Nórcia, NW Costa Jovine, 1245 m s.m., Straßenrand. MTB3244/2, 42°46′2″ N, 13°9′37″ E,” 28 May 2009, *Gottschlich 54606* in *Hieracia* Europ. Sel. #351 (holotype: FI; other isotypes: B, GOET, H, IBF, LI, M, PR, W, hb. Brandstätt., hb. Dunkel, hb. Gottschl.).

***Hieracium schmidtii*** subsp. ***crinitisquamum*** Gottschl. in Stapfia 89: 95. 2009 [https://www.zobodat.at/pdf/STAPFIA_0089_0001-0328.pdf (accessed on 6 March 2024)].

**Isotype: PAL-Gr 45227**—Italy, “Abruzzen, Prov. L’Áquila, L’Áquila, Camarda Richtung Aragno: W Autobahn, beim Wasserwerk, 1010 m s.m., Eichen-Mischwald. MTB 3646/2, 42°23′ N, 13°27′ E,” 29 May 2006, *Gottschlich 50994* in *Hieracia* Europ. Sel. #198 (holotype [to be specified]: FI; other isotypes: B, FI, GOET, IBF, H, LI, M, PR, W, hb. Brandstätt., hb. Dunkel, hb. Gottschl.).

***Hieracium scorzonerifolium*** subsp. ***nudissimum*** Gottschl. in Stapfia 89: 77. 2009 [https://www.zobodat.at/pdf/STAPFIA_0089_0001-0328.pdf (accessed on 6 March 2024)].

**Isotype: PAL-Gr 45207**—Italy, “Abruzzen, Prov. Chieti, Majella, Abstieg M. Cavallo Richtung Selvaromana, Valle del Infierno,” 1200–1400 m s.m., artenreiche Hochstauden. MTB 3850/4, 42°0′ N, 14°9′ E,” 1Aug 2002, *Gottschlich 46300* and *Conti* in *Hieracia* Europ. Sel. #178 (holotype [to be specified]: FI; other isotypes: B, FI, GOET, H, IBF, LI, M, PR, W, hb. Brandstätt., hb. Dunkel, hb. Gottschl.).

***Hieracium segusianum*** Gottschl. in Stapfia 95: 35. 2011 [https://www.zobodat.at/pdf/STAPFIA_0095_0033-0045.pdf (accessed on 6 March 2024)].

**Isotype:PAL-Gr 52614**—Italy, “Piemont, Prov. Torino, Valle di Susa, S unterhalb Colle di Finestre, 1995 m s.m., im Schatten von Lärchen und Wacholderbüschen. MTB: 0908/1, 45°4′44″ N, 7°3′8″ E,” 9 Aug 2010, *Gottschlich 55979* in *Hieracia* Europ. Sel. #361 (holotype: FI; other isotypes: B, GOET, H, IBF, LI, M, PRC, W, hb. Brandstätt., hb. Dunkel, hb. Gottschl.).

***Hieracium semipallescens*** Gottschl. in Webbia 66: 210. 2011 [52609].

**Isotype: PAL-Gr 52609**—Italy, “Marche, Prov. Pésaro-Urbino: Nordseite M. Nerone, 1470 m s.m., Kalkfelsstufen, Kalkschutt. MTB 2441/1, 43°33′30″ N, 12°31′3″ E,” 10 Jul 2010, *Gottschlich 55844* in *Hieracia* Europ. Sel. #356 (holotype: FI; other isotypes: B, GOET, H, IBF, LI, M. PR, W, hb. Brandstätt., hb. Dunkel, hb. Gottschl.).

***Hieracium squarrosofurcatum*** Gottschl. in Stapfia 105: 75. 2016 [https://www.zobodat.at/pdf/STAPFIA_105_0064-0091.pdf (accessed on 6 March 2024)].

**Isotype: PAL-Gr 121159**—Italy, “Toskana, Prov. Lucca, Alpi Apuane, Arni, Le Gobbie, Passo degli Uncini, 1370–1400 m s.m., aufgelichteter Buchenwald. MTB 1927/1, 44°3′8″ N, 10°13′35″ E,” 1 Jul 2015, *Gottschlich-64318* in *Hieracia* Europ. Sel. #615 (holotype: FI; other isotypes: B, FR, H, IBF, LI, M, PRA, W, hb. Brandstätt., hb. Dunkel, hb. Gottschl.).

***Hieracium stranigense*** Gottschl. in Linzer Biol. Beitr. 32: 373. 2000 [https://www.zobodat.at/pdf/LBB_0032_1_0363-0398.pdf (accessed on 6 March 2024)].

**Isotype: PAL-Gr 45130**—Austria, “Kämten, Karnische Alpen, Stranig: Fahrweg zur Staniger Alm oberhalb des Stranig-Baches 50 m unterhalb der Gedenktafel für Georg Hohenwarter, 1350 m s.m., durch Wegebau angerissene Steilböschung (paläozoische Schiefer) mit Grünerlen, darüber Buchen-Fichten-Wald. MTB 9344/4, 46°36′ N, 13°8′ E,” 11 Aug 1993, *Gottschlich 22471* in *Hieracia* Europ. Sel. #99 (holotype: LI; other isotypes: KL, M, hb. Gottschl.).

***Hieracium thesauranum*** Gottschl. in Stapfia 89: 126. 2009 [https://www.zobodat.at/pdf/STAPFIA_0089_0001-0328.pdf (accessed on 6 March 2024)].

**Isotype: PAL-Gr 45256**—Italy, “Abruzzen, Prov. L’Áquila, Roccaraso, an der Straße zum Wintersportgebiet “Aremogna,” oberhalb der großen Hotels, 1530 m s.m., Kalkschutthang. MTB 4150/1, 41°50′ N, 14°3′ E,” 1 Jun 2001, *Gottschlich 43903* in *Hieracia* Europ. Sel. #227 (holotype [to be specified]: FI; other isotypes: B, FI, GOET, H, IBF, LI, M, PR, W, hb. Brandstätt., hb. Dunkel, hb. Gottschl.).

***Hieracium thessalonikense*** Gottschl. and Dunkel in Stapfia 111: 15. 2019 [https://www.zobodat.at/pdf/STAPFIA_0111_0005-0032.pdf (accessed on 6 March 2024)].

**Isotype: PAL-Gr 129008**—Greece, “Zentralmakedonien, Nom. Thessaloniki, Mount Chortiatis: Kisson, nahe dem Sperrgebiet der Sendemastanlage, 934 m s.m., Eichenwald. 40°35′51″ N, 23°6′18″E,” 11 Jul 2018, *Gottschlich 70162* in *Hieracia* Europ. Sel. #803 (holotype: B; other isotypes: FI, FR, H, IBF, LI, M, MSTR, UPA, W, hb. Brandstätt., hb. Dunkel, hb. Gottschl.).

***Hieracium tommasinii*** subsp. ***adenothyrsum*** Sagorski and Zahn in Magyar Bot. Lapok 6: 223. 1907 [=*Hieracium macrodontoides* (Zahn) Zahn 1909] [https://www.zobodat.at/pdf/Ungarische-Botanische-Blaetter_6_0212-0229.pdf (accessed on 6 March 2024)].

**Lectotype [designated here]: PAL-Gr 66707**—Montenegro, “in einer Doline bei Njegus. Die zahlreichen Drüssen von *stupposum*, Habitus undBlattform von *racemosum*, Behaarung und kleine Köpfe von *stepposum.* Zahn,” Jul 1904, *Sagorski* (isolectotypes: none known).

***Hieracium torrigliense*** Gottschl. in Webbia 64: 184. 2009 [49584].

**Isotype: PAL-Gr 49584**—Italy, “Ligurien, Prov Génova, Propata → Capp.la Tre Croci, 1195 m s.m., grasige Straßenböschung unter Buchenwald. MTB 1421/3, 44°34′32″ N, 9°11′45″ E,” 27 Jun 2008, *Gottschlich 53414* in *Hieracia* Europ. Sel. #301 (holotype: FI; other isotypes: B, GOET, H, IBF, LI, M, PRA, W, hb. Brandstätt., hb. Dunkel, hb. Gottschl.).

***Hieracium** truttae*** Gottschl. in Stapfia 107: 16. 2017 [https://www.zobodat.at/pdf/STAPFIA_0107_0011-0028.pdf (accessed on 6 March 2024)].

**Isotype: PAL-Gr 125289**—Italy, “Friaul-Julisch Venetien, Prov. Udine, Paluzza, Sutrio, Fahrweg zum Hotel “Alle Trote,” 521 m ü.m., Wegrand und Gebüsch unter Kiefern-Fichten-Wäldchen. MTB 9443/4, 46°30′5″ N, 12°59′59″ E,” 27 Jul 2015, *Gottschlich 64604* in *Hieracia* Europ. Sel. #705 (holotype: FI; other isotypes: B, FR, H, IBF, LI, M, PRA, W, hb. Brandstätt., hb. Dunkel, hb. Gottschl.).

***Hieracium vermiense*** Gottschl. and Dunkel in Stapfia 111: 18. 2019 [https://www.zobodat.at/pdf/STAPFIA_0111_0005-0032.pdf (accessed on 6 March 2024)].

**Isotype: PAL-Gr 129009**—Greece, “Zentralmakedonien, Nom. Iamthia, M. Vermio: Sraße Richtung Kato Vermio, 1390 m s.m., Kalkfelsböschung. 40°33′57″ N, 22°1′36″ E,” 17 Jun 2019, *Gottschlich 72322* and *Dunkel* in *Hieracia* Europ. Sel. #804 (holotype: B #101042256; other isotypes: FI, FR, H, IBF, LI, M, MSTR, UPA, W, hb. Brand, Hb. Gottschl.).

***Hieracium xanthoprasinophyes*** Gottschl. in Willdenowia 37: 181. 2007 [https://www.jstor.org/stable/20371337?seq=43 (accessed on 6 March 2024)].

**Isotype: PAL-Gr 45131**—Austria, “Tirol, Verwallgruppe, Galtür, Kopsstraße bei Überquerng durch denSchilift, 1700 m s.m., Schipisten mit Erlengebüschen. MTB 9026/2, 46°58′ N, 10°8′ E,” 22 Aug 1999, *Gottschlich 38955* in *Hieracia* Europ. Sel. #100 (holotype: B; other isotypes: B, H, IBF, LI, M. PRA, W, hb. Brandstätt., hb. Dunkel, hb. Gottschl., etc.).

***Holosteum breistrofferi*** Greuter and Charpin in Exsicc. Genav. Conserv. Bot. Distrib. Fasc. 2: 23. 1971 [=***Holosteum umbellatum*** subsp***. hirsutum*** (Mutel) Breistr. 1970] [6635].

**Isotype: PAL-Gr 6635**—France, “prope jugum ‘pas de l’Echelle’ ad septemtrionem pagi ‘Saint-Geniez’ provinciae Alpium inferiorum, 1200 supra maris aequor. In petrosis calcareis ad pedes rupium praeruptarum ad meridiem spectantium, uno in loco abundans in consortio *Veronicae hederifoliae* subsp. *trilobae*, *Anthrisci caucalidis*, etc., 12 May 1970, *Charpin*, *Greuter and Monthoux* in Exsicc. Genav. #57 (holotype: G; other isotypes: GB, GZU, LD, NO, OSC, RSA, SP).

***Horstrissea dolinicola*** Greuter, Gerstb. and Egli in Willdenowia 19: 391. 1990 [https://www.jstor.org/stable/3996647?seq=3 (accessed on 6 March 2024)].

**Isotypes: PAL-Gr 134863 and 134864**—Greece, Crete, “Ep. Milopotamos, Idhi Oros, Doline N des Berges Piperos an der Straße zwischen Kirche Ajios Fanourios uns Berg Skinakas (35°12′30″ N; 24°52′30″ E); lehmig-kiesiger Dolinengrund und Felswand, ca 1450 m; 7 Sep 1895, *Risse 1938* (holotype: B; other isotypes: none).

***Huttia conspicua*** Drumm. ex Harv. In Hooker’s J. Bot. Kew Gard. Misc. 7: 51. 1855 (≡*Hibbertia conspicua* (Harv.) Gilg) [https://www.biodiversitylibrary.org/page/767289#page/53/mode/1up (accessed on 6 March 2024)].

**Isotype: PAL-Gr 39666**—Australia, [“between Hutt and Murchison R, W. Australia”] (from holotype label)], [*Drummond* VI] *115* (holotype [specified here]: TCD #9741; other isotypes: K [2×], MEL [2×]).

***Hygrophila urquiolae*** Greuter, R. Rankin and Palmarola in Willdenowia 39: 288. 2010 [https://doi.org/10.3372/wi.39.39207 (accessed on 6 March 2024)].

**Isotype: PAL-Gr 44388**—Cuba, “Prov. Matanzas: Municipio Martí, Ciénaga del Majaguillar al NO de Martí, Ciénaga de Gonzalito cerca del Canal de Blanquizal, alt. 5 m, 22°58′39″ N, 80°58′06″ W. Herbazales de ciénaga,” 28 Feb 2009, *Greuter 27013 & al.* (holotype: HAJB #505; other isotypes: B, JE, NY).

***Hypericum hircinum*** var. ***albimontanum*** Greuter, Colloque OPTIMA Crète Guide Excurs.: 25. 1975 (≡***Hypericum hircinum*** subsp. ***albimontanum*** (Greuter) N. Robson) [13837].

**Holotype: PAL-Gr 13837**—Greece, Crete, “Ep. Sfakia: b. Aj. Rùmeli, 100 m ü.M. Schattige Grasborde längs Wassergräben, unter Gebüsch stellenw. hfg.; abweichend durch breitere, am Grund herzförmige, am Rand knorpelig gezähnelte Blätter,” 10 Jul 1961, *Greuter 3837* (isotypes: G, W, Z).

***Hypericum jovis*** Greuter, Colloque OPTIMA Crète Guide Excurs.: 25. 1975 [14571].

**Holotype: PAL-Gr 14571**—Greece, Crete, “Ep. Monofatsi: N-Hang des Berges Kòfinas, 1100 m.ü.M., Kalk. Hfg. in schattigen Ritzen der N-exp. Felswände; die erheblich abweichende mittelkretische Felsrasse, durch vermutlich hybride Zwischenformen mit dem Typus verbunden,” 27 May 1962, *Greuter 4571* (isotypes: G, LJU, M, UPA, W, Z).

***Hypericum perfoliatum*** subsp*. **phitosianum*** Greuter and Rain. Karl in Bot. Chron. (Patras) 22: 41. 2019 [=Hypericum cyladicum Trigas in Nordic J. Bot. 2018]. [https://www.researchgate.net/publication/349350832_Botanika_Chronika-2019_vol_22_Edited_by_W_Greuter_S_Kokkini_Y_Manetas_P_Bareka_G_Kamari#fullTextFileContent (accessed on 6 March 2024)].

**Holotype: PAL-Gr 126167**—Greece, “Nomos Kiklades, Paros, Stroumboulas SSW von Lefkes, S-Seite, SE-exponierter Hang, 695 m; (37°2′6″ N, 25°11′26″ E). Kalk, Steinfluren in der *Pistacia lentiscus-Sarcopoterium-*Phrygana. 4 Jun 2017, *Karl* (isotypes: B, C, LD, M, MSB, W).

***Iberis roseopurpurea**** (‘roseo-purpurea’)* Sagorski in Mitth. Thüring. Bot. Vereins, nov. ser., 16: 49. 1901 [=***Iberis umbellata*** L.] [https://www.zobodat.at/pdf/Mitt-thueringischen-Bot-Ver_NF_16_0033-0050.pdf (accessed on 6 March 2024)].

**Isolectotype: PAL-Gr 66357**—Bosnia and Herzegovina, “auf Steingeröll neben der Eisenbahn unterhalb des Hmubergs, bei Mostar, ca. 75 m, Jul 1901, *Sagorski* (lectotype [designated here]: JE # 6943; other isolectotypes: B, HBG [3×], JE #6944].

***Inula prostrata*** Rothm. in Bol. Soc. Brot. 13: 282. 1939 [= ***Dittrichia viscosa*** subsp. ***revoluta*** (Hoffmanns. and Link) P. Silva and Tutin 1973 [https://bibdigital.rjb.csic.es/viewer/14682/?offset=#page=377&viewer=picture&o=bookmark&n=0&q= (accessed on 6 March 2024)].

**Isotype: PAL-Gr 66296**—Portugal, “Alentejo litoral. S. Tiago do Cacém in arenaceis maritimis promontorii Cabo de Sines, 50 m s.m., 8 Sep 1938, *Rothmaler 14196* (holotype [specified here]: JE #2232; other isotypes: B, JE #2233, S).

***Isatis raimondoi*** Di Grist., Scafidi and Domina in Fl. Medit. 25(Special Issue): 298. 2015 [https://www.herbmedit.org/flora/FL25SI_297-304.pdf (accessed on 6 March 2024)].

**Isotype: PAL-Gr 66721**—Italy, “Basilicata, Pollino National Park, Mt Alpi (Latronico, Potenza), 40°7′9.65″ N, 15°59′16.95″ E, carbonate stony slopes. Alt. 1750 m,” 11 Jul 2014, *Scafidi* and *Di Gristina* (holotype: PAL #102699; other isotype: FI).

***Isoetes haussknechtii*** Troìa and Greuter in Willdenowia 45: 395. 2015 [https://doi.org/10.3372/wi.45.45303 (accessed on 6 March 2024)].

**Holotype: PAL-Gr 49913**—Greece, “Nom. Arkadia, Ep. Mantinia: 6 km NNE of Tripolis, Tripolis-Pirgos road, by the branching of the road to Nestani, alt. 650 m, 37°33′30″ N, 22°24′10″ E. Cultivated fields, wetland patches, roadsides,” 31 May 1995, *Kamari & al.* in OPTIMA Iter Medit. 7 #322 (Isotypes: B, BEO, BRNM, MA, SALA, UPA, W).

***Jacksonia cupulifera*** Meisn. in Bot. Zeitung (Berlin) 13: 27. 1855 [https://www.biodiversitylibrary.org/item/104887#page/30/mode/1up (accessed on 6 March 2024)].

**Isotype: PAL-Gr 39677**—Australia, [Western Australia, “Swan River, 1854” (from holotype label)], [*Drummond* VI] *11 *(holotype [specified here]: G #388717; other isotypes: BM, G #388716, K, LD, MEL, NSW, NY, P).

***Jacksonia macrocalyx*** Meisn. in Bot. Zeitung (Berlin) 13: 26. 1855 [https://www.biodiversitylibrary.org/item/104887#page/29/mode/1up (accessed on 6 March 2024)].

**Isotype: PAL-Gr 39678**—Australia, [Western Australia, “Swan River, 1854”] (from lectotype label), [*Drummond* VI] *15 *(holotype [specified here]: G #388691; other isotypes: BM, G #388690, E, K, MEL, NSW, NY, P, PERTH, TCD).

***Jacksonia stricta*** Meisn. in Lehmann in Bot. Zeitung (Berlin) 13: 27. 1855 [=***Jacksonia fasciculata*** Meisn. 1848] [https://www.biodiversitylibrary.org/item/104887#page/30/mode/1up (accessed on 6 March 2024)].

**Isotype:PAL-Gr 39676**—Australia, [Western Australia, “Swan River, 1854”] (from holotype label), [*Drummond* VI] *12 *(holotype [specified here]: G #388683; other isotypes: BM, G #388682, MEL [2×], NSW, P, PERTH).

***Jacksonia ulicina*** Meisn. in Bot. Zeitung (Berlin) 13: 26. 1855 [=***Jacksonia hakeoides*** Meisn. in Lehmann, Pl. Preiss. 1: 45. 1844] [https://www.biodiversitylibrary.org/item/104887#page/29/mode/1up (accessed on 6 March 2024)].

**Isotype: PAL-Gr 39675**—Australia, [Western Australia, “Swan River, 1854”] (from holotype label)], [*Drummond* VI] *13 *(holotype [specified here]: G #388685; other isotypes: BM, G #388684, K, MEL, NY, P [2×], TCD).

***Klasea moreana*** Greuter in Bocconea 25: 110. 2012 [https://www.herbmedit.org/bocconea/25_005.pdf#page=106 (accessed on 6 March 2024)].

**Holotype: PAL-Gr 51901**—Greece, “Nom. Lakonia, Ep. Epidhavros Limiras: Mt. Korakia N of the village Richea, alt. 500–700 m, 36°51′35″ N, 23°0′0″ E, 8 Jun 1995, *Iatrou* & al. In OPTIMA Iter Medit. 7, #1901 (isotypes: B, BRNM, SALA, UPA).

***Lamyropsis carpini*** Greuter in Willdenowia 9: 60. 1979 [https://www.jstor.org/stable/pdf/3996117.pdf?refreqid=fastly- (accessed on 6 March 2024)].

**Holotype: PAL-Gr 28414**—Greece, “Macedonia occidentalis (UTM: DK-3; Distr. No. 340), a pago Eptahori 2 km occidentem versus, alt. 700–750 m. In clivis sterilibus detritu schistoso margaceo obtectis,” 24 Jul 1977, *Greuter15152*, *Charpin*, *Bernardi & al.* (isotypes: ATH, B, BM, C, E, ERE, G, M, TAD, UPA).

***Lantana elenievskii*** I. E. Méndez in Kew Bull. 54: 487. 1999. [https://www.jstor.org/stable/pdf/4115832.pdf (accessed on 6 March 2024)].

**Isotype: PAL-Gr 126211**—Cuba, “Prov. Guantánamo. Mun. San Antonio del Sur, Baitiquirí, camino a la Mina de Yeso, montes secos, caliza, 200–400 m,” 13 May 1980, *Álvarez de Zayas & al.* In HFC 43147 (holotype [specified here]: HAJB #1461; other isotypes: B, HAJB #1460, JE [2×]).

***Limonium capitis-eliae*** Erben in Sendtnera 7: 65. 2001 [https://www.biodiversitylibrary.org/page/15042580#page/67/mode/1up (accessed on 6 March 2024)].

**Isotype: PAL-Gr 129315**—Italy, Sardinia, “südöstlich von Cágliari, am Rand der Staubstraße zum Capo S. Elia, auf Kalksteinfelsen und sandigen Flächen, 15–25 m; 39°11′5″ N, 9°8′45″ E,” 16 Nov 2000, *Erben E1145* (holotype: MSB; other isotype: FI).

***Limonium cephalonicum*** R. Artelari, Biosust. Melet. Gen. Limonium: 57. 1984 [38814].

**Isotype: PAL-Gr 38814**—Greece, “Ins. Kephallinia: ad castrum pagi Assos; in saxosis calcareis maritimis,” 9 Oct 1982, *Artelari 376* (holotype: UPA; other isotypes: B [2×]).

***Limonium connivens*** Erben in Mitt. Bot. Staatssamml. München 28: 353. 1989 [https://www.biodiversitylibrary.org/page/27802202#page/358/mode/1up (accessed on 6 March 2024)].

**Isotype: PAL-Gr 129640**—Spain, Balearic Islands: “Mallorca: Alcudia, nordöstlich von Alcudia, Felsküste bei El Mal Pas, 2–8 m (2n = 27),” 10 Oct 1989, *Erben E766*, *Morales Valverde* and *Rosselló* (holotype: M; other isotype: MA).

***Limonium coronense*** R. Artelari, Biosust. Melet. Gen. Limonium: 71. 1984 [38812].

**Isotype: PAL-Gr 38812**—Greece, “Prov. Messinia: prope Castrum oppidi Coroni, in saxosis calcareis maritimis,” 28 Aug 1982, *Artelari 356* (holotype: UPA; other isotypes: none known).

***Limonium himariense*** F. K. Mey. in Haussknechtia Beih., 15: 113. 2011 [52733].

**Isotype: PAL-Gr 52733**—Albania, “Himara, Dhermi, Strand, an Felsen im flachen Meer,” 7 Nov 1961, *Meyer 5964* (holotpe [specified here]: JE #16701; other isotype: JE #16702).

***Limonium longespicatum*** (‘*longispicatum*’) Erben in Mitt. Bot. Staatssamml. München 14: 555. 1978 [=***Limonium avei*** (De Not.) Brullo and Erben 1988] [https://www.biodiversitylibrary.org/page/15235890#page/577/mode/1up (accessed on 6 March 2024)].

**Isotype: PAL-Gr 129193**—Italy, “Prov. Savona: Ventimiglia, Punta delle Roccia, lehmige Felshänge am Meer,” 27 Sep 1977, *Erben E229* (holotype: M; other isotype: G).

***Limonium multifurcatum*** Erben in Sendtnera 7: 61. 2001 [https://www.biodiversitylibrary.org/page/15042576#page/63/mode/1up (accessed on 6 March 2024)].

**Isotype: PAL-Gr 129736**—Italy, Sardinia, “Prov. Sassari. Ca. 1 km westlich von Porto Cervo, feucht-sandige Flächen an der Mündung eines kleinen Flusses ins Meer,” 25 Aug 1973, *Erben E67* (holotype: MSB; other isotypes: FI, M).

***Limonium oligotrichum*** Erben and Brullo in Phytotaxa 240(1): 65. 2016 [https://phytotaxa.mapress.com/pt/issue/view/phytotaxa.240.1 (accessed on 6 March 2024)].

**Isotype: PAL-Gr 129712**—Greece, “Karpathos, Damatria,” 30 Jun 2002, *Brullo and Giusso* (MSB; other isotypes: CAT, FI).

***Limonium phitosianum*** R. Artelari in Mitt. Bot. Staatssamml. München 20: 430. 1984 [https://www.biodiversitylibrary.org/page/27801773#page/434/mode/1up (accessed on 6 March 2024)].

**Isotype: PAL-Gr 38815**—Greece, “Ins. Zakynthos: ad pomontorium Skinari, prope pagum Korithi: in saxosis calcareis maritimis,” 19 Feb 1982, *Phitos and Kamari 18978* (holotype: UPA; other isotype: B).

***Limonium saracinatum*** R. Artelari, Biosust. Melet. Gen. Limonium: 42. 1984 [38819].

**Isotype: PAL-Gr 38819**—Greece, “Ins. Kephallinia: inter promontorium Sarakinato et pagum Poros, in saxosis calcareis maritimis,” 27 Aug 1976, *Artelari 93* (holotype: UPA; other isotypes: none traced).

***Lolium scholzii*** Greuter in Bocconea 25: 111. 2012 [https://www.herbmedit.org/bocconea/25_005.pdf#page (accessed on 6 March 2024)].

**Holotype: PAL-Gr 53056**—Greece, “Nom. Messinia, Ep. Kalamata: Taijetos Pass between Tripi and Artemisio, alt. 1200–1350 m, 37°4′0″ N, 22°16′0″ E, 10 Jun 1995, *Kamari* & al. In OPTIMA Iter Medit. 7: #2056 (isotypes: B, UPA).

***Malcolmia graeca*** var. ***tenuior*** Hausskn. in Mitth. Thüring. Bot. Vereins, ser. 2, 3-4: 108. 1893 [https://www.zobodat.at/pdf/Mitt-thueringischen-Bot-Ver_NF_3-4_0096-0116.pdf (accessed on 6 March 2024)].

**I*s*otype: PAL-Gr 66705**—Greece, “Attica: Laurion,” 1885, *Haussknecht* (holotype [specified here]: JE #5696–5697 [2 sheets]; other isotypes: JE #6132–6133 [2 sheets], LD).

***Malcolmia × hybrida*** Hausskn. in Mitth. Thüring. Bot. Vereins, ser. 2, 3-4: 108. 1893 [https://www.zobodat.at/pdf/Mitt-thueringischen-Bot-Ver_NF_3-4_0096-0116.pdf (accessed on 6 March 2024)].

**I*s*olectotype: PAL-Gr 66696**—Greece, “Laurion,” 1885, *Haussknecht* (lectotype [designated here]: JE #7424¸other isolectotypes: JE ##7420, 7421, 7422, 7423).

***Mandragora haussknechtii*** Heldr. in Mitth. Geogr. Ges. Jena 4: 78. 1886 [=***Mandragora autumnalis*** Bertol. in Elench. Pl. Hort. Bot. Bon.: 6. 1820] [https://zs.thulb.uni-jena.de/pdf/Mitt-Geog-Ges-Thuer-und-Organ-Bot-Ver_4_0075-0080.pdf (accessed on 6 March 2024)].

**Isotype: PAL-Gr 66698**—Greece, “Korinth; am Meeresufer,” 28 Apr 1885, *Haussknecht* (holotype [specified here]: JE #1663¸other isotypes: JE ##1661–1662, PI).

***Maytenus** **revoluta*** subsp. ***bissei*** Mory in Feddes Repert. 96: 549. 1985 [48553].

**Isotypes: PAL-Gr 48553 and 128076**—Cuba, “Prov. Holguín, Mun. Moa. La Veguita, monte la Breña, alrededores del campamento Los Carboneros, pluviosilva destruída, cerca 400–500 m s.m,” 14 Apr 1981, *Bisse & al.* in HFC 44129 (holotype [specified here]: HAJB #676; other isotypes: B [2×], HAJB, JE, NY).

***Medicago heyniana*** Greuter in Candollea 25: 190. 1970 [https://www.e-periodica.ch/digbib/view?pid=can-002%3A1970%3A25%3A%3A4#218 (accessed on 6 March 2024)].

**Holotype: PAL-Gr 15346**—Greece, “Dodecanesus, ins. Kárpathos: in latere austro-orientali et ad cacumen montis Kollas, 750 et 900 m supra maris libram. In herbidis inter saxa, solo calcareo, perrara (duo specimina tantum detexi). 17 May 1963, *Greuter 5346* (isotypes: none).

***Medicago strasseri*** Greuter, Matthäs, and Risse in Willdenowia 12: 201. 1982 (≡***Medicago arborea*** subsp. ***strasseri (***Greuter, Matthäs, and Risse) Sobr.-Vesp. and Ceresuela, 2000) [https://www.jstor.org/stable/3995928 (accessed on 6 March 2024)].

**Isotype: PAL-Gr 34773**—Greece, Crete, “Eparchie Rethimni: Petres-Schlucht N von Karoti, 30°20′30″ N, 24°21‘30″ E. Kalkfelswände, 30 m. 26 May 1982, *Greuter 19632*, *Matthäs and Risse* (holotype: B; other isotypes: LD).

***Merendera greuteri*** Gabrieljan in Fl. Rastitel’nost’ Rast. Res. Arm. 12: 15. 1999 (≡***Colchicum greuteri*** (Gabrieljan) K. Perss. 2007) [38798].

**Isotype: PAL-Gr 38798**—Armenia, “Shirak distr., in vicinitate pag. Areg, m. Arteni in steppis tragacanthaceis, 1500–1700 m s.m.,” 9 Apr 1998, *Gabrielian* in ERE #147519 (holotype: ERE #661; other isotypes: B, ERE #662, 663).

***Minuartia greuteriana*** Kamari in Willdenowia 25: 99. 1995 [https://www.jstor.org/stable/3996976?seq=1 (accessed on 6 March 2024)].

**Isotypes (2): PAL-Gr 45665 and 38805**—Greece, “Nomos Evros, Eparhia Soufli, 1 km S of Dhadhia, 100 m, 41°7′20″ N, 26°13′ E. Grassland on micaschist hill,” 13 Jun 1992, *Greuter 23299* (holotype: UPA; other isotypes B [3×], C, G, LD).

***Minuartia greuteriana*** var. ***pessana*** Kamari in Willdenowia 25: 101. 1995 [https://www.jstor.org/stable/3996976?seq=3 (accessed on 6 March 2024)].

**Isotype: PAL-Gr 32305**—Greece, “Thraki, eparhia Soufli, montes Boukate dag, a pago derelicto Pessani 6 km septemtriones versus, in colle ad occidentem stationis televisoriae, alt. 400 m. In cacumine rupestri granitico e sylva *Quercus frainetto* proeminente. In fissuris,” 15 Jul 1978, *Greuter 15988* (holotype: UPA; other isotypes B, C).

***Minuartia kamariana*** Greuter in Bocconea 25: 113. 2012 [? = *Minuartia attica* (Boiss. and Spruner) Vierh. 1914) [https://www.herbmedit.org/bocconea/25_005.pdf#page=109 (accessed on 6 March 2024)].

**Holotype:PAL-Gr 51721**—Greece, “Nom. Lakonia, Ep. Epidhavros Limiras: Between Metamorfosi and Richea, summit area of Mt. Koulochera, alt. 800–950 m, 36°49′40″ N, 22°58′50″ E, 6 Jun 1995, *Kamari & al.* In OPTIMA Iter Medit. 7: #1721 (isotypes: B, BRNM, MA, SALA, UPA).

***Momordica lanata*** Thunb., Prodr. Pl. Cap.: 13. 1794 (≡*Citrullus lanatus* (Thunb.) Matsum. and Nakai 1916) [https://bibdigital.rjb.csic.es/viewer/14226/?offset=#page=25&viewer=picture&o=bookmark&n=0&q= (accessed on 6 March 2024)].

**Isotype (by conservation): PAL-Gr 62559**—USA, “Grown in S. Renner’s garden in St. Louis from the seed of watermelon ‘Crimson Sweet’ bought in a store in St. Louis,” *Renner 2816* (holotype: M; other isotypes: B, BM, K, L, LE, MO, P).

***Myosotis refracta*** subsp. ***aegagrophila*** Greuter and Grau in Candollea 25: 8. 1970 [https://www.e-periodica.ch/digbib/view?pid=can-002%3A1970%3A25%3A%3A4#13 (accessed on 6 March 2024)].

**Holotype: PAL-Gr 14855**—Greece, Crete “Ep. Sfakia: b. der Quelle Dzaràni am W-Abfall des Berges Dzaranokefàla, 1850 m.ü.M., Kalk. Schattige Höhlung der Felswand mit lehmiger Komposterde (Ziegenläger), lokal reichl.; durch grössere Zartheit u. schwächere Behaarung aller Teile, längere, weniger gekrümmte Fr-Stiele abweichend,” 4 Jul 1962, *Greuter 4855* (isotypes: G, LD, M, W, Z).

***Myosotis solange*** Greuter and Zaffran in Willdenowia 11: 38. 1981 [https://www.jstor.org/stable/3995788?seq=16 (accessed on 6 March 2024)].

**Holotype: PAL-Gr 1844**—Greece, Crete, “Lefka Ori N., Hagios Pneuma, Falaise calcaire à 1500 m, exposition NE,” 20 Jul 1964, *Zaffran 5907* (isotype: hb. Zaffran).

***Myrcia urquiolae*** Mory in Phytotaxa 549(1): 113. 2022 [https://doi.org/10.11646/phytotaxa.549.1.10 (accessed on 6 March 2024)].

**Isotype: PAL-Gr 129938**—Greece, Cuba, “Prov. Holguín, Mpio. Mayarí: charrascales de La Caridad cerca del río Naranjo. Alt. 300–450 m s.m., 20.474592° N, 75.740322° W,” 13 Jun 2018, *Acosta & Gómez* in HFC 42165 (holotype: HAJB #1255; other isotypes: B, FTG, HAC, HAJB #1254, HAJU, herb. Jard. Bot. Holguín).

***Myrsine pipolyi*** Panfet in Willdenowia 33: 177. 2003 [https://doi.org/10.3372/wi.33.33117 (accessed on 6 March 2024)].

**Isotype: PAL-Gr 58192**—Greece, Cuba, “Prov. Sancti Spíritus. Trinidad. Topes de Collantes. Mogote mi Retiro,” *Olmstead & al.* in HFC 72727 (holotype [specimen shown in Willdenowia 33: 177, in Fig. 3]: HAJB #711; other isotypes: B, HAJB ##710, 712 [2×], JE).

***Narcissus tortifolius*** Fern. Casas in Saussurea 8: 43. 1977 [26657].

**Isotype: PAL-Gr 26657**—Spain, “Almería, Sorbas: Venta de los Castaños, 2oS WG 8511, ad 400 m, in gypsaceis siccis,” 23 Mar 1977, *Fernández Casas 1484 and Pueche* (holotype [specified here]: MA #227840; other isotypes: BCN, C, MA #208486.

***Nepeta hystrix*** Greuter in Bocconea 25: 113. 2012 [https://www.herbmedit.org/bocconea/25_005.pdf#page=109 (accessed on 6 March 2024)].

**Holotype: PAL-Gr 53511**—Greece, “Nom. Messinia, Ep. Kalamata: 6–8 km NE of Ano Amfia along the road to Poliana, alt. 600–800 m. 37°8′20″ N, 22°7′30″ E. Dry river bed and rocky limestone slopes with maquis,” 14 Jun 1995, *Kamari* & *al*. In OPTIMA Iter Medit. 7: #2511 (isotypes: B, BEO, BRNM, MA, SALA, UPA, W, etc.).

***Nigella carpatha*** Strid in Opera Bot. 28: 49. 1970 [15134].

**Holotype: PAL-Gr 15134**—Greece, “Regio aegaea austr., Insula Karpathos: ad boreo-occidentem porti Finiki versus promontorium, alt. 8–15 m. In petroso-argillosis plani sublitoralis. Ad pedes muriculi, uno in loco copiose,” 11 May 1963, *Greuter 5134* (isotypes: G, LD, HUJ, M, W, Z).

***Oenanthe tricholoba*** Greuter in Bocconea 25: 114. 2012 [https://www.herbmedit.org/bocconea/25_005.pdf#page=110 (accessed on 6 March 2024)].

**Holotype: PAL-Gr 49916**—Greece, “Nom. Arkadia, Ep. Mantinia: 6 km NNE of Tripolis, Tripolis-Pirgos road, by the branching of the road to Nestani, alt. 650 m, 37°33′30″ N, 22°24′10″ E,” 31 May 1995, Kamari & al. in OPTIMA Iter Medit. 7 #325 (isotypes: B, UPA, SALA, BRNM, MA, BEO, W, etc.).

***Onobrychis aragatzi*** Arevsch. in Novosti Sist. Vysš. Rast. 41: 93. 2010 [48158].

**Isotype: PAL-Gr 48158**—Armenia, “Aragatz mountain, South macroslope, meadow-steppe, varied grasses, 1800 m a.s.l.,” 19 Jul 2007, *Arevschatian 2009* (holotype [specified here]: ERE #531; other isotypes: ERE ##532, 533, LE).

***Onobrychis sphaciotica*** Greuter in Candollea 20: 213. 1965 [https://www.e-periodica.ch/digbib/view?pid=can-002%3A1965%3A20#217 (accessed on 6 March 2024)].

**Holotype: PAL-Gr 14638**—Greece, Crete, “Ep. Sfakia: S-Abfall des Grates swl. des Xilòskala-Passes, 1400–1500 m.ü.M., Plattenkalk. Verbr. u. hfg. in Ritzen sonniger Felswände, in grossen Büschen; schöne neue Art aus der sect. *Eubrychis* subsect. *Albae* ser. *Verae*,” 2 Jun 1962, *Greuter 4638* (isotypes: ATH, G, HUJ, LD, LJU, M, UPA, W, Z, hb. Creutzburg, Hb. Zaffran).

***Onosma albanica*** (‘*albanicum*’) Dörfl. and Ronniger in Anz. Akad. Wiss. Wien, Math.-Naturwiss. Kl. 55: 283. 1918 (≡***Onosma pseudoarenaria*** subsp. ***albanica*** (Dörfl. and Ronniger) Rauschert 1976) [=***Onosma pseudoarenaria*** subsp. ***fallax*** (Borbás) Rauschert 1976 ≡ *Onosma helvetica* subsp. *fallax* (Borbás) Teppner 1971] [https://www.zobodat.at/pdf/AAWW_55_0001-0404.pdf (accessed on 6 March 2024)].

**Isotype: PAL-Gr 5314**—Albania, “Distr. Luma. Zwischen Eichengebüsch auf Weiden am Wege von Kula Luma nach Göstil,” [25 Jun 1916/1918, *Dörfler 687*] (Holotype (specified by Vogt & al. in Willdenowie 48: 78. 2018): B #9006177; other isotypes: WU, Z).

***Onosma psammophila*** (‘*psammophilum*’) Rech. f. and Riedl in Bot. Not. 124: 79. 1971 [30626].

**Isotype: PAL-Gr 30626**—Greece, “Macedonia orientalis: in arenosis maritimis ad sinum Orphani ab ostiis fluvii Strymon c. 10 km occidentem versus,” 21 Jul 1970, *Rechinger 39000* (holotype: W; other isotypes: B, G).

***Ophrys argolica*** H. Fleischm. ex Vierh.? in Verh. K. K. Zool.-Bot. Ges. Wien 69: 295. 1919 [https://www.biodiversitylibrary.org/page/30341427#page/519/mode/1up (accessed on 6 March 2024)].

**Isolectotypes (2): PAL-Gr 6445 and 8516**—Greece, “bei Argos, Tiryns-Ruinen,” 17 Apr 1911, *Müllner* (lectotype [designated here]: WU [Halácsy, Greek hb.]; other isolectotypes: GB?, WU?).

***Ophrys candica*** Greuter, Matthäs and Risse in Willdenowia 15: 53. 1985 [=***Ophrys holosericea*** subsp. ***candica*** (E. Nelson ex Soó) Renz and Taubenheim [https://www.jstor.org/stable/3996541?seq=31 (accessed on 6 March 2024)].

**Holotype: PAL-Gr 12518**—Greece, Crete, “Ep. Kidonia: b. Dorf Prasses, 500 m.ü.M. Raine und Wegborde der Kulturzone,” 24 Apr 1960, *Greuter 2518* (isotypes: none).

***Orobanche baumanniorum*** Greuter in Willdenowia 16: 448. 1987 [https://www.jstor.org/stable/3996512?seq=10 (accessed on 6 March 2024)].

**Holotype: PAL-Gr 7093**—Greece, “Attika, Mt. Parnis. Parasitisch auf *Pterocephalus perennis*,” end May 1974, *Baumann and Baumann* (isotype: B).

***Papaver gabrielianae*** M. V. Agab. and Fragman in Fl. Rastitel’nost’ Rastitel’nye Resursy Armenii 16: 27. 2007 [44990].

**Isotype: PAL-Gr 44990**—Armenia, “Sjunik, m. Mets Ishkhanasar, road from lake Sevlich, 2680 m, 39°49′ N, 44°45′ E,” 6.Jul 2006, *Aghababian*, *Gabrielian*, *Nersessian,* and *Sarkissian* (holotype: [specified here]: ERE #635; other isotypes: ERE #636–638 [3×]).

***Papaver gorovanicum*** M. V. Agab. In Takhtajania 2: 134. 2013 [https://takhtajania.am/assets/pdf/takhtajania-2.pdf (accessed on 6 March 2024)].

**Isotype: PAL-Gr 62627**—Armenia, “Ararat distr., Gorovan, sandy soils, 39°53′40.31″ N, 44°43′59.89″ E, 940 m,” 21 May 2010, *Gabrielian* and *Aghababyan* (holotype[specified here]: ERE #639; other isotypes: B, ERE ##640–642 [3×]).

***Papaver paphium*** M. V. Agab., Christodoulou and Hand in Willdenowia 41: 349. 2011 [https://www.jstor.org/stable/41548998?seq=9 (accessed on 6 March 2024)].

**Isotype: PAL-Gr 52145**—Cyprus, “Division 2 sensu Meikle 1977/1985, Kykko, Vrysi tou Klamenou (‘Klamenon’), road towards Kykko monastery, screes, alt. 900 m,” 14 Apr 2010, *Christodoulou* (holotype [specified here]: B #100413396; other isotypes: B #100413395, CYP, ERE).

***Papaver roseolum*** M. V. Agab. And Fragman in Fl. Rastitel’nost’ Rastitel’nye Resursy Armenii 16: 27. 2007 [44991].

**Isotype: PAL-Gr 44991**—Armenia, “Avan, dry stony slopes,” 25 May 2006, *Gambarian* (holotype [specified here]: ERE #597; other isotypes: ERE ##598–602 [5×], HUJ).

***Parietaria cardiostegia*** Greuter in Fl. Australia 3: 190. 1989 [33354].

**Isotype: PAL-Gr 33354**—Australia, New South Wales, “‘Mesa’ hills c. 12 km NW of Fowlers Gap beyond NW domain border, alt. 250–300 m, 31°1′ S, 141°39′ E. Hillsides in open semi-desert area,” 4 Sep 1981, *Greuter 18387* (holotype: B; other isotypes: K, NSW).

***Petrorhagia suffruticosa*** Rech. F. and Phitos in Österr. Bot. Z. 113: 271. 1966 [=***Bolantus fruticulosus*** (Bory and Chaub.) Barkoudah] [https://www.jstor.org/stable/43337480?seq=1 (accessed on 6 March 2024)].

**Isotype: PAL-Gr 23946**—Greece, “Laconia: Peninsua Malea, in arenosis N Neapolis,” 8 May 1964, *Rechinger 24505* (holotype: W; another isotype: B, S).

***Phlomis*** × ***cytherea*** Rech. F. in Boissiera 13: 115. 1967 [https://www.e-periodica.ch/digbib/view?pid=boi-001%3A1967%3A13#119 (accessed on 6 March 2024)].

**Isotype: PAL-Gr 30640**—Greece, “Insula Kythera: Kapsali, substr. Calc., 50–200 m,” 5 May 1964, *Rechinger 24363* (‘*24505*’) (holotype: W #1981–10835; other isotypes: B, E, GZU, MA, MO, S, WU).

***Phoenix theophrasti*** Greuter in Bauhinia 3: 243. 1967 [https://botges.ch/bauhinia/Bauhinia_3_0243-0254.pdf (accessed on 6 March 2024)].

**Holotype (2 sheets): PAL-Gr 17650/1 and 2**—Greece, Crete: “Ep. Sitia: prope Vái, 5 m. In plano arenoso vel argilloso juxta mare lucum formans. Arbor dioica, truncis pluribus fasciculato-aggregatis. Baccae cum axibus fructiferis vivide miniatae,” 2 Oct 1966, *Greuter 7650* (isotypes: ATH, B, E, G, GB, K, LD, M, UPA, W, hb. Zaffran).

***Picrasma pauciflora*** A. Noa and P. A. González in Willdenowia 49: 189. 2019 [https://doi.org/10.3372/wi.49.49207 (accessed on 6 March 2024)].

**Isotype: PAL-Gr 126168**—Cuba: “Prov. Holguín. Rafael Freyre: Loma El Templo, al oeste de la Bahía Naranjo. Bosque semideciduo microfilo. 70–80 m s.m.,” 23 May 2017, *González & al.* In UCLV 12355 (holotype: ULV [specimen shown in Willdenowia 49: 188, Fig. 1]; other isotypes: B, HAC, HAJB, ULV).

***Plantago squarrosa*** var. ***Gaudensis*** Dörfl. Ex Vierh. And Rech.f. in Oesterr. Bot. Z. 84: 179 (1935) [https://www.jstor.org/stable/43336133?seq=19 (accessed on 6 March 2024)].

**Isotype: PAL-Gr 5298**—Greece, Crete: “Distr. Sphakia. Insel Gaudos, Im Dünensande der Nordküste. (Neu für Europa!),” 20 Mar 1904, *Dörfler* in Iter Cret. #1148 (holotype [specified here: WU #99192; other isotypes: B).

***Polygala helenae*** Greuter in Boissiera 13: 65. 1967 [https://www.e-periodica.ch/digbib/view?pid=boi-001%3A1967%3A13#69 (accessed on 6 March 2024)].

**Holotype** [specified here]: **PAL-Gr 16568**—Greece, “Aegäis, Nom. Attikì, Ep. Kìthira: Hügel südl. Kàlamas, 100 m.ü.M., Neogen. Mehrfach unter Sklerophyllen-Gebüsch,” 12 May 1964, *Greuter 6568* (isotypes: none).

***Polygala negevensis*** Danin in Israel J. Bot. 36: 67. 1987 [123562].

**Isotype: PAL-Gr 123562**—Israel: “S. Negev: 22 km NNW of Elat, crevices of hard limestone in a wadi,” 1 Jul 1986, *Danin* and *Chouate* (holotype: HUJ; other isotypes: B, G, K, MSB, P, W [2×]).

***Ptilostemon chamaepeuce*** var. ***elegans*** Greuter in Boissiera 22: 117. 1973 [https://www.e-periodica.ch/digbib/view?pid=boi-001%3A1973%3A22#120 (accessed on 6 March 2024)].

**Isotype: PAL-Gr 6018**—Greece, “Karpathos, ad rupes calcareas maritimas litor. Bor. Or. Insulae, prope promont Grea,” 25 May 1886, *Major119* (holotype: G #301451); other isotypes: K, LE).

***Ptilostemon diacantha*** subsp. ***turcicus*** Greuter in Boissiera 22: 103. 1973 [https://www.e-periodica.ch/digbib/view?pid=boi-001%3A1973%3A22#106 (accessed on 6 March 2024)].

**Isotype: PAL-Gr 6131**—Turkey, “Pentes schisteuses bordant le Guzel-Déré, en amont de Sédichig, à 4 lieues au NO. De Mersina,” 5 Jun 1855, *Balansa* in Pl. d’Orient #640 (holotype [specified here]: G #301447 [2 sheets]; other isotypes: B, BM, C, FI-W, G #301448, G-BOIS, GOET, JE, LAU, LE, MANCH, MPU, P, TL, W [2×], WAG).

***Ptilostemon greuteri*** Raimondo and Domina in Willdenowia 36: 171. 2006 [https://www.jstor.org/stable/3997692?seq=3 (accessed on 6 March 2024)].

**Isotype** (2 sheets): **PAL-Gr 44578/1 and 2**—Italy, Sicilia “Prov. Trapani, loco classico et unico. In dumetis et ad rupes calcareas clivium asperorum sylvestrium septemtriones spectantium,” 4 Jun 2005, *Raimondo*, *Greuter 26423* and *Aghababyan* (holotype: PAL; other isotypes: B, FI).

***Ptilotrichum cyclocarpum*** subsp. ***pindicum*** Hartvig in Strid, Mountain Fl. Greece 1: 305. 1986 (≡***Phyllolepidium cyclocarpum*** subsp***. Pindicum*** (Hartvig) Cecchi) [28326].

**Isotype: PAL-Gr 28326**—Greece, “Epirus (UTM: DK-4; Distr. No. 514), montes Timfi: In latere orientali montis Astraka, alt. 2000–2100 m. In clivis asperis herbosis petroso-glareosis calcareis,” 23 Jul 1977, *Greuter 15066 & al.* (holotype: C #10008875; other isotypes: ATH, G).

***Pycnolachne ledifolia*** Turcz. In Bull. Soc. Imp. Naturalistes Moscou 36(2): 215. 1863 [=***Lachnostachys eriobotrya*** (F. Muell.) Druce] **[**https://www.biodiversitylibrary.org/item/124487#page/221/mode/1up (accessed on 6 March 2024)].

**Isotype: PAL-Gr 39660**—Australia, [Western Australia, “Swan River”] (from holotype label), [*Drummond* VI] *220*, [1854]” (holotype [specified here]: KW #1001667; other isotypes: A, BM, CGE, E, G, K, LD, MEL, W).

***Ranunculus radinotrichus*** Greuter and Strid in Willdenowia 11: 267. 1981 [https://www.jstor.org/stable/3996012 (accessed on 6 March 2024)].

**Isotype: PAL-Gr 35216**—Greece, Crete, “Prov. Hania. Distr. Sfakion. Lefka Ori, Mt. Svouritchi [Svourihti], summit area, c 2 km NE of the summit of Mt. Pachnes. Alt: 2100–2300 m. N-NE facing slopes with limestone cliffs or E facing scree slopes. Petals yellow,” 12 Jul 1980, *Baden* and *Franzen 537* (holotype: C; other isotypes: none).

***Rostraria hadjikyriakou*** Christodoulou and Hand in Fl. Medit. 31: 72. 2021 [https://www.herbmedit.org/flora/FL31_071-082.pdf (accessed on 6 March 2024)].

**Isotype: PAL-Gr 129151**—Cyprus, “Division 7 (sensu Miekle 1985), Ypsarovounos forest, c. 2.5 km southwest of Mandres Ammochostou, on somewhat vertical or steep, almost bare and soft gypsum faces of few square metres, with flattish patches on or at their base, alt. c. 280 m, UTM 36 N 572184 E, 3909366 N,” 5 Apr 2018, *Hadjikyriakou 7600* (CYP; other isotypes: ARI, B, G, STU, hb. Hadjikyr.).

***Salix talenceana*** Gand., Fl. Lyonn.: 205. 1875 [https://www.biodiversitylibrary.org/page/12416181#page/275/mode/1up (accessed on 6 March 2024)].

**Holotype [specified here]: PAL-Gr 66351** p.p., fertile branches only—France, “Arnas (Rhône),” 11 Apr 1874 (isotype: LY).

***Salsola carpatha*** P. H. Davis in Notes Roy. Bot. Gard. Edinburgh 21: 139. 1953 (≡***Caroxylon carpathum*** (P. H. Davis) Akhani and Roalson 2007) [1108].

**Isotype** (holotype fragment)**: PAL-Gr 1108**—Greece, “Karpathos, Vurgunda (N.W. of Olymbos), 5–20 m. Calc. sea rocks with *Galium canum*,” 24 Jul 1950, *Davis 18025* (holotype [specified here]: K #899553; other isotypes: B, E, K #899552, S, W).

***Saponaria haussknechtii*** Simmler in Denkschr. Kaiserl. Akad. Wiss., Wien. Math.-Naturwiss. Kl. 85: 472. 1910 [https://www.biodiversitylibrary.org/item/110762#page/580/mode/1up (accessed on 6 March 2024)] (≡ *Saponaria depressa* var. *minor* Hausskn. In Mitth.Thüring Bot. Vereins, ser. 2, 5: 53. 1893) [https://books.google.it/books?id=APQZAAAAYAAJ&printsec=frontcover&hl=it#v=onepage&q&f=false (accessed on 6 March 2024)].

**Isolectotype: PAL-Gr 66367**—Greece, “Pindus Tymphaeus: in summo montis Zygos (Lakmon veter.) supra Metzovo, alt 4500–5000′, substratu silico-serpentino,” Jul 1885, *Haussknecht* (lectotype [designated here]: JE #16780; other isolectotypes: JE #16781, K).

***Saponaria jagelii*** Phitos and Greuter in Fl. Medit. 3: 277. 1993 [https://www.herbmedit.org/flora/3-277.pdf (accessed on 6 March 2024)].

**Isotype: PAL-Gr 38806**—Greece, “Nomos Lakonias, Ep. Epidavrou-Limiras, Ins. Elafonisos: in arenosis littoreis ad occidentem insulae spectantibus. 36°29′ N, 22°56′ E, 24 Mar 1993, *Phitos*, *Kamari, and Christodoulakis* 23209 (holotype: UPA; other isotype: B).

***Saxifraga camposii*** Boiss. and Reut., Pugill. Pl. Afr. Bor. Hispan.: 47. 1852 [https://bibdigital.rjb.csic.es/viewer/9477/?offset=#page=47&viewer=picture&o=bookmark&n=0&q= (accessed on 6 March 2024)].

**Isotype: PAL-Gr 66359**—Spain, “Sierra de Loxa,” Jun [1849], *del Campo* in Hohenacker, Pl. Hispan. #41 (lectotype [Webb ms. in schedis, designated here]: G #388986; other isolectotypes: BM; G [2×], JE ([6×], K [2×], MICH, WAG).

***Scabiosa albocincta*** Greuter in Candollea 22: 242. 1967 (≡***Lomelosia albocincta*** (Greuter) Greuter and Burdet 1985) [https://www.e-periodica.ch/digbib/view?pid=can-002%3A1967%3A22%3A%3A172#259 (accessed on 6 March 2024)].

**Holotype: PAL-Gr 17713**—Greece, Crete, “Ep. Sfakia: supra hiatum Kakískala propr fontrm Linoseli, 1450 m. Ad rupes praeruptas calcareas sole illustres frequens,” 11 Oct 1966, *Greuter 7713* (isotypes: ATH, B, E, G #388987, GB, K, LD, M, UPA, W).

***Scabiosa minoana*** subsp. ***asterusica*** Greuter in Candollea 22: 241. 1967 (≡***Lomelosia minoana*** subsp. ***asterusica*** (Greuter) Greuter and Burdet in Willdenowia 15: 75. 1985 [https://www.e-periodica.ch/digbib/view?pid=can-002%3A1967%3A22%3A%3A172#258 (accessed on 6 March 2024)].

**Holotype: PAL-Gr 14574** –Greece, Crete, “Ep. Monofatsi: Gipfel des Berges Kòfinas, 1200 m.ü.M., Kalk. S-exp. Felswände in grossen polsterförmigen Büschen, 27 May 1962, *Greuter 4574* (isotypes: G, HUJ, LD, W, Z).

***Scutellaria holguinensis*** I. E. Méndez in Rodriguésia 70(e02982017): 2. 2019. [https://www.scielo.br/j/rod/a/NRV7jFtbM85JKmZL4qmDgYr/?lang=es (accessed on 6 March 2024)] [122002].

**Isotype: PAL-Gr 122002**—Cuba: “Provincia de Holguín. Cerro Verde, km 10, carretera Holguín-Gibara. Colinas al este de la carretera. 80°57′80″ N, 76°15′93″ W. 180 m s.n.m. Matorral xeromorfo espinoso sobre serpentina degradado. Forma parte del estrato herbáceo que rodea los arbustos,” 21 Oct 2016, Méndez Santos and Hernández Peña in HPC 12029 (holotype: HIPC [specimen shown in Rodriguésia 70(e02982017): 3, Fig. 1]; other isotypes: B, HAC, HAJB, JE, ULV).

***Sedum praesidis*** Runemark and Greuter in Willdenowia 11: 18. 1981 [https://www.jstor.org/stable/3995787?seq=6 (accessed on 6 March 2024)].

**Isotype: PAL-Gr 14720**—Greece, Crete, “Ep. Lassithi: Eingang der diktäischen Zeusgrotte ob Psichrò, 1000 m.ü.M., Kalk. Häufig an feucht-schattigen, bemoosten Felsen,” 13 Jun 1962, *Greuter 4720* (holotype: LD; other isotypes: G, W, Z).

***Seseli halkense*** (‘*halkensis*’) C. Catt., Kit Tan and Biel in Phytol. Balcan. 22(3): 438. 2016 [http://www.bio.bas.bg/~phytolbalcan/PDF/22_3/PhytolBalcan_22-3_15_Vladimirov_&_Tan_NFRs_31.pdf (accessed on 6 March 2024)].

**Isotype: PAL-Gr 121142**—Greece, “Nomos Dhodhekanisos, Eparhia Rodhos: Chalki, Klisoures, crevices of vertical N-facing limestone cliffs at Klisoura, 200–400 m, 36°13′58.3″ N, 27°32′34.28″ E,” 2 Aug 2016, *Cattaneo CK35* (holotype C; other isotypes ATH, herb. Cattaneo).

***Silene damboldtiana*** Greuter and Melzh. in Willdenowia 8: 614. 1979 [https://www.jstor.org/stable/3996169?seq=2 (accessed on 6 March 2024)].

**Holotype: PAL-Gr 25109**—Greece, “Macedonia occ. (distr. Almopia/Peonia): montes Paiko (Meniki), in colle Pirgos ad sept. Pagi Livadhia, alt. 1400 m. In pascuis petrosis calcareis zonae silvticae *Fagi*,” 29 Jul 1976, *Greuter 13993 *(isotypes: G, LJU).

***Silene fabaria*** subsp. ***domokina*** Greuter in Willdenowia 25(1): 122 (1995). [https://www.jstor.org/stable/3996977?typeAccessWorkflow=login&seq=18 (accessed on 6 March 2024)].

**Isotype: PAL-Gr 35207**—Greece, “(Phthiotis): in collibus denudatis an fontem 4–6 km a Domokos meridiem versus, ca. 550 m, substr. serpent.,” 6 May 1961, *Rechinger 22877* (holotype: B; other isotype: G).

***Silene gigantea*** subsp. ***hellenica*** Greuter in Willdenowia 25: 114. 1995 [https://www.jstor.org/stable/3996977?seq=10 (accessed on 6 March 2024)].

**Holotype: PAL-Gr 15928/1 and 2** (2 sheets):—Greece, “Sterea, prov. Viotia, distr. Levadhia: supra pagum Arahova secus viam ad Kalivia ducentem, alt. 1050–2000 m. In rupestribus apricis clivorum asperorum meridiem spectantium,” 14 Jun 1963, *Greuter 5928* (isotypes: B, C, LD, UPA).

***Silene greuteri*** Phitos in Bot. Chron. 2: 53. 1983 (≡***Silene integrifolia*** subsp. ***greuteri*** (Phitos) Akeroyd in Bot. J. Linn. Soc. 97: 341. 1988 [35227].

**Isotype: PAL-Gr 35227**—Greece, Crete, “prov. Rethimno: prope pagum Mariou in faucibus Kourtaliotiko, alt. c. 350 m, in glareosis,” 13 Apr 1982, *Tsanoudakis 7067* (holotype: UPA; other isotypes: none traced).

***Silene intonsa*** Melzh. and Greuter in Willdenowia 12: 29. 1982 (≡***Heliosperma intonsum*** (Greuter and Melzh.) Niketić and Stevan. 2007) [https://www.jstor.org/stable/3996065 (accessed on 6 March 2024)].

**Isotype: PAL-Gr 29746**—Greece, “Ipiros: prov. Ioannina, distr. Konitsa: Konitsa. Valley of the river Aoos. On both sides of the river and E. of the bridge. Alt. 460–500 m. Stony, rocky slopes with scattered shrubs. In clefts of moist rocks. Flowers white,” 22 May 1973, *Stamatiadou 17079* (holotype: ATH; other isotype: B).

***Silene italica*** subsp. ***peloponnesiaca*** Greuter in Willdenowia 25: 110. 1995 [https://www.jstor.org/stable/3996977?seq=6 (accessed on 6 March 2024)].

**Isotype: PAL-Gr 29638**—Greece, “Nom. Messinias, Ep. Kalamon: Mt. Taygetos, S part, NE of Makrovouna, above a place called Ag. Dimitrios, along the valley leading to the summit ridge, 1600–1950 m. Chiefly limestone but some outcrops of schist. Timberline at c. 1900 m formed by *Pinus nigra* and *Abies cephalonica.* Rocky places, 1800 m, limest. Petals white above, pinkish beneath,” 30 Jun 1979, *Strid and Papanicolaou 15236* (holotype: B; other isotypes: C, G).

***Silene nutabunda*** Greuter in Willdenowia 25: 137. 1995 [https://www.jstor.org/stable/3996977?seq=33 (accessed on 6 March 2024)].

**Holotype: PAL-Gr 30997**—Greece, “Peloponnes, Messenien, Ep. Pilia: Bucht Vodhokoilia zwischen Petrohori und Paleokastro Navarinou. Ölhaine; 0–5 m. Blüten weiß,” 17 Apr 1979, *Greuter 17203* and *Merxmüller* (isotype: B).

***Silene orbelica*** Greuter in Willdenowia 25: 126. 1995 [https://www.jstor.org/stable/3996977?seq=22 (accessed on 6 March 2024)].

**Isotype: PAL-Gr 31507**—Greece, “Macedonia or., prov. Dhrama: mons Orvilos, in crista meridionali verticis Trisla supra refugium veterum Preseki, alt. 1450 m (41°20′30″ N, 23°38′ E), In clivis graminosis, solo calcareo,” 21 Aug 1978, *Greuter 16619* (holotype: B; other isotypes: AAU, C, G, M, UPA).

***Silene parnassica*** subsp. ***vourinensis*** Greuter in Willdenowia 14: 47. 1984 [https://www.jstor.org/stable/3995694?seq=11 (accessed on 6 March 2024)].

**Isotype: PAL-Gr 31764**—Greece, “Macedonia occ., inter prov. Kozani et Grevena: montes Vourinos, in summo vertice Vourinos ejusque crista septemtrionali, alt. 1600–1850 m (40°10′30″ N, 21°40′ E). In fissuris et scansilibus rupium serpentinicarum septemtriones spectamntium,” 26 Jul 1978, *Greuter 16349* (holotype: B; other isotypes: AAU, C, G, UPA, hb. Düll).

***Silene pusilla*** subsp. ***tymphaea*** Greuter in Willdenowia 25: 131, 132. 199 [=***Heliosperma pusillum*** (Waldst. and Kit.) Rchb. 1844] [https://www.jstor.org/stable/3996977?seq=27 (accessed on 6 March 2024)].

**Holotype: PAL-Gr 20697**—Greece, “Epirus, prov. Ioannina, distr. Dhodhoni: montes Timfi, in latere boreo-occidentali montis Ploskos, alt. 1800–1900 m. Ad rupes calcareas umbrosas. Flores albi,” 20 Aug 1974, *Charpin 11294*, *Dittrich*, *Greuter 12466* and *von Auw* (isotypes: ATH, G, UPA).

***Silene vulgaris*** subsp. ***suffrutescens*** Greuter, Matthäs, and Risse in Willdenowia 14: 34. 1984 [https://www.jstor.org/stable/3995693?seq=8 (accessed on 6 March 2024)].

**Isotype:PAL-Gr 17514**—Greece, Crete, “Ep. Temenos: ad cacumen montis Júktas, 700 m. In praeruptis rupestribus calcareis ad occidentem spectantibus haud raro. Plane suffruticosa, ex axillis foliorum emortuorum fasciculos foliorum juvenilium proferens,” 14 Sep 1966, *Greuter 7514* (holotype: G; other isotypes: B, E, LD, W).

***Silene vulgaris*** subsp. ***vourinensis*** Greuter in Willdenowia 25: 122. 1995 [https://www.jstor.org/stable/3996977?seq=18 (accessed on 6 March 2024)].

**Isotype: PAL-Gr 31777**—Greece, “Macedonia occ., inter prov. Kozani et Grevena: montes Vourinos, in summo vertice Vourinos ejusque crista septemtrionali, alt. 1600–1850 m (40°10′30″ N, 21°40′ E). In pratis et pascuis rupestribus, solo ophiolithico,” 26 Jul 1978, *Greuter 16333* (holotype: B; other isotypes: none).

***Smyrnium rotundifolium*** var. ***ovatifolium*** Halácsy, Consp. Fl. Graec. 1: 658. 1901 [https://www.biodiversitylibrary.org/item/40146#page/670/mode/1up (accessed on 6 March 2024)].

**Isolectotypes** (2): **PAL-Gr 8365 and 8366**—Greece, Crete, “Amalos,” 12 Jun 1884, *Reverchon* [Pl. Cret. #249 &] in Baenitz, Herb. Europ. [#5239] (lectotype [specified here]: WU #76665; other isotypes: WU #76664, etc.).

***Solenopsis minuta*** subsp. ***annua*** Greuter, Matthäs, and Risse in Willdenowia 14: 30. 1984 (≡***Solenopsis annua*** (Greuter, Matthäs, and Risse) Hand and Christodoulou 2020) [https://www.jstor.org/stable/3995693?seq=4 (accessed on 6 March 2024)].

**Isotype: PAL-Gr 34297**—Greece, Crete, “Eparchie Rethimni: Tal des Sfakoriako S der Prasiano-Schlucht, 35°18′30″ N, 24°33′ E. Grasflächen und Pterideten auf Schiefer, 100–120 m,” 24 May 1982, *Greuter 19301*, *Matthäs* and *Risse* (holotype: B; other isotypes: none known).

***Sphaerolobium pulchellum*** Meisn. in Bot. Zeitung (Berlin) 13: 28. 1855 [https://www.biodiversitylibrary.org/page/33612450#page/30/mode/1up (accessed on 6 March 2024)].

**Isotype: PAL-Gr 39674**—Australia, [Western Australia, “among the *Eucalypti* thickets on the Gardner and Fitzgerald rivers”] (from holotype label), [*Drummond* VI] *19* (holotype [specified here]: BM #550676; other isotypes: G, LD, K, MEL, NWS, NY, P).

***Staehelina arborea*** Schreb., Icon. Descr. Pl.: ad t. 1. 1766 [=***Staehelina petiolata*** (L.) Hilliard and B. L. Burtt 1973] [https://www.bavarikon.de/object/bav:SBB-BOT-00000BAV80005701?lang=de (accessed on 6 March 2024)].

**Isolectotype** (lectotype fragment): **PAL-Gr 6005**—[Greece, Crete], no label data, H[erbarium] G[undelsheimer]]; (lectotype [designated here]: M #30481).

***Statice doerfleri*** (‘*Dörfleri*’) Halácsy in Allg. Bot. Z. Syst. 5: 1. 1899 (***Limonium doerfleri*** (Halácsy) Rech. f.) [https://www.biodiversitylibrary.org/page/10085726#page/19/mode/1up (accessed on 6 March 2024)].

**Isolectotype: PAL-Gr 5268**– Greece, “Insula Denusa,” 10 Jun 1898, *Leonis* in Dörfler, Fl. Aegaea #172 (lectotype [designated by Brullo and Erben in Phytotaxa 240: 177. 2016]: M #173917; other isolectotypes: B [2×], FI [3×], FR, MPU).

***Stipa mayeri*** Martinovský in Acta Bot. Croat. 30: 145. 1971 [https://hrcak.srce.hr/file/230870 (accessed on 6 March 2024)].

**Isotype: PAL-Gr 7758**—Serbia, Kosovo, “SW-Serbia (Kosmet): Miruša sub monte Koznik, in declivibus lapidosis glareosis, solo serpent., cca. 550 m s.m.,” 30 May 1968, *Mayer* (holotype: LJU #63171; other isotype: BM).

***Styphelia crassiflora*** F. Muell., Fragm. 6: 40. 1867 (≡***Leucopogon crassiflorus*** (F.Muell.) Benth. 1868) [https://www.digitale-sammlungen.de/en/view/bsb10302569?page=46,47 (accessed on 6 March 2024)].

**Isotype: PAL-Gr 39664**—Australia, [Western Australia, “Between Moore and Murchison Rivers” (from holotype label)], [*Drummond* VI] *120* (holotype [specified here]: MEL #75843; other isotypes: BM, L, NY, PERTH).

***Tabebuia triorbicularis*** var. ***obovata*** Borhidi in Acta Bot. Hung. 26: 18 1980. [=**Tabebuia calcicola** Britton] [http://real-j.mtak.hu/3636/ (accessed on 6 March 2024)].

**Isotype: PAL-Gr 123178**—Cuba, “Prov. Sancti Spíritus. Mun. Trinidad: Trinidad, Loma del Mirador, en la carretera a Topes de Collantes, 600 m,” 3 May 1977, *Bisse & al.* in HFC 34683 (holotype [specified here]: HAJB #882; other isotypes: B, BP, HAC, HAJB, JE).

***Taraxacum basalticum*** Soest in Bull. Soc. Echange Pl. Vasc. Eur. Occid. Bassin Médit. 15: 111. 1974 [9270].

**Isotype: PAL-Gr 9270**—France, “Saint-Georges-sur-Allier (France, Puy-de-Dôme), alt. 600 m, pelouse sur basalte,” 26 Apr 1973, *Billy* in Soc. Echange Pl. Vasc. Eur. #7090 (holotype: LG other isotypes: GENT, LD, MA [2×], MSB, P, SEV).

***Taraxacum caudatuliforme*** Soest in Proc. Kon. Ned. Akad. Wetensch., Ser. C, Biol. Med. Sci. 69: 467. 1966 [30620].

**Isotype: PAL-Gr 30620**—Bulgaria, “ad versuras, 13 km E Plovdiv (Philippopel),” 18 Apr 1961, *Rechinger 21758* (holotype [specified here]: W #1981–10461; other isotypes: B, G, L, hb. Soest).

***Taraxacum euranum*** Soest in Bull. Soc. Echange Pl. Vasc. Eur. Occid. Bassin Médit. 16: 130. 1976 [23785].

**Isotype: PAL-Gr 23785**—France, “Jaudrais (France, Eure-et-Loire), friche argileuse humide le long de la route de Senonches (D 140),” 23 Apr 1975, *de Retz 70916* in Soc. Echange Pl. Vasc. Eur. #8011 (holotype: LG; other isotypes: C, G, GENT, M, MA [2×], MSB, P [2×], SEV).

***Taraxacum exiguum*** Soest in Bull. Soc. Echange Pl. Vasc. Eur. Occid. Bassin Médit. 16: 130. 1976 [23786].

**Isotype: PAL-Gr 23786**—France, “Villliers-en-Arthies (France, Val-d’Oise), prairie rase et clairière du bois de la Bucaille,” 25 Apr 1975, *de Retz 70935* in Soc. Echange Pl. Vasc. Eur. #8012 (holotype: LG; other isotypes: C, G, GENT, MA [2×], MSB, P, SEV).

***Taraxacum graniticum*** Soest in Bull. Soc. Echange Pl. Vasc. Eur. Occid. Bassin Médit. 16: 131. 1976 [23787].

**Isotype: PAL-Gr 23787**—France, “Gelles (France, Puy-de-Dôme), près de Say-Soubre, alt. 750 m, pré fumé sur granite,” 4 May 1975, *Billy* in Soc. Echange Pl. Vasc. Eur. #8013 (holotype: LG; other isotypes: C, G, GENT, LD, M, MA, MSB, P, SEV).

***Taraxacum inconspicuum*** Soest in Bull. Soc. Echange Pl. Vasc. Eur. Occid. Bassin Médit. 15: 111. 1974 [9266].

**Isotype: PAL-Gr 9266**—France, “Saint-Genés-Champanelle (France, Puy-de-Dôme), alt. 850 m, pré sur granite, à Thèdes,” 13 May 1973, *Billy* in Soc. Echange Pl. Vasc. Eur. #7094 (holotype: LG: other isotypes: G, GENT, L, LD, MA [2×], MSB, P [2×], SEV).

***Taraxacum retortum*** Soest in Bull. Soc. Echange Pl. Vasc. Eur. Occid. Bassin Médit. 16: 131. 1976 [23797].

**Isotype: PAL-Gr 23797**—France, “Urdos (France, Pyrénées-Atlantiques), le long de la route du Col du Somport, 200 m avant la frontière avec l’Espagne, alt. 1600 m,” 13 Jun 1975, *de Retz 71454* in Soc. Echange Pl. Vasc. Eur. #8023 (holotype: LG; other isotypes: C, G, GENT, L, LD, MA [2×], MSB, P [3×], SEV).

***Taraxacum roseolepis*** in Bull. Soc. Echange Pl. Vasc. Eur. Occid. Bassin Médit. 16: 132. 1976 [23799].

**Isotype: PAL-Gr 23799**—France, “Villiers-en-Arthies (France, Val-d’Oise), prairie arbustive le long de la route de Villarceaux (D 142),” 24 Apr 1975, *de Retz 70938* in Soc. Echange Pl. Vasc. Eur. #8025 (holotype: LG; other isotypes: C, G, GENT, LD, M, MA [2×], MSB, P, SEV).

***Telephium imperati*** var. ***pauciflorum*** Greuter in Candollea 20: 179. 1966 (≡***Telephium imperati*** subsp. ***pauciflorum*** (Greuter) Greuter and Burdet in Willdenowia 12: 191. 1982) [https://www.e-periodica.ch/digbib/view?pid=can-002%3A1965%3A20#183 (accessed on 6 March 2024)].

**Holotype: PAL-Gr 13683**—Greece, Crete, “Ep. Lassithi: N-Hang des Berges Spathì von der Alp Lekanìda bis zum Gipfel, 1650–2150 m.ü.M., Kalk. Auf lockerem u. verfestigtem Schutt verbreitet u. meist hfg.,” 23 Jun 1961, *Greuter 3683* (isotypes: G, LD, W, Z).

***Tetrazygia decorticans*** Bécquer in Willdenowia 37: 314. 2007. (≡***Miconia decorticans*** (Bécquer) Bécquer and Majure) [https://doi.org/10.3372/wi.37.37120 (accessed on 6 March 2024)].

**Isotype: PAL-Gr 121242**—Cuba, prov. Sancti Spíritus “Reserva Ecológica Alturas de Banao, base de la Teta de Juana, orillas del arroyo del tunel de Caja de Agua,” 640 m, 21°51’50.1″ N, 79°35’59.7″ W (holotype [specified by Bécquer in Revista Jard. Bot. Nac. Univ. Habana 44: 189. 2023]: HAJB #897; other isotypes: B, FLAS, GH, HAC, HAJB ##898–899 [2×], JE [2×], K, MO, NY, US).

***Teucrium chamaedrys*** var. ***oxyodon*** Heldr. and Halácsy in Oesterr. Bot. Z. 47: 324. 1897 [https://www.jstor.org/stable/23653429?typeAccessWorkflow=login (accessed on 6 March 2024)].

**Isolectotype: PAL-Gr 5249**—Greece, “m. Malevo Laconiae,” Jun 1896, [*Leonis*] in Heldreich, Pl. Exsicc. Fl. Hellen. (lectotype [designated here]: WU (Halácsy, Greek hb.); other isolectotypes: none traced).

***Trifolium eriosphaerum** Boiss.*, Diagn. Pl. Orient. 9: 25. 1849 [https://archive.org/details/e.-boissier-diagnoses-plantarum-orientalium-novarum-series-secunda.-n.o-8-13-1842-1854/page/n163/mode/2up?q=trifolium (accessed on 6 March 2024)].

**Isotype: PAL-Gr 39366**—Israel, “Jerusalem,” May 1846, *Boissier* (holotype [specified here]: G-BOIS #G404534; other isotypes: JE [2×], K, MEL).

***Trifolium michaelis*** Greuter in Bocconea 25: 115. 2012 [https://www.herbmedit.org/bocconea/25_005.pdf#page=111 (accessed on 6 March 2024)].

**Holotype: PAL-Gr 53188**—Greece, “Nom. Arkadia, Ep. Megalopolis: NE foothills of the Taijetos range, c. 0.5 km W of Neochori, alt. c. 1100 m, 37°11′0″ N, 22°13′45″ E. Mixed fir, pine and chestnut wood by a stream, on mixed substrate,” 11 Jun 1995, *Kamari* & al. in OPTIMA Iter Medit. 7 #2188 (isotypes: B, UPA).

***Turnera diminuta*** Greuter and R. Rankin in Revista Jard. Bot. Nac. Univ. Habana 43: 125. 2022. [https://revistas.uh.cu/rjbn/article/view/85/7179 (accessed on 6 March 2024)].

**Holotype: PAL-Gr***** 40262**—*Cuba, “Prov. Camagüey, km 3 de la carretera a Lesca al nordeste de Camagüey, alt. 120 m, 21°27′10″ N, 77°50′00″ W. Matorral xeromorfo espinoso degradado en serpentina,” 2 Mar 2000, *Greuter 25137 & al.* (isotypes: B, HAJB, HIPC, US).

***Ventenata subenervis*** Balansa in sched. Pl. d’Orient, 1854: #7. 1854 [8494].

**Isotype: PAL-Gr 8494**—Turkey, “Smyrne, sur les collines incultes,” 8 May 1854, *Balansa* in Pl. d’Orient, 1854 #7 (holotype [specified here]: P #3645076; other isotypes: B, BM, BR [2×], FI, FR, G [2×], GOET, K [3×], KFTA, MPU, P [15×], US, W [3×], WAG).

***Veronica stamatiadae*** M. A. Fisch. and Greuter in Pl. Syst. Evol. 125: 249. 1976 [27056].

**Isotype: PAL-Gr 27056**—Greece, “Insula Kastellorizo (Megisti) Dodekanesi, Insula Ro: ad orientem sinûs Ajios Jeorjios, alt. 10–30 m. In rupestribus calcareis dumulosis. Ad rupes,” 12 Apr 1974, *Greuter 11808* (holotype [specified here]: WU #70317; other isotypes: ATH, C, ERE, G, H, WU #70316).

***Vicia davisii*** Greuter in Willdenowia 8: 565. 1979 [https://www.jstor.org/stable/3996168?seq=35 (accessed on 6 March 2024)].

**Holotype: PAL-Gr 27107**—Greece, “Insula Kastellorizo (Megisti) Dodekanesi, in scopulo Psomi, alt. c. 15 m. In rupestribus calcareis. Flores coeruleo-violascentes,” 17 Apr 1974, *Greuter 11938* (isotypes: ATH, G).

***Vicia sericocarpa*** var. ***microphylla*** Boiss., Fl. Orient. 2: 571. 1872 (*Vicia podocarpa* Boiss. and Hausskn., l.c., pro syn.) [https://www.biodiversitylibrary.org/page/18113023#page/578/mode/1up (accessed on 6 March 2024)].

**Isolectotype: PAL-Gr 66704**—Syria “borealis. Aleppo. Très remarquable par ses feuilles courtes, son calyce très-obliquement [tronqué] et son fruit longuement stipité dans le calyce. A placer [près] du *V. sericocarpa*! [manu Boissier],” 1865, *Haussknecht* (lectotype [designated here]: JE #6825; other isolectotypes: G-BOIS, JE # 6827).

***Viola orphanidis*** var. ***cyanea*** Hausskn. in Mitth. Thüring. Bot. Vereins, ser. 2, 5: 44. 1893 [https://books.google.it/books?id=APQZAAAAYAAJ&printsec=frontcover&hl=it#v=onepage&q&f=false (accessed on 6 March 2024)].

**Isotype: PAL-Gr 66362**—Greece, “Agrapha (Dolopia veterum): in reg. super. Pindi summi montis Karáva alt 5500–6500′ substratu schistoso,” 1–3 Jul 1885, *Haussknecht* (holotype [specified here]: JE #20586–20587 [2 sheets]).

***Zanthoxylum rolandii**** Beurton in* Feddes Repert. 97: 35. 1986 [126212].

**Isotype: PAL-Gr 126212**—Cuba, “Prov. Pinar del Río, al pie oeste de la Loma Peluda (Preluda) de Cajálbana, cuabales, serpentina,” 21 Jan 1981, *Berazaín & al.* in HFC 43476 (holotype [specified here]: HAJB ##958; other isotypes: B [2×], HAJB #959–961 [3×], JE [2×]).

## 5. Conclusions

The type herbarium in PAL-Gr is composed of four packages with 338 sheets, corresponding to 335 specimens, three of them of two sheets each (*Code*, Art. 8.3), and pertaining to 327 different names. Grouped by type categories, there are 61 holotypes (+2 fragments), 6 second-step holotypes, 236 isotypes, 3 lectotypes (+1 fragment), 17 isolectotypes, 1 neotype, 2 isoneotypes, and 1 isoepitype. Of these, 82 are here newly designated (or, in 67 cases, specified), but 22 remain unspecified first-step holotypes, as the information necessary for the specification of a second-step type is lacking.

A grouping of the type specimens (not counting duplicates) by country/region of origin reveals the large predominance of Mediterranean, principally Greek material:
Albania7Armenia5Bosnia and Herzegovina1Bulgaria1Cyprus1Greece150France13Israel3Italy70Libya2Montenegro3Morocco1Portugal2Serbia3Spain9Syria1Turkey2Σ Mediterranean274Austria4Germany4Σ Central Europe8Australia13Cuba31USA1Σ Extra-European45Σ Total327

## Data Availability

Data are contained within the article.

## References

[B1-plants-13-01086] Turland N.J., Wiersema J.H., Barrie F.R., Greuter W., Hawksworth D.L., Herendeen P.S., Knapp S., Kusber W.-H., Li D.-Z., Marhold K. (2018). International Code of Nomenclature for Algae, Fungi, and Plants (Shenzhen Code), Adopted by the Nineteenth International Botanical Congress, Shenzhen, China, July 2017.

