# Peer review of "The Greuter Herbarium in Palermo: An Inventory of Its Type Specimens Available Online, with Some Thoughts on Type Terminology (Occasional Papers from the Herbarium Greuter, N° 5)"

_plants, 2024, doi:10.3390/plants13081086_

Round 1

Reviewer 1 Report

Comments and Suggestions for Authors

Title: The Greuter Herbarium in Palermo: An Inventory of Its Type Specimens Available Online, with Some Thoughts on Type Terminology (Occasional Papers from the Herbarium Greuter, N° 5)

Author: Werner Greuter

This is an interesting, and well-written manuscript in which the author of the manuscript presents an Inventory of the Type specimens of his personal Herbarium, the Greuter Herbarium in Palermo (PAL-Gr), including transcribed label data, information on the type duplicates in other Herbaria, and links to the digital specimen images and to the protologue texts.

            Abstract, Keywords, and Literature cited are pertinent and up-to-date. Typification section is, in general, adequate and well documented.

The author of the manuscript is a well-known expert in taxonomy and nomenclature of vascular plants.

Therefore, I recommend the publication of this manuscript and consider it appropriate to be published in the journal Plants MDPI though this last aspect should be, obviously, decided by the editor.

Some aspects of the manuscript should be also revised or modified. I have only reviewed in depth the information on the nomenclatural types that begin with the letters A and B and I have observed that the author must complete the information throughout the manuscript regarding the duplicates of the mentioned types present in other Herbaria.

Some other minor points that should be corrected are marked in red in the manuscript.

Santiago Ortiz

Author Response

Thank you for review's comments. I found them to be useful nd to the point, so I followed  suggestions in all cases but one (his marginal note to Campanula pinatzii), as this was in conflict with the vews I had expressed elsewhere in the paper).

Reviewer 2 Report

Comments and Suggestions for Authors

Thank you for the opportunity to review this. 

This paper is of importnace to "clasical" taxonomists and nomenclaturalists, but will not find a readsership beyond that realm. 

HOWEVER, it does address an interesting lacunae in the Code and the suggested solution presented here is logical and in my opinion not controversial. I hope the formal proposal to ammend the code accordingly will eventually be submitted and ultimateny passed at a future IBC nomenclature session. 

Comments on the Quality of English Language

I have made a few minor edits concrning the English on the pdf version (uploaded). I have also suggested some formatting changes.

Author Response

Thank you for review's comments. I have found your suggestions positive and helpful and have incorporated all of them into the manuscript. The revised manuscript is attached.
